

# ACCESS-OM2: A Global Ocean-Sea Ice Model at Three Resolutions

Andrew E. Kiss[1,2], Andrew McC. Hogg[1,2], Nicholas Hannah[3], Fabio Boeira Dias[2,4,5,6], Gary
B. Brassington[7], Matthew A. Chamberlain[4], Christopher Chapman[4], Peter Dobrohotoff[4,5], Catia
M. Domingues[2,5,6], Earl R. Duran[8], Matthew H. England[2,8], Russell Fiedler[4], Stephen M. Griffies[9,10],
Aidan Heerdegen[1,2], Petra Heil[6,11], Ryan M. Holmes[2,8,12], Andreas Klocker[2,6], Simon J. Marsland[2,4,5,6],
Adele K. Morrison[1,2], James Munroe[13], Peter R. Oke[4], Maxim Nikurashin[2,5], Gabriela S. Pilo[2,5],
Océane Richet[4,14], Abhishek Savita[2,4,5,6], Paul Spence[2,8], Kial D. Stewart[1,8], Marshall L. Ward[9,15],
Fanghua Wu[16], and Xihan Zhang[1,2]

[1]Research School of Earth Sciences, Australian National University, Canberra, Australia
[2]ARC Centre of Excellence for Climate Extremes, Australia
[3]Double Precision, Sydney, Australia
[4]CSIRO Oceans and Atmosphere, Australia
[5]Institute for Marine and Antarctic Studies, University of Tasmania, Hobart, Australia
[6]Antarctic Climate and Ecosystems Cooperative Research Centre, Hobart, Australia
[7]Bureau of Meteorology, Melbourne, Australia
[8]Climate Change Research Centre, University of New South Wales, Sydney, Australia
[9]NOAA Geophysical Fluid Dynamics Laboratory, Princeton, New Jersey, USA
[10]Atmospheric and Oceanic Sciences Program, Princeton University, Princeton, New Jersey, USA
[11]Australian Antarctic Division, Kingston, Tasmania, Australia
[12]School of Mathematics and Statistics, University of New South Wales
[13]Memorial University of Newfoundland, St John's, Canada
[14]Centre for Southern Hemisphere Ocean Research, Hobart, Tasmania, Australia
[15]National Computational Infrastructure, Australian National University, Canberra, Australia
[16]Beijing Climate Centre, Beijing, China

**Correspondence:** Andrew E. Kiss (Andrew.Kiss@anu.edu.au)

**Abstract.** We introduce a new version of the ocean-sea ice implementation of the Australian Community Climate and Earth System Simulator, ACCESS-OM2. The model has been developed with the aim of being aligned as closely as possible with the fully coupled (atmosphere-land-ocean-sea ice) ACCESS-CM2. Importantly, the model is available at three different horizontal resolutions: a coarse resolution (nominally 1° horizontal grid spacing), an eddy-permitting resolution (nominally 0.25°) and an eddy-rich resolution (0.1° with 75 vertical levels), where the eddy-rich model is designed to be incorporated into the Bluelink operational ocean prediction and reanalysis system. The different resolutions have been developed simultaneously, both to allow testing at lower resolutions and to permit comparison across resolutions. In this manuscript, the model is introduced and the individual components are documented. The model performance is evaluated across the three different resolutions, highlighting the relative advantages and disadvantages of running ocean-sea ice models at higher resolution. We find that higher resolution is an advantage in resolving flow through small straits, the structure of western boundary currents and the abyssal overturning cell, but that there is scope for improvements in sub-grid scale parameterisations at the highest resolution.



## 1   Introduction

Ocean-sea ice models have extensive applications. They form the oceanic component of coupled climate and Earth system models that are used for projecting future climate, and can incorporate biogeochemical and ecosystem dynamics which extend the realm of application. They are also needed for forecasting on shorter timescales — both forecasting in the ocean and for
seasonal prediction of the ocean/sea ice/atmosphere state. As a research tool, ocean-sea ice models can be used to quantitatively test, or experiment with, the dynamics of the climate system; such process studies have been invaluable in forming a broad understanding of the drivers of climate change and variability.

Modelling studies face the challenge of compromising between resolving critical processes and computational expense. For example, the standard grid spacing for the ocean component of coupled climate models is 1°, with indications that some models
being prepared for the next Coupled Model Intercomparison Project (CMIP6) will use 0.25° horizontal spacing. However, 1° models do not resolve the ocean mesoscale, meaning that they miss key processes that can influence the climate. Higher resolution models usually have improvements in the climate state with better estimates of vertical heat transport (Griffies et al., 2015), enhancement of boundary currents (Hewitt et al., 2016), better resolution of ocean straits and improved Southern Ocean state (Bishop et al., 2016). On the other hand, high resolution simulations consume huge computational resources, which
limits the length of runs and the capacity to optimise the model configuration (or minimise biases) by testing the model over a wide parameter space. There is also less experience in the coupled ocean-atmosphere-ice modelling communities in the integration of these high resolution ocean models for climate simulations. Thus, while higher resolution models are becoming computationally feasible, the additional resolution does not necessarily result in improved simulations.

One of the complexities in characterising model performance as a function of resolution is the influence of model biases
governing the model state. It is well known, for example, that different models subjected to the same atmospheric state produce differing mean states (e.g. Griffies et al., 2009; Danabasoglu et al., 2014). It follows that investigating the effects of model resolution requires a clean hierarchy; a model suite in which variations in resolution are available with homogeneous code, forcing and, as far as possible, parameter choices. This is a technique successfully employed by the DRAKKAR consortium (Barnier et al., 2014) as well as climate model developers (e.g. Griffies et al., 2015; Hewitt et al., 2016).

In this manuscript we outline development of the latest version of the ocean-sea ice component of the Australian Community Climate and Earth System Simulation, known as ACCESS-OM2. This model was developed to serve the twin aims of underpinning climate model development and ocean state forecasting in Australia; it therefore includes parallel development of low, medium and high resolution options. It is based on the ocean-sea ice components of the Australian Community Climate Earth System Simulator (ACCESS), which was originally formulated for coupled climate simulations (Bi et al., 2013a),
and therefore designed to support Australian efforts in developing models for the upcoming Coupled Model Intercomparison Project (CMIP6). The high-resolution configuration of ACCESS-OM2 is intended for use in the next version of the Ocean Forecasting Australia Model (OFAM4), a component of the Australian Bureau of Meteorology's operational ocean reanalysis and forecasting system (Bluelink; see Oke et al., 2013, and references therein). Finally, the model suite is also intended for use in ocean and sea ice process studies and sensitivity tests.



In this manuscript we aim to document the model formulation (Sect. 2) and the computational performance of the model at each resolution (Sect. 3). We also undertake a preliminary evaluation of the global model state (Sect. 4.1), of the circulation in selected regions (Sect. 4.2) and of the model representation of sea ice (Sect. 4.3). In Sect. 5 we summarise and conclude the study.

## 2    Model formulation

Model configurations at three horizontal resolutions have been developed, named ACCESS-OM2 (nominally 1° horizontal grid spacing), ACCESS-OM2-025 (nominally 0.25° spacing) and ACCESS-OM2-01 (nominally 0.1° spacing). The suite of three resolutions is also collectively referred to as ACCESS-OM2. Configurations (e.g. run parameters and forcing) are as consistent as possible across the three resolutions to facilitate studies of resolution dependence and sub-gridscale parameterisations. The coarser models serve as testbeds for developing configurations at higher resolutions, and are suitable for long experiments covering climatological timescales of hundreds of years, but lack an explicit representation of mesoscale eddies. In contrast, the ACCESS-OM2-01 configuration resolves the first baroclinic deformation radius away from shelves and equatorward of about 50° (Hallberg, 2013; Stewart et al., 2017), and therefore represents an active transient mesoscale eddy field in most of the world ocean.

ACCESS-OM2 consists of two-way coupled ocean and sea ice models driven by a prescribed atmosphere (see Fig. 1). The ocean model component is the Modular Ocean Model (MOM) version 5.1 from the Geophysical Fluid Dynamics Laboratory (https://mom-ocean.github.io), and the sea ice component (https://github.com/COSIMA/cice5/) is a fork from the Los Alamos sea ice model (CICE) version 5.1.2 from Los Alamos National Laboratories (https://github.com/CICE-Consortium/CICE-svn-trunk/tree/cice-5.1.2). For brevity we refer to these as MOM5 and CICE5 below. These components are forced by prescribed atmospheric conditions taken from the 55-year Japanese Reanalysis for driving oceans (JRA55-do, Tsujino et al., 2018) via YATM (https://github.com/COSIMA/libaccessom2/). The model components are coupled via Ocean Atmosphere Sea Ice Soil (OASIS3-MCT) version 2.0 from CERFACS and CNRS, France (https://portal.enes.org/oasis). The ACCESS-OM2 model source code and configurations are hosted at https://github.com/COSIMA/access-om2; the specific versions used in this paper are at doi:10.5281/zenodo.2653246. The following subsections provide further details on these model components.

### 2.1    MOM5 ocean model

MOM5 is a hydrostatic primitive equation ocean model with a free surface discretized using the Arakawa B-grid for the horizontal stencil along with a variety of vertical coordinate options (Griffies, 2012). We make use of the Boussinesq (volume-conserving) version of the code and choose the $z^*$ vertical coordinate. The model is derived from a long history of use in climate and ocean modelling (documented in an unpublished manuscript available from https://mom-ocean.github.io/assets/pdfs/mom_history_2017.09.19.pdf) and is comprehensively documented (Griffies, 2012); in this manuscript we therefore focus on detailing aspects of the MOM configuration that are specific to ACCESS-OM2.



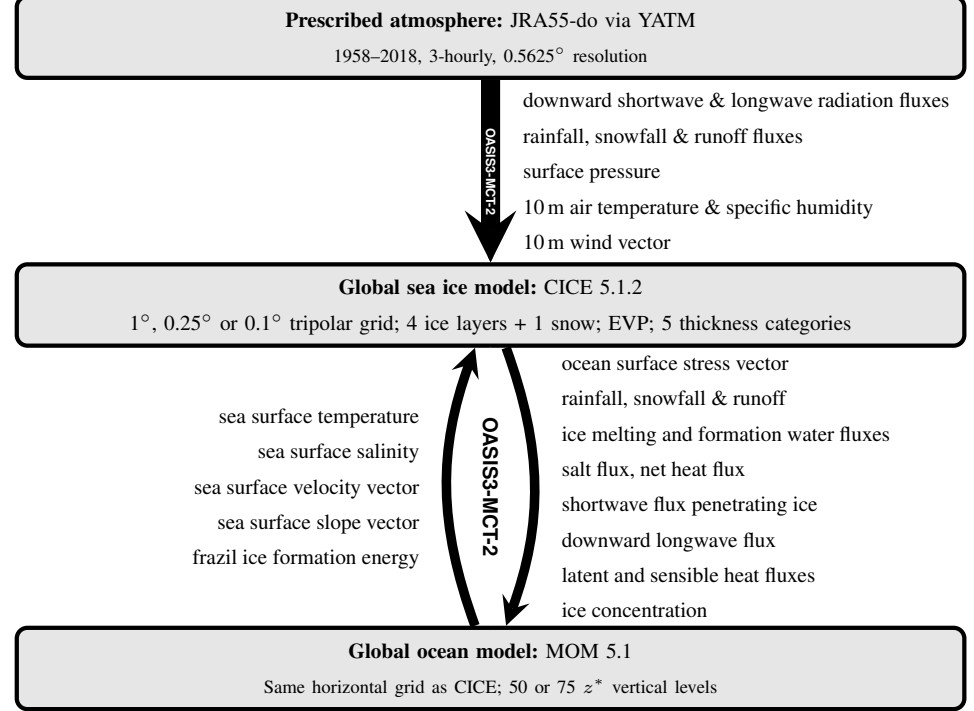

**Figure 1.** ACCESS-OM2 model components and coupling fields. OASIS3-MCT-2 is used to couple the model components. Notice that MOM 5.1 receives atmospheric forcing via CICE 5.1.2 rather than directly from YATM (CICE 5.1.2 is configured with the same global domain and horizontal grid as MOM 5.1). Surface pressure is also passed from CICE 5.1.2 to MOM 5.1 but is not used by MOM 5.1 in the current configuration.

### 2.1.1 Vertical grid

The configurations use a $z^*$ vertical coordinate (Stacey et al., 1995; Adcroft and Campin, 2004) with partial cells (Adcroft et al., 1997; Pacanowski and Gnanadesikan, 1998). The vertical grids are optimised for resolving baroclinic modes, based on the KDS grids recommended by Stewart et al. (2017). The vertical grids in ACCESS-OM2 and ACCESS-OM2-025 are slightly

5  modified versions of KDS50 with 50 levels and 2.3 m spacing at the surface, increasing smoothly to 219.6 m by the bottom at 5363.5 m. The vertical grid in ACCESS-OM2-01 is a slightly modified version of KDS75, with 75 levels and 1.1 m spacing at the surface, increasing smoothly to 198.4 m by the bottom at 5808.7 m. Details of the vertical discretisation can be found in Table 1.

### 2.1.2 Horizontal grid

10  In the horizontal direction, MOM5 and CICE5 both use the same orthogonal curvilinear Arakawa B-grid with velocity components co-located at the northeast corner of tracer cells. Model configurations have been developed with zonal grid spacings of





**Table 1.** Vertical grid parameters: $n$ levels, with spacing of $\Delta z_{\mathrm{min}}$ and $\Delta z_{\mathrm{max}}$ at the surface and maximum depth $H_{\mathrm{max}}$, respectively, and median spacing $\Delta z_{\mathrm{median}}$. The depth at level $n/2$ is denoted $H_{n/2}$.

| Model | $n$ | $\Delta z_{\mathrm{min}}$ (m) | $\Delta z_{\mathrm{median}}$ (m) | $\Delta z_{\mathrm{max}}$ (m) | $H_{n/2}$ (m) | $H_{\mathrm{max}}$ (m) |
|---|---|---|---|---|---|---|
| ACCESS-OM2 | 50 | 2.3 | 93.0 | 219.6 | 627 | 5363.5 |
| ACCESS-OM2-025 | 50 | 2.3 | 93.0 | 219.6 | 627 | 5363.5 |
| ACCESS-OM2-01 | 75 | 1.1 | 42.6 | 198.4 | 423 | 5808.7 |

$1°$, $0.25°$ and $0.1°$ south of $65°$ N. Globally, the median cell size is $92\,\mathrm{km}$, $18.1\,\mathrm{km}$ and $7.2\,\mathrm{km}$, respectively, at $1°$, $0.25°$ and $0.1°$ resolution. Although the CICE model is global, the sea ice is mostly confined to latitudes poleward of $60°$, where most cell dimensions are finer than $47.4\,\mathrm{km}$, $11.3\,\mathrm{km}$ and $4.5\,\mathrm{km}$, respectively, at the three resolutions.

The horizontal meshes are $360 \times 300$, $1440 \times 1080$ and $3600 \times 2700$ at $1°$, $0.25°$ and $0.1°$, respectively (see Table 2). In all
cases the grid extends from the North Pole to $81.1°$ S, covering the global ocean to the Antarctic ice shelf edge but omitting the Antarctic landmass. The longitude range is $-280°$ E to $+80°$ E, placing the join in the middle of the Indian Ocean. In all configurations the grid is tripolar (Murray, 1996), with poles placed on land at $65°$ N, $-100°$ E and $65°$ N, $80°$ E and the South Pole; consequently the grid directions are zonal and meridional only south of $65°$ N. In the $0.25°$ and $0.1°$ configurations the grid is Mercator (i.e. the meridional spacing scales as the cosine of latitude) between $65°$ N and $65°$ S; south of $65°$ S,
the meridional grid spacing is held at the same value as at $65°$ S. The meridional variation of meridional grid spacing is more complicated in the $1°$ model, and incorporates a refinement to $1/3°$ within $10°$ of the Equator (Bi et al., 2013b).

### 2.1.3   Bathymetry

The model bathymetry for ACCESS-OM2 makes use of legacy datasets from ACCESS-OM at $1°$ resolution (Bi et al., 2013b). ACCESS-OM2-025 uses a modified version of the GFDL-CM2.5 bathymetry as used by Griffies et al. (2005). Significant
effort was deployed to create a new model bathymetry file for ACCESS-OM2-01, based on version 20150318a of the GEBCO 2014 30 arc second grid (GEBCO, 2014). To obtain the depth of each model cell a simple mean of points with depth greater than zero within the cell was calculated along with the fraction of such wet points. After elimination of isolated lakes, the coasts were inspected by eye and hand edited to ensure major straits and channels (e.g. Lombok Strait) could be represented by the B-grid, small shallow inlets were removed and global connectivity was maintained. Further checks were also made to
eliminate non-advective cells and to ensure that the averaging process did not remove significant islands or smooth subsurface features too aggressively (e.g. Macquarie Island). In regions where ice shelves are found, topography ends at a vertical wall at the ice shelf edge (the calving line, not the grounding line); there are no ice shelf cavities as these are not supported in MOM5. The coarser two resolutions expanded the land mask to remove small wet cells close to the northern tripoles, but the ACCESS-OM2-01 bathymetry retains these cells and therefore includes the Gulf of Ob in Siberia and many additional
channels in the Canadian Archipelago. We use partial cells (Adcroft et al., 1997; Pacanowski and Gnanadesikan, 1998) to obtain a more accurate representation of bottom topography in all three configurations. In ACCESS-OM2 and ACCESS-OM2-



025 the minimum thickness of partial cells is 20% of the full cell thickness $\Delta z$. In ACCESS-OM2-01 the minimum thickness of partial cells is $0.2\Delta z$, or $\min(10\,\mathrm{m}, \Delta z)$, whichever is greater. The minimum water depth is 45.11 m (10 levels) in ACCESS-OM2, 40.36 m (9 levels) in ACCESS-OM2-025, and 10.43 m (7 levels) in ACCESS-OM2-01.

### 2.1.4 Parameterisations and equation of state

A sub-grid scale parameterization for mesoscale eddies is included in the ACCESS-OM2 and ACCESS-OM2-025 models but not in ACCESS-OM2-01 as this resolution is considered "eddy-resolving". In the two coarser configurations the Gent and McWilliams (1990) (GM) parameterization is used to represent the quasi-Stokes transport associated with mesoscale eddies (McDougall and McIntosh, 2001), and the neutral direction diffusive tracer transport is parameterised by a neutral diffusivity (Redi, 1982). The GM parameterization is implemented as a skew-diffusive flux (Griffies, 1998) and further formulated as a

boundary value problem by Ferrari et al. (2010). The associated diffusivity is depth-independent but flow-dependent, and is the product of an inverse timescale, a squared length scale, and a grid scaling factor as detailed in Section 3.3 of Griffies et al. (2005). The length scale is 50 km at $1°$ and 20 km at $0.25°$. The inverse timescale is an Eady growth rate determined from the horizontal density gradient averaged between 100 m and 2000 m using a constant buoyancy frequency of $0.004\,\mathrm{s}^{-1}$. The Eady growth rate is subject to a limiter and is smoothed both vertically and horizontally, and is vertically averaged in the mixed layer.

The GM diffusivity is also scaled in proportion to how well the numerical grid resolves the first baroclinic Rossby radius (or the equatorial Rossby radius within $\pm 5°$ latitude), as suggested by Hallberg (2013). The GM diffusivity is limited to the ranges $50$–$600\,\mathrm{m}^2\mathrm{s}^{-1}$ in ACCESS-OM2 and $1$–$200\,\mathrm{m}^2\mathrm{s}^{-1}$ in ACCESS-OM2-025.

  The neutral tracer diffusion is implemented according to Griffies et al. (1998) and is used in the two coarser configurations with a diffusivity that differs from GM. A constant coefficient of $600\,\mathrm{m}^2\mathrm{s}^{-1}$ is used in ACCESS-OM2, while in ACCESS-

OM2-025 the coefficient is scaled by the resolution of the grid relative to either the first baroclinic Rossby radius, or the equatorial Rossby radius for latitudes between $\pm 5°$ N, with a diffusivity no greater than $200\,\mathrm{m}^2\mathrm{s}^{-1}$.

  All three configurations include a parameterization for re-stratification in the surface mixed layer due to submesoscale eddies (Fox-Kemper et al., 2008). This parameterization applies an overturning circulation dependent on the horizontal buoyancy gradients within the mixed layer. The optional horizontal diffusive portion of this parameterization is not used.

Horizontal friction is implemented with a biharmonic operator and a horizontally isotropic Smagorinsky scaling for the viscosity coefficient (Griffies and Hallberg, 2000; Griffies, 2012). The ACCESS-OM2 configuration also has a grid spacing-dependent horizontally isotropic biharmonic background viscosity set by a velocity scale $0.04\,\mathrm{m\,s}^{-1}$, with the NCAR scheme applied to enhance the background horizontal viscosity near western boundaries in order to ensure the western boundary currents are resolved (see Section 3.4 of Griffies et al., 2005). ACCESS-OM2 also uses a Laplacian bottom viscosity set by a

velocity scale $0.01\,\mathrm{m\,s}^{-1}$. There is no background viscosity at the other resolutions. The overall biharmonic viscosity is limited to 25% of the numerical instability threshold in ACCESS-OM2, or 100% in the other two configurations. The lateral boundary condition for velocity is no-slip, as a consequence of using a B-grid (Griffies, 2012).

  A constant background vertical viscosity of $10^{-4}\,\mathrm{m}^2\mathrm{s}^{-1}$ is used at all resolutions. The background vertical tracer diffu-sivity is zero in ACCESS-OM2-025 and ACCESS-OM2-01, but at $1°$ it is dependent on latitude following Jochum (2009),





smoothly increasing from $1 \times 10^{-6} \mathrm{m^2 s^{-1}}$ at the Equator to a constant $5 \times 10^{-6} \mathrm{m^2 s^{-1}}$ poleward of $\pm 20°$ N. The K-profile parameterization (KPP, Large et al., 1994) determines additional vertical diffusivities of both tracers and momentum to represent mixing within the surface boundary layer, and also Richardson number-based shear instability (active mainly in the equatorial undercurrents), internal wave breaking and double-diffusion in the interior. KPP maintains static stability by applying large

vertical diffusivity in regions with small or negative Richardson number, obviating the need for explicit convective adjustment. There are no explicit tides, but bottom-enhanced internal tidal mixing is parameterized following Simmons et al. (2004) and barotropic tidal mixing is parameterized following Lee et al. (2006). We calculate bottom drag from the law of the wall using prescribed bottom roughness and spatially resolved but temporally constant tidal current speed, with residual $0.05 \mathrm{\,m\,s^{-1}}$. Overflow and downslope mixing schemes are not used.

At 1° Rayleigh damping is used to improve the Indonesian Throughflow transport; a damping timescale of 1.5 hr is applied at all but the bottom 2 (3) U-cells in Lombok (Ombai) Strait and for 3/4 of the width of the Torres Strait at all depths. At 0.1° a damping timescale of 1.5 hr is used at all depths across the full width of Kara Strait to constrain the velocity, which otherwise leads to numerical instability unless an unfeasibly small timestep is used. There is no Rayleigh drag in the 0.25° configuration.

We use the Jackett et al. (2006) pre-TEOS10 seawater equation of state and freezing temperature. The prognostic temperature

variable is conservative temperature in the 1° and 0.25° configurations, and potential temperature at 0.1°. All configurations use practical salinity as the prognostic variable for salt.

### 2.1.5   Numerical methods

All configurations use a baroclinic dynamics (and tracer) timestep that is 80 times longer than the barotropic timestep. The barotropic dynamics use a predictor-corrector method with dissipation parameter $\gamma = 0.2$ and Laplacian smoothing of sea

surface height. The baroclinic time stepping uses a two-level volume- and tracer-conserving staggered second-order forward method, with implicit vertical mixing and semi-implicit Coriolis calculations. Momentum advection is achieved via a second-order centred operator in space and third-order Adams-Bashforth time stepping. We use a multi-dimensional piece-wise parabolic scheme for tracer advection (Colella and Woodward, 1984), with a monotonicity-preserving flux limiter following Suresh and Huynh (1997).

## 2.2   CICE5 sea ice model

CICE5 is a thermodynamic-dynamic sea ice model, including advective transport of the state variables and an energy-conserving ridging parameterisation that transfers ice between thickness categories in response to the energy budget and strain rates. The CICE5 sea ice model is well documented (Hunke et al., 2015) so we only provide an overview of key aspects here. We use CICE version 5.1.2 with parameters that are largely based on those used for CICE4.1 in ACCESS-OM (Bi et al., 2013b). In

our configuration CICE5 uses the same horizontal tripolar Arakawa B-grid as MOM5, and its thermodynamic timestep is the same as the MOM5 baroclinic timestep (Table 2). For the thermodynamics we use four ice layers and one snow layer for each of the five thickness categories (discussed below). We use the Bitz and Lipscomb (1999) thermodynamics formulation at 1° and 0.25°, but at 0.1° this occasionally failed to converge so the mushy ice thermodynamics formulation of Turner et al. (2013)





was used instead in the highest-resolution simulation. Other thermodynamic parameters are the same as those listed in Bi et al. (2013b, table 2) for ACCESS-OM, including the use of the Pringle et al. (2007) thermal conductivity parameterisation, which improves the otherwise slow thermodynamic ice growth rate in the Antarctic (Hunke, 2010).

Horizontal advection of conserved properties is handled by the incremental remapping scheme of Dukowicz and Baum-
gardner (2000) and Lipscomb and Hunke (2004). Internal ice stresses are represented by a visco-plastic rheology, via the "classic" elasto-visco-plastic method (Hunke and Dukowicz, 1997, 2002; Hunke, 2001) in which a fictitious elastic term is added to facilitate efficient numerical convergence to the Hibler (1979) visco-plastic solution via damped elastic waves which are supposed to decay to negligible amplitude via elastic sub-timesteps within each dynamic timestep. For pragmatic reasons we follow Hunke and Dukowicz (2002) in using 120 elastic timesteps per dynamic timestep, but we note that this may be
insufficient to completely eliminate the elastic transients from the solution (Losch and Danilov, 2012; Lemieux et al., 2012; Kimmritz et al., 2017, 2015). The ice dynamics have the same timestep as the thermodynamics at the two coarser resolutions, but at $0.1°$ it was necessary to reduce the dynamic timestep to a third of the thermodynamic timestep, due to a more restrictive CFL condition because the land mask edge is closer to the northern poles in our tripolar grid, producing some very small grid cells. The resulting load imbalance was mitigated by allocating relatively more cores to CICE at $0.1°$ (Table 2).

In all three configurations the vertical grid resolution is sufficient to resolve the surface Ekman layer (Table 1), so we use a turning angle of zero, consistent with ACCESS-OM (Bi et al., 2013b, table 2). We use an ice-ocean drag coefficient of 0.00536, consistent with ACCESS-OM (Bi et al., 2013b, table 2) and very close to the value of 0.0054 measured at 0.5 m below first-year landfast ice by Shirasawa and Ingram (1997).

Importantly every model grid cell may contain a mixture of open water and sea ice, with the ice itself being split into a
number of thickness categories, chosen to represent the inhomogenous thickness distribution of sea ice. We use five thickness categories, with lower bounds of 0, 0.64, 1.39, 2.47 and 4.57 m. Following Thorndike et al. (1975) ice mass is moved between these categories as a function of ice motion and advection, the thermodynamic ice growth rate, and the ridging redistribution. We use the Lipscomb et al. (2007) ridging scheme, with $e$-folding scale parameter taking the default value $3\,\mathrm{m}^{1/2}$ (rather than $2\,\mathrm{m}^{1/2}$ as used in ACCESS-OM Bi et al., 2013b, table 2).

The CICE5 configurations of the final runs reported here subdivided the computational domain horizontally into tiles, with around 4 (6) tiles allocated to each CPU at $0.25°$ ($0.1°$) by a `roundrobin` distribution (Craig et al., 2015), which omits land-only tiles and improves the load balance by having a mix of ice-containing and ice-free tiles allocated to each CPU. At $1°$ we allocate one pole-to-pole meridional strip to each processor in the interests of load balancing. We also use halo masking at all resolutions, which eliminates MPI updates in ice-free halos.

## 2.3 Forcing

The ACCESS-OM2 configurations are forced with the latest JRA55-do v1.3 atmospheric product (Tsujino et al., 2018) which has significantly improved bias, spatial and temporal resolution (55 km, 3-hourly), temporal extent (1958–2018), and dynamical self-consistency compared to the Large and Yeager (2009) CORE-II dataset (200 km, 6-hourly, 1948–2009) used in many previous modelling studies. JRA55-do includes recent Antarctic calving and basal melt estimates from Depoorter et al. (2013),



which are spatially variable and somewhat larger than the uniform values in CORE-II (Tsujino et al., 2018). The improved spatial resolution of wind is important for better representation of coastal polynyas (Stössel et al., 2011; Zhang et al., 2015) and upwelling (Taboada et al., 2019). The temporal coverage currently extends from 1st January 1958 to 1st February 2018, but it is regularly updated to near present day; we use 1958–2017 inclusive for experiments described in this paper. JRA55-

do version 1.3 provides 3-hourly liquid and solid precipitation, downwelling surface longwave and shortwave radiation, sea level pressure, 10 m wind velocity components, 10 m specific humidity and 10 m air temperature on a TL319 grid ($0.5625°$ resolution), and daily river flux at $0.25°$ resolution.

Surface forcing is handled globally by CICE5, which then passes various forcing fluxes on to MOM5 (figure 1). We use Large and Yeager (2004) turbulent flux bulk formulas to calculate the air-ocean drag coefficient, evaporative transfer coefficient

and sensible heat transfer coefficient. The calculation uses the air-ocean velocity difference with an additional component to account for gustiness. Note that the implementation differs from Large and Yeager (2004) in having a $0.5\,\mathrm{m\,s^{-1}}$ floor for the 10 m relative windspeed and a ceiling of 10 for the absolute value of the stability parameter $\zeta$. We used two Monin-Obukhov iterations, which Large and Yeager (2004) state is appropriate over the ocean, but less than their suggested value of 5 over sea ice. In calculating the wind stress on the ocean we use the wind velocity relative to the ocean surface velocity, whereas we use

the absolute wind velocity to calculate wind stress on sea ice. Wind velocity in JRA55-do has been adjusted to match time-mean scatterometer and radiometer winds, which are relative to the ocean surface current; Tsujino et al. (2018) recommend adding a climatological mean surface current to JRA55-do winds to better represent absolute winds, but this suggestion has not been tested in an ocean model and so we did not take that approach here.

Ocean albedo has the constant value 0.1, which is larger than the value 0.06 used in ACCESS-OM (Bi et al., 2013b). In

CICE5 we use the same NCAR CCSM3 shortwave distribution method, ice and snow albedos in the visible and infrared bands and melt albedo parameters as in ACCESS-OM (Bi et al., 2013b, their table 2), but we use the default snow patchiness parameter 0.02 m instead of the value 0.01 m used in ACCESS-OM. Shortwave penetration into the ocean is handled by the GFDL scheme, with maximum penetration of 300 m and Manizza et al. (2005) optics using a prescribed monthly chlorophyll-a climatology as used in GFDL's CM2.5 and CM2.6 (Delworth et al., 2012; Griffies et al., 2015), based on SeaWiFS data from

1998–2006 and the method of Sweeney et al. (2005). There is no representation of geothermal heating.

We restore sea surface salinity (SSS) to the interpolated $0.25°$ World Ocean Atlas 2013 v2 monthly climatology (WOA13, also used for initial conditions; see Sect. 2.4), with a spatially constant offset to ensure that the net restoring salt flux is zero for each timestep. Restoring is applied globally (including under ice) via a salt flux, with a timescale set by the "piston velocity" (surface vertical grid spacing divided by restoring timescale) of 33 m / 300 days in all cases. The SSS mismatch is limited to

0.5 psu for the purposes of calculating the restoring, in order to avoid excessively large fluxes. These choices are typical of CORE-II models (Danabasoglu et al., 2014, table 2).

The model sea ice has a fixed bulk salinity (5 g/kg). This salt is obtained from the seawater when sea ice is formed; this can drive ocean salinity below zero in regions fresher than the ice salinity, for example during the spring melt in the shallow Siberian gulfs that are resolved in the ACCESS-OM2-01 model bathymetry. This problem in ACCESS-OM2-01 was resolved

by setting the local ocean-ice salt flux to zero in regions where the seawater salinity is less than 6 g/kg. Salinity restoring is





globally balanced; thus, in these regions the sea ice salt is instead obtained from the global surface ocean at large. Over a sea ice formation and melt cycle this produces a small unphysical transport of salt from the global surface ocean to regions where such sea ice melts.

### 2.3.1 YATM and libaccessom2

ACCESS-OM2 uses a new atmospheric driver, known as YATM, which implements a file-based atmosphere and replaces MATM which was used in ACCESS-OM (Bi et al., 2013b). Its purpose is to track model time and, when necessary, read the appropriate forcing fields from files and deliver them to the coupler. This is implemented via an associated library (libaccessom2, https://github.com/COSIMA/libaccessom2) that is linked into YATM, CICE and MOM to provide shared functionality and an interface to inter-model communication and synchronisation tasks.

YATM is also responsible for remapping river runoff in real-time. This is done separately from the other forcing fields because it is difficult to do in a distributed memory setting, since ensuring runoff is on coastal points may require interprocess communication. Remapping is done in two steps: first runoff is moved to the destination grid using conservative interpolation, and then it is distributed from land to coastal points using an efficient nearest neighbour algorithm based on a pre-computed k-dimensional tree (k-d tree) data structure (Bentley, 1975). We use the KDTREE2 Fortran k-d tree implementation (https://github.com/jmhodges/kdtree2). The k-d tree is also used to conservatively spread runoff into the neighbouring ocean grid points 15  to ensure it does not exceed a prescribed cap. ACCESS-OM2 and ACCESS-OM2-025 use a runoff cap of $0.03\,\mathrm{kg\,m^{-2}s^{-1}}$ globally. In ACCESS-OM2-01 there are reduced caps of 0.001 and $0.003\,\mathrm{kg\,m^{-2}s^{-1}}$ at the mouths of some Arctic rivers to produce broader spreading and avoid excessively low salinity, and a cap of $0.03\,\mathrm{kg\,m^{-2}s^{-1}}$ everywhere else.

### 2.4 Initial conditions

The ACCESS-OM2 and ACCESS-OM2-025 runs were started at rest, with zero sea level and with temperature and salinity from World Ocean Atlas 2013 v2 (Locarnini et al., 2013; Zweng et al., 2013) 0.25° "decav" product (the average of six decadal averages spanning 1955–2012) and run for five 60-year cycles (1 Jan 1958 – 31 Dec 2017) of JRA55-do. The initial sea ice concentration and thickness are 100% and about 2.5 m (respectively) in regions north of 70° N and south of 60° S where the sea surface temperature is less than 1°C above freezing, with a parabolic distribution of area across the five ice thickness categories. 25  The initial snow thickness in each category is 0.2 m or 20% of the ice thickness in that category, whichever is smaller.

  The ACCESS-OM2-01 experiment ran for 33 years from 1 Jan 1985 – 31 Dec 2017. It was started from a 40-year spinup under repeated 1 May 1984 – 30 April 1985 JRA55-do forcing (repeat-year forcing, RYF, Stewart et al., in prep.). This spinup began from the same initial condition as the ACCESS-OM2 and ACCESS-OM2-025 runs. The RYF spinup contained some parameter changes, in particular the ice-ocean stress turning angle was changed from 16.26° to zero at the start of August in 30  the 12th year. Before this change the Arctic ice volume built up significantly (and unrealistically) in the thickest category, but it began a slow decline when the turning angle was set to zero which persisted into the first ∼6 years of the interannually-forced run (Fig. 27c). The 1984–85 repeat-year forcing contained some biases relative to climatology; for example, biases in the North Pacific wind stress curl produced late separation of the Kuroshio Current.



## 2.5 Coupling

Figure 1 shows the fields that are transferred between the coupled model components. The prescribed atmosphere drives the global ice model, which is two-way coupled to the ocean model. Coupling is implemented using the Ocean Atmosphere Sea Ice Soil (OASIS3-MCT,  Valcke et al., 2015) coupler version 2.0, developed at CERFACS and CNRS, France (https://portal.
enes.org/oasis). The coupling strategy is based on the ACCESS-OM model (Bi and Marsland, 2010), but uses a newer library-based version of OASIS which is capable of parallel coupling. The atmosphere-to-CICE5 coupling timestep is determined by the frequency of the atmospheric forcing dataset (i.e. 3 hourly for JRA55-do). Two-way CICE5-MOM5 coupling takes place on every timestep (i.e. every ocean baroclinic timestep and ice thermodynamic timestep). In order to avoid coupled ice-ocean instabilities (Hallberg, 2014), MOM5 is configured to neglect the weight of sea ice when determining the sea surface height.
No grid remapping is needed between CICE and MOM because they use identical grids, but the atmospheric forcing requires remapping onto the CICE grid. The default remapping method used within OASIS3-MCT (SCRIP, https://github.com/SCRIP-Project/SCRIP) does not scale to 0.1° grid spacing for global models. Instead the grid remapping interpolation weights are calculated using the RegridWeightGen application which is part of the ESMF framework (https://www.earthsystemcog.org/projects/esmf/). Fluxes are remapped using conservative interpolation (second-order for ACCESS-OM2 and ACCESS-
OM2-025, first-order for ACCESS-OM2-01). Non-flux fields do not need conservative remapping, so we use patch recovery (Kritsikis et al., 2017; Khoei and Gharehbaghi, 2007) to produce very smooth destination fields; this is particularly important for the ACCESS-OM2-025 and ACCESS-OM2-01 configurations because they have finer resolution than the forcing dataset.

## 3   Computational performance

The computational performance of a coupled model depends upon the runtimes of each of its component models, the coupler
overhead, and any potential load imbalances between each component. Here we report measurements of runtime of the MOM5 and CICE5 model components as a function of resolution and core count. The computational load of ACCESS-OM2 is dominated by MOM5, which typically comprises around 90% of CPU time at the lower resolutions, and 75% of this time at 0.1° resolution. The remainder of CPU time is predominantly due to CICE5, with a negligible contribution from the coupling and the YATM file-based atmospheric model. Despite playing a smaller computational role, CICE5 can limit the overall scalability
and performance of the coupled ocean/sea-ice model because its runtime depends on sea ice covered area, which changes seasonally and causes load imbalances that are exacerbated at higher resolutions and core counts. In all cases the initial condition was taken from spun-up runs on 1st January (in 2003 in the 5th cycle at 1°, 2000 in the 5th cycle at 0.25°, and 2000 at 0.1°; see Sect. 4).

All simulations were performed on Raijin, the peak machine at Australia's National Computation Infrastructure. Runtimes
were measured on a 3592-node platform, where each node includes two Xeon Sandy Bridge (E5-2670) CPUs of speeds 2.6 GHz (base) to 3.0 GHz (turbo), with a total of 16 cores per node, and 32 GiB of external DDR3-1600 RAM. Nodes are connected by an InfiniBand FDR-14 interconnect, with peak transfer speeds of 56 Gb/s. Model data is stored on a Lustre parallel file system. On this platform, MOM5 and CICE5 computation is observed to be predominantly memory-bound, with



computational performance limited by DRAM bandwidth and on the order of 1 GFLOP/s per CPU, although the heterogeneity of the various model solvers will exhibit different degrees of performance throughout the codes. In general, performance will improve as the number of CPUs are increased, and there will be a transition from a memory-bound to a communication-bound state as the data transfer costs exceed RAM speeds. Executables were compiled with Intel compiler suite 2019 and OpenMPI

5  3.0.3.

We first present measurements of MOM5 and CICE5 runtime scalability with respect to computational core count, and consider performance over several CPU configurations. The performance of the coupled ocean/sea-ice model is inferred from the independent performance of CICE5 and MOM5, leading to recommended standard configurations.

### 3.1  MOM5 scalability

We conduct a series of tests into the scalability of MOM5 at each of the three model resolutions tested here, configured as described in Sect. 2.1. Baroclinic timesteps are 5400 s, 1800 s and 400 s at 1°, 0.25° and 0.1° (respectively). In each test we record the main loop runtime per baroclinic timestep for three short simulations at a prescribed core count, and repeat this test for different core counts. The length of each simulation in model time is set to ensure a runtime of approximately three minutes, and varies from two months at the lowest resolution to one day at the highest resolution. We report the runtime per

model timestep (Fig. 2a) and the total CPU time (Fig. 2b) for each configuration. Each point denotes the average runtime over all MPI ranks for each run. Note that the numbers provided here do not include startup time or any infrequent I/O events, typically on monthly or annual timescales, which can add considerable cost, especially at high core counts.

These tests demonstrate the highly efficient scalability of MOM5. For the standard (1° grid spacing) ACCESS-OM2 configuration, the model scales well to core counts of 400, and only begins to degrade at 800 cores. At 400 cores, one model

year takes just 10 minutes (equivalent to a theoretical maximum of 144 model years per day). While higher resolutions require considerably more computational time, MOM5 still scales outstandingly well – up to 4000 cores for ACCESS-OM2-025 and 16,000 cores for ACCESS-OM2-01 – and runtimes can often be sustained when provided with a sufficient number of CPUs. This demonstrates efficient scaling well beyond the 512 cores investigated by Schmidt (2007). On our platform, a MOM5 configuration of 0.1° grid spacing could achieve a maximum theoretical performance of almost 5 years per day, although startup

and model I/O would reduce the speed in any practical case.

These results highlight the scaling efficiency of MOM5 and give us flexibility to choose different MOM5 core counts for different configurations. However, the behaviour of the coupled ocean/sea-ice model is also dependent on CICE5 performance, as discussed below.

### 3.2  CICE5 scalability

Analogous scaling tests for CICE5 were also undertaken, configured as described in Sect. 2.2, except that the distribution scheme was sectrobin rather than roundrobin (Craig et al., 2015) at 0.25° and 0.1°, and at 1° an ice-ocean stress turning angle of 16.26° was used (instead of zero). Thermodynamic timesteps were 5400 s, 1800 s and 400 s at 1°, 0.25° and





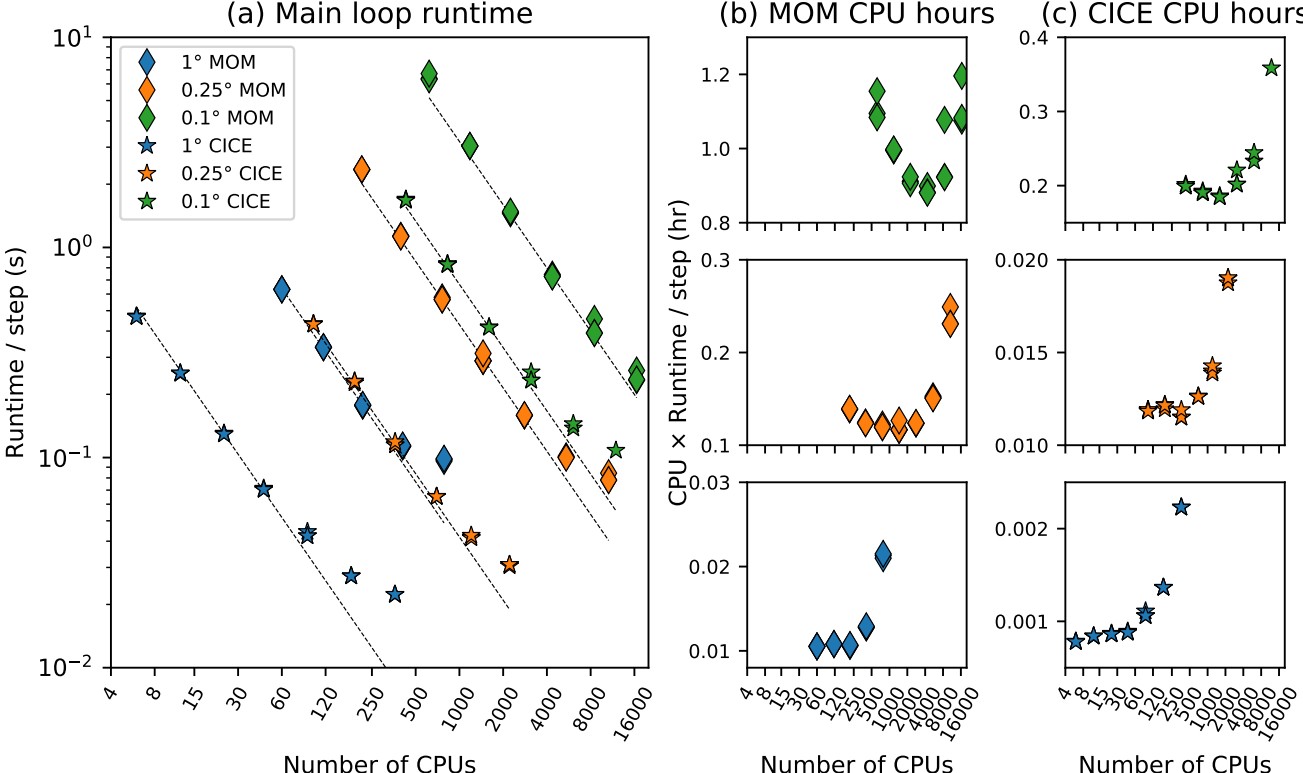

**Figure 2.** Scaling performance for MOM5 and CICE5 global model simulations showing (a) walltime, (b) CPU time per ocean baroclinic timestep, and (c) CPU time per ice thermodynamic timestep as a function of the number of processors for a short simulation at each model resolution. The dashed lines in panel (a) indicate perfect linear scaling.

0.1° (respectively). These tests show that at 1° grid spacing the model scales up to ∼50 cores, and up to 500 cores at 0.25° and 2000 cores at 0.1° (Fig. 2a, c). Even at higher than optimal core counts, scaling is acceptable.

However, these numbers obscure some more complex issues. First, the effective performance of CICE5 is variable, partly because ice cover is seasonal, so different tiles have a different amount of work to do at different times of year. This variability is

5 mitigated by using the `sectrobin` scheme in CICE5 (Craig et al., 2015) in these scaling tests at 0.25° and 0.1°, which selects tiles from different locations within a sector to be computed on the same core; we use four tiles per core for this scheme at 0.1° grid spacing. While the code itself scales well with the number of cores, the total amount of work required increases more rapidly in CICE5 at higher resolutions. That is, the additional CPU time required for each MOM5 model timestep increases by a factor of 90 in going from 1° to 0.1° grid spacing (Fig. 2b); for CICE5 the CPU time required per timestep increases by a factor

10 of 200, with a disproportionately large increase between 0.25° and 0.1° (Fig. 2c). We mainly attribute this to changing from one to three dynamic timesteps per thermodynamic timestep between 0.25° and 0.1°, exacerbated by using the slower mushy thermodynamics, but residual load imbalances may also contribute. Consequently, we use relatively more cores for CICE5 than



**Table 2.** Outline of model grid, size, cores and typical performance. Model timestep is given by the ocean baroclinic timestep, which equals the ice thermodynamic timestep (but is three times longer than the ice dynamic timestep at 0.1°). Configuration improvements subsequent to the runs shown here have halved the ACCESS-OM2-01 walltime and CPU-hour cost.

| **Model** | Lateral Spacing | Model Domain | Ocean Timestep (s) | Ocean Cores | Ice Cores | Walltime (hours/year) | CPU (hours/year) | Memory (Gb) |
|---|---|---|---|---|---|---|---|---|
| ACCESS-OM2 | 1° | $360 \times 300 \times 50$ | 5400 | 216 | 24 | 0.38 | 118 | 83 |
| ACCESS-OM2-025 | 0.25° | $1440 \times 1080 \times 50$ | 1350–1800 | 1455 | 361 | 2.6 | 4,700 | 522 |
| ACCESS-OM2-01 | 0.1° | $3600 \times 2700 \times 75$ | 450 | 4538 | 1600 | 19.9 | 118,000 | 2689 |

for MOM5 at higher resolution – the ratio of ice to ocean cores is 0.11 in ACCESS-OM2, 0.25 in ACCESS-OM2-025 and 0.35 in ACCESS-OM2-01 (Table 2).

### 3.3 Coupled ocean/sea-ice model configuration

The performance of the coupled ocean-sea ice model is limited by the scaling performance and resource requirements of
both components, as well as their load balancing and variability. This load balance is further complicated by the differing performance of components as a function of resolution, and by the need to alter the model timestep for some simulations. It is clear from Fig. 2a that balancing the runtime of the model components requires more cores for MOM5 than CICE5, but the ratio depends on resolution. Since CICE5 has a lower core count than MOM5 at all resolutions, if imbalances are to exist, we aim to ensure that CICE5 is waiting for MOM5. The standard configurations, core counts and typical performance of differing
model resolutions are shown in Table 2.

The configurations shown here are under continuous development and optimisation, and it is anticipated that improvements in model stability and load balancing will continue to improve performance in the future (for example, we have recently doubled ACCESS-OM2-01 performance relative to the figures in Table 2). We have also configured a minimal ACCESS-OM2-01 configuration with a total core count of ∼2000, to aid in testing and to run on smaller systems. These configurations
will be continually released and documented on the ACCESS-OM2 code repository as they are developed.

### 4 Model evaluation

We now outline results from ACCESS-OM2 using simulations with each of the three horizontal resolutions outlined in Table 2. Each of the three simulations is forced by the interannual JRA55-do forcing dataset (Tsujino et al., 2018), which currently covers 60 years from 1958 until the end of 2017. The lower resolution simulations (both ACCESS-OM2 and ACCESS-OM2-
025) continuously cycle through five iterations of this dataset, giving a 300-year simulation, following the CORE-II protocol (e.g. Danabasoglu et al., 2014) and the CMIP6/OMIP protocol (Griffies et al., 2016). Selected global diagnostics from these simulations are shown by the blue and orange lines (respectively) in Fig. 3, where dates have been aligned so that the last





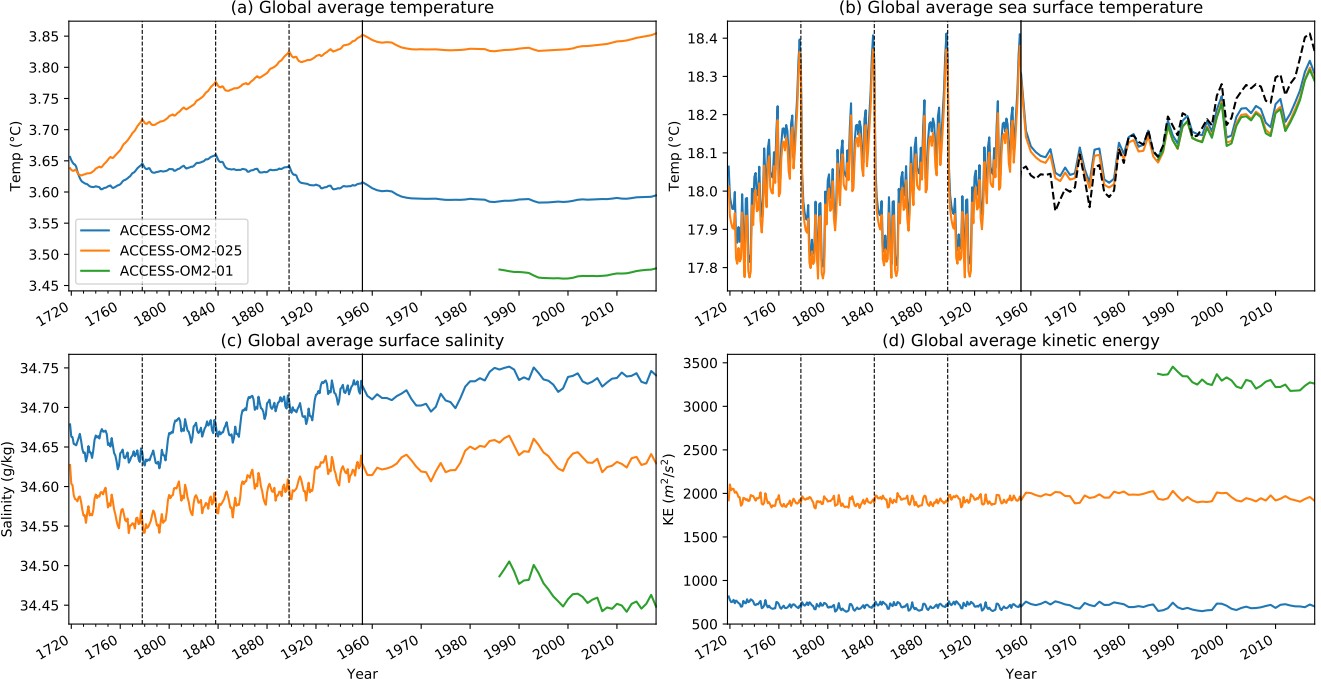

**Figure 3.** Timeseries of the global average of annual mean (a) ocean temperature; (b) sea surface temperature; (c) sea surface salinity and (d) ocean kinetic energy for each of the simulations. Output is shown for the full interannually forced model simulations, including 5 interannual forcing cycles for ACCESS-OM2 and ACCESS-OM2-025, with the timescale compressed for the first 4 cycles and the end of each forcing cycle indicated by a vertical line. The dashed black curve in (b) is the observed sea surface temperature anomaly (ERSST v4; Huang et al., 2019), offset by 18°C to compare the rate of warming over the final cycle of forcing. Model time has been offset to ensure that the final cycle has a date that is consistent with the forcing date, allowing the short, 33 year, ACCESS-OM2-01 simulation to be plotted on the same time axis.

cycle of forcing matches the calendar dates of the forcing dataset (giving a nominal start year of 1718). The main period of model evaluation will be the final interannual forcing cycle, years 1958–2017 inclusive. When time-averaged fields are shown, averages are taken over the last 25 years of simulation (years 1993-2017 inclusive) unless stated otherwise.

The highest resolution simulation, using ACCESS-OM2-01, is ∼ 1000 times more computationally intensive than ACCESS-
5   OM2, and ∼ 25 times more computationally intensive than ACCESS-OM2-025 (see Table 2). A full simulation with five internannual forcing cycles of ACCESS-OM2-01 is not possible with current computing resources, and hence we use an alternative spinup strategy. As discussed in Sect. 2.4, in this case we select a Repeat Year Forcing (RYF) spinup strategy (Stewart et al., in prep.), in which the time period 1 May 1984 – 30 April 1985 is repeated continuously. This spinup has been run for 40 years, after which time we conduct an interannually forced simulation from 1985 through to 2017. It is this
10  interannual simulation period which is used for the model evaluation in this manuscript, as indicated by the green line in Fig. 3. It is important to note that, in keeping with the CORE-II and OMIP protocols, we make no attempt to account for model drift



in the analysis of these simulations; in the case of ACCESS-OM2-01 this means that the simulation is less well equilibrated than the lower resolution simulations, and contains some biases from the repeat year spinup strategy. Care must be taken in interpreting the influence of model drift on the different cases.

The timeseries of globally averaged quantities shown in Fig. 3 indicate that there is an initial drift due to heat uptake in
ACCESS-OM2-025, while ACCESS-OM2-01 drifts cold relative to the initial state (Fig. 3a), primarily during the repeat year spinup phase. On the other hand, the global average Sea Surface Temperature (SST) in the models is dominated by the forcing field, with only weak variations between each cycle (Fig. 3b). This variation of SST within the final forcing cycle is closer to observations than that seen with CORE-II forced models (see Fig. 2 of Griffies et al., 2014), and includes a reasonable representation of the slowdown in warming in the decade preceding 2010. The high resolution model also drifts towards
surface freshening, which is partly offset by the surface salinity restoring that is incorporated into the model (Fig. 3c), but this drift predominantly occurs during the Repeat Year Forcing spinup, and the rate of drift is reduced when the interannual forcing is used. As expected, the kinetic energy of the simulations is a strong function of resolution, with higher resolution models containing more turbulent processes (Fig. 3d). Each of these aspects of the simulations is investigated in greater depth by the following analysis, where we first focus on global circulation metrics, then look to better characterise important regional ocean
circulation differences and finally investigate the representation of sea ice.

## 4.1   Global circulation

### 4.1.1   Horizontal circulation

One of the most commonly used integrated metrics for ocean model circulation is the transport of the Antarctic Circumpolar Current (ACC) through Drake Passage. Fig. 4 shows the Drake Passage transport for each of the three simulations being
compared. There is a clear distinction between the ACC transport in the lower resolution simulations and in ACCESS-OM2-01. In the former case, the larger ACC transport is closer to the observational estimate of 173 Sv (black dashed line; Donohue et al., 2016), with significant variability over the course of the interannual forcing cycle. Drake Passage transport in ACCESS-OM2-01 is significantly lower, and is more stable. The underestimated ACC transport in these models is characteristic of this class of ocean-sea ice models (e.g. Farneti et al., 2015). It is notable that the higher resolution case does not lead to an improved
transport prediction, although this could be a result of the much shorter spinup at 0.1°.

To evaluate the capacity of the different model configurations to represent the mean state of the broad-scale horizontal ocean circulation, the simulated mean dynamic sea level (MDSL), averaged over years 1993–2012, is compared with the CNES-CLS13 observational product distributed by AVISO, which combines data from satellites, surface drifters and in-situ measurements to reconstruct a time mean, global Mean Dynamic Topography (MDT) field for the same time period. The
reconstruction procedure is described in Rio et al. (2009). The comparisons between the simulated MDSL and observed MDT (Fig. 5) indicate broad-scale agreement in this metric. There is a noticeable improvement in western boundary current structure in the higher resolution models, and a shallower MDSL minimum in the ACCESS-OM2-01 case, consistent with the reduced Drake Passage transport shown in Fig. 4.





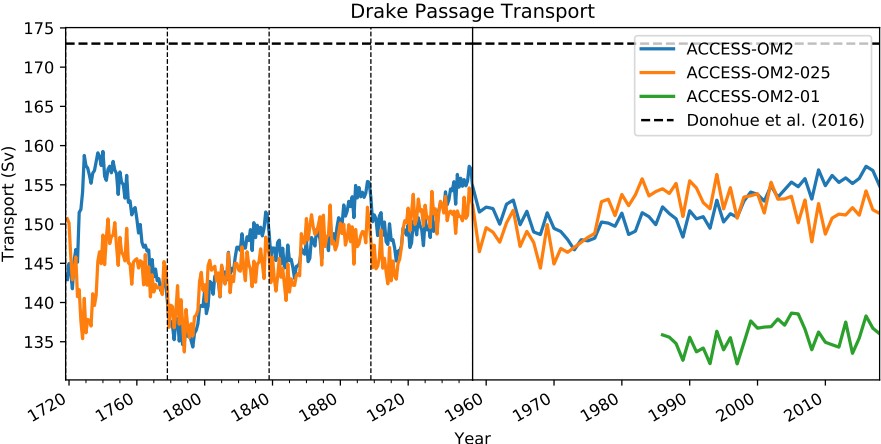

**Figure 4.** Annual mean transport through Drake Passage as a function of time for each of the 3 model simulations. The horizontal black dashed line shows the observational estimate of Donohue et al. (2016). The vertical solid line divides the first 4 cycles (on the left) from an expanded view of the final cycle (on the right), with vertical dashed lines marking the end of each forcing cycle.

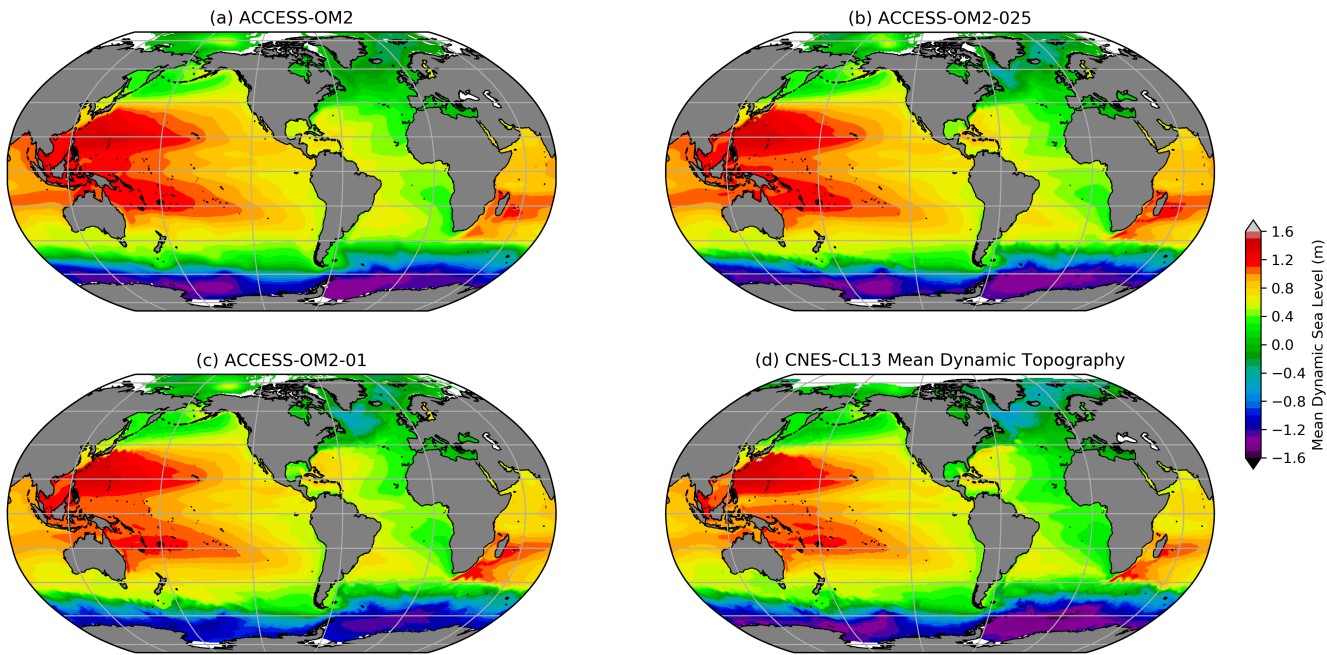

**Figure 5.** 1993–2012 Mean Dynamic Sea Level in (a) ACCESS-OM2; (b) ACCESS-OM2-025 and (c) ACCESS-OM2-01. (d) Observational reconstruction of 1993–2012 Mean Dynamic Topography from the CNES-CLS13 product. The model outputs have had a 0.5 m offset added for clarity.



The capacity of the different model configurations to represent the spatial patterns of sea-level variability provides a reliable proxy for surface mesoscale activity. To do this, we compute the standard deviation of the sea-level anomaly (SLA) for each configuration at each model grid point, to produce global maps of SLA deviation for the years 1993–2014 (where long-term linear trends are removed). The simulated SLA standard deviation is then compared with the objectively interpolated, multi-

mission satellite derived, SLA product (AVISO Ssalto/Duacs), also from 1993–2014. We use the gridded observational product for convenience in comparing global maps of SLA variability. However, we note that the optimal interpolation procedure used to produce the gridded product from satellite ground tracks tends to smooth the underlying fields and hence may underestimate the SLA variance by as much as 50% in certain regions (Chambers, 2018). As such, the true SLA variability may be higher than is indicated here.

The maps of SLA standard deviation are plotted in Fig. 6. Elevated SLA variability typically occurs in regions rich in energetic mesoscale eddies. For example, the altimetric product (Fig. 6(d)) shows elevated SLA variability in western boundary currents and their jet extensions, as well as in the Southern Ocean, which are regions where mesoscale dynamics are most active. Both the ACCESS-OM2-01 and ACCESS-OM2-025 simulations appear to capture this spatial pattern of SLA variability well, with both boundary currents and the Southern Ocean playing host to regions of enhanced SLA standard deviation. However,

the ACCESS-OM2-025 configuration is unable to capture the magnitude of the observed variability, with values a factor of two or more below those of the observational product (which is itself an underestimate). The SLA variability magnitude in ACCESS-OM2-01 is closer to the observational estimate but still somewhat low; the highest values are found south of the African continent in the Agulhas retroflection region, and the Gulf Stream and Kuroshio extension. The coarse-resolution ACCESS-OM2 configuration is unable to represent any significant SLA variability.

Fig. 6 also shows broad regions of sea level variability at lower latitudes, typically associated with slower modes of climate variability and with less amplitude than the western boundary currents. In the Pacific Ocean, all resolutions simulate variability associated with ENSO, though all underestimate the observed variability by 10–20 %. ENSO cycles drive variability in the Eastern Equatorial Pacific, and the Western Pacific, east of the Phillipines and Papua New Guinea (Han et al., 2017; Mu et al., 2018). In the Indian ocean, variability is associated with both the Indian Ocean Dipole and ENSO (Li and Han, 2015).

Anomalies in the south-central Indian Ocean are driven by wind anomalies related to ENSO events (Xie et al., 2002) and are simulated in each model resolution. ACCESS-OM2-01 also simulates variability from Indonesia down the West Australian coast which then propagates as Rossby waves westward (Potemra and Lukas, 1999), which is muted in the coarser resolution simulations.

### 4.1.2 Overturning circulation

Fig. 7 shows the overturning circulation, computed on potential density surfaces (referenced to 2000 dbar), averaged over the last 25 years of simulation (1993-2017) for each of the three cases. In this figure the positive cell, which has a maximum near potential density 1036.5 kg m$^{-3}$, is the interhemispheric upper overturning cell which is dominated by the Atlantic component. This Atlantic Meridional Overturning Circulation (AMOC) involves sinking of dense water in the North Atlantic, which re-emerges at the surface in the Southern Ocean (Marshall and Speer, 2012). There are clear differences between the





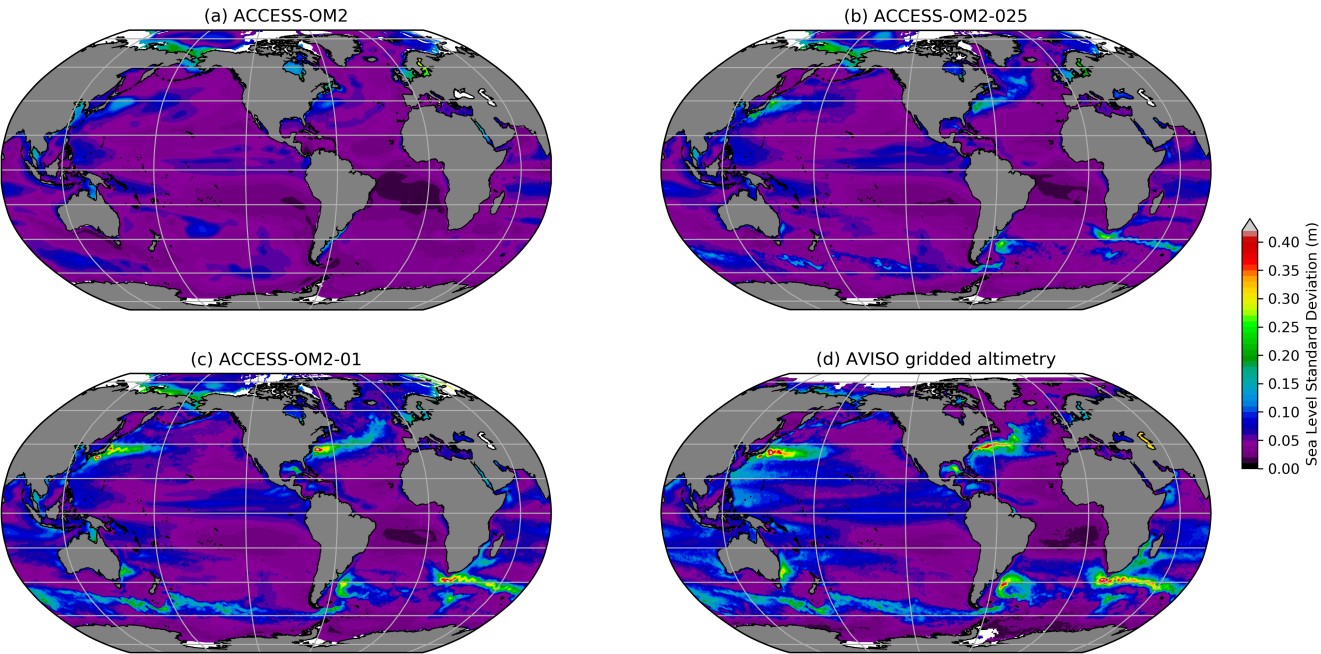

**Figure 6.** Standard deviation of Sea Level Anomaly for the years 1993–2014. (a) ACCESS-OM2; (b) ACCESS-OM2-025; (c) ACCESS-OM2-01 and (d) AVISO Ssalto/Duacs gridded analysis of satellite altimetry.

three resolutions in their ability to represent the AMOC cell, which has an estimated transport from 2004 to 2012 of 17.2 Sv at 26° N (McCarthy et al., 2015). In ACCESS-OM2, the AMOC at 26° N is weak, with a maximum value of around 10 Sv, but it retains a strong interhemispheric character. In ACCESS-OM2-025 and ACCESS-OM2-01 the AMOC is considerably stronger, with the maximum value of the circulation only weakly decaying with latitude. A timeseries of the AMOC, measured

at 26° N and constrained to the Atlantic basin only, clearly shows these differences (Fig. 8a), but also shows a decline over the duration of the ACCESS-OM2-01 case. The comparison of the AMOC in ACCESS-OM2-025 and ACCESS-OM2-01 with the years of McCarthy's estimate is favourable, although we cannot determine from the ACCESS-OM2-01 simulation whether this agreement will persist in future cycles. (Note that the first cycle of ACCESS-OM2-025 has a lower AMOC of ∼14 Sv; data not shown). Longer simulations are required before we can firmly establish the equilibrium behaviour of the AMOC for

ACCESS-OM2-01.

The other circulation cell of interest in Fig. 7 is the abyssal overturning cell (the negative cell centred at 1037 kg/m$^{-3}$), which occupies a small part of density space but comprises a significant fraction of global water volume. Here, the strongest modelled overturning cell is in ACCESS-OM2-01 (12–15 Sv at 40° S, see Fig. 8b; compared with poorly constrained observational estimates of 20–50 Sv; Sloyan and Rintoul, 2001; Lumpkin and Speer, 2007; Talley, 2013). We argue that in ACCESS-OM2-01,

this abyssal cell is partly driven by the more realistic process of surface water mass transformation on the Antarctic continental shelf (data not shown). On the other hand, the ACCESS-OM2-025 and ACCESS-OM2 abyssal cells are more dependent on



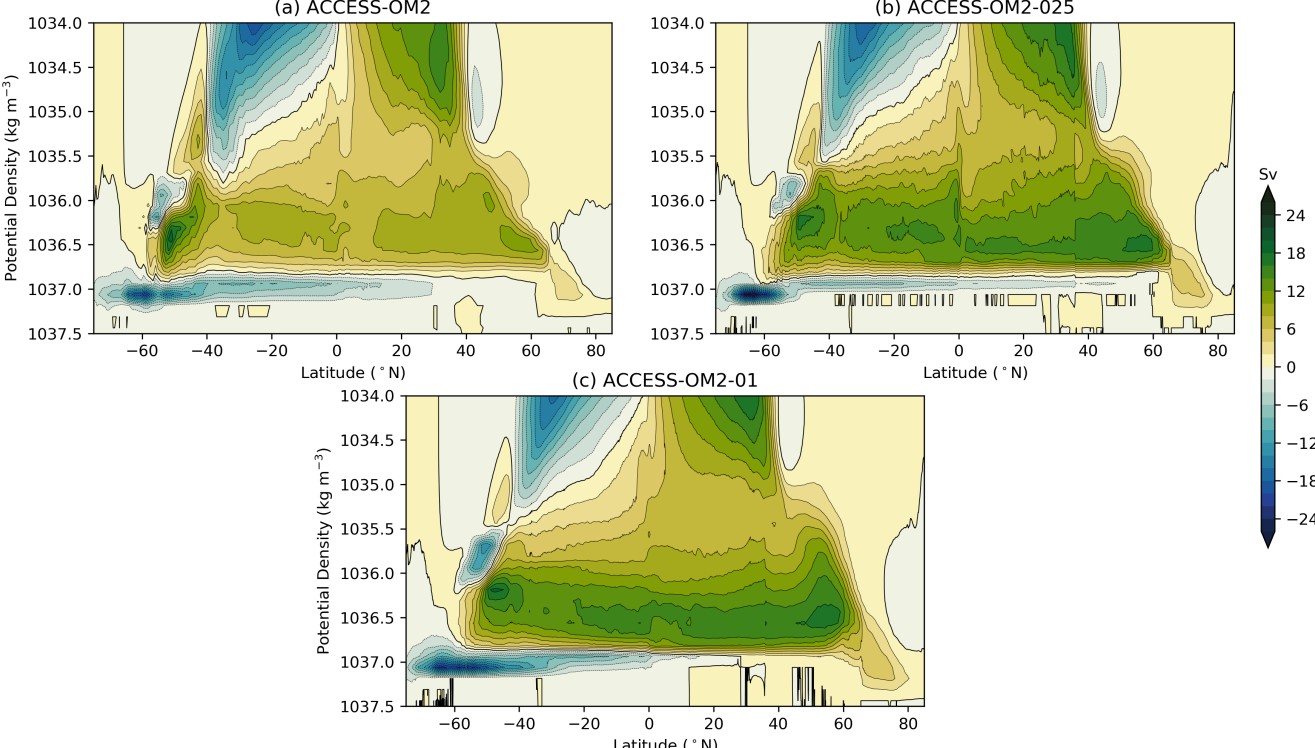

**Figure 7.** Time-mean zonally-integrated overturning circulation computed on potential density surfaces (referenced to 2000 dbar) for (a) ACCESS-OM2; (b) ACCESS-OM2-025 and (c) ACCESS-OM2-01 simulations. Overturning is computed on density surfaces, integrated zonally around the globe and averaged over the last 25 years of simulation (1993–2017). The contour interval is 2 Sv.

water mass transformation due to open ocean convection, but this dense water is poorly connected with the rest of the global ocean, leading to a weaker (∼6–10 Sv) overturning transport at 40° S. These biases in water mass transformation are common in coarse resolution ocean and climate models (e.g. Heuzé et al., 2015), indicating the potential of ACCESS-OM2-01 to be used for studies into the abyssal cell sensitivity to changes in climate. Overturning biases are further discussed in Sect. 4.1.4.

5 ### 4.1.3 Ocean heat uptake

The evolution of global mean ocean temperature (Fig. 3a) is shown in more detail as horizontally-averaged temperature anomalies in Fig. 9. Behaviour in the upper ocean (< 100 m), the mid-depths (100–1500 m) and the abyssal ocean (>1500 m) is distinct. In ACCESS-OM2 and ACCESS-OM2-025, warming at mid-depth and cooling in the abyssal ocean are consistent with Bi et al. (2013b). These opposing trends largely cancel in ACCESS-OM2, but produce a net warming in ACCESS-OM2-025

10 which was established in the four cycles prior to the one shown (Fig. 3a). For these experiments, the evolution of temperature anomalies is similar for the full 300-year simulations (not shown), from which we infer that model drift dominates the temperature bias. The temperature drifts at mid- and abyssal depths in ACCESS-OM2-01 have opposite sign to the coarser models,





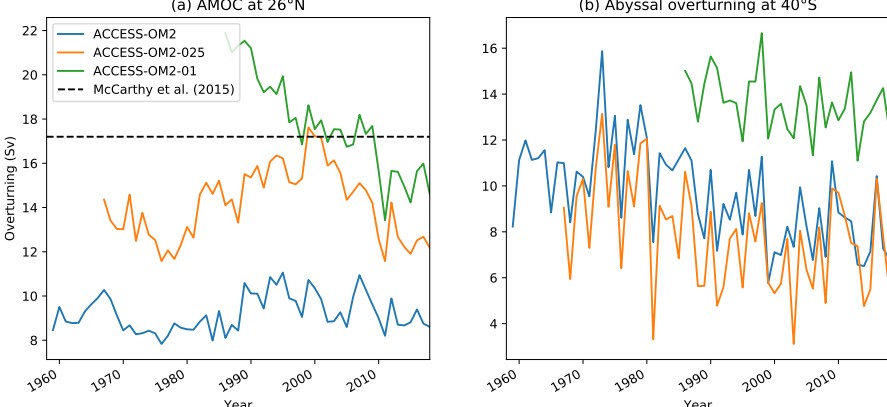

**Figure 8.** (a) Annual mean upper overturning cell (AMOC) magnitude as a function of time, defined as the maximum value of the global overturning streamfunction computed on density surfaces, measured at $26°$ N, integrated between $103°$ W and $5°$ W and for potential density classes that exceed $1035.5 \, \mathrm{kg \, m^{-3}}$. Observational estimate from McCarthy et al. (2015) for 2004-2012 is shown by the dashed black line. (b) Annual mean abyssal cell overturning magnitude as a function of time, defined as the minimum value of the global overturning streamfunction computed on density surfaces, measured at $40°$ S, integrated zonally around the globe and for potential density classes exceeding $1036 \, \mathrm{kg \, m^{-3}}$; its sign is changed to yield a positive value.

with cooling at mid-depth (centred at 300 m) and weak warming in the deep ocean. The spatial structure of these drifts is shown in Figs. 10 and 12(a, c, e), and discussed further in Sect. 4.1.4.

Differences in temperature drift between different resolutions of the same ocean model component are usually due to differing resolved and parameterised processes. Previous studies with the GFDL-MOM CM2 suite (which shares a similar ocean model component with ACCESS-OM2) showed a consistent pattern of model drift across model resolutions (with the $0.25°$

warming the fastest), although in the coupled system all models warmed at a more rapid rate (Griffies et al., 2015). In particular, the warming tendency at mid-depths is larger when the mesoscale eddy processes (namely eddy-advection and isoneutral diffusion) are not well resolved. This occurs especially in eddy-permitting models (such as GFDL-MOM CM2.5, 0.25 degree) which neither fully resolved eddy processes nor include any eddy parameterisation. Our 0.25 degree model (ACCESS-OM2-

025) includes both eddy-advection (Gent and McWilliams, 1990) and neutral diffusion (Redi, 1982; Griffies et al., 1998) parameterisation with weak coefficients which act to limit mid-depth warming (not shown), although it still has larger mid-depth drift than ACCESS-OM2.

Differences in mesoscale eddy processes can account for quantitative differences in model drift, but the distinct temperature drift in ACCESS-OM2-01 is intriguing. Additional experiments with ACCESS-OM2 and ACCESS-OM2-025 following the

15 same spin-up strategy used in ACCESS-OM2-01 (i.e. forced with JRA55-do Repeat Year Forcing from 1984-1985) reveal a temperature drift generally colder than the equivalent interannually forced experiments, especially in the upper 1000 m (not shown). These experiments suggest that the different spin-up approach in the ACCESS-OM2-01 simulation partially drives the mid-depth cooling observed at that resolution; differences in resolved/parameterised processes might play a secondary role.





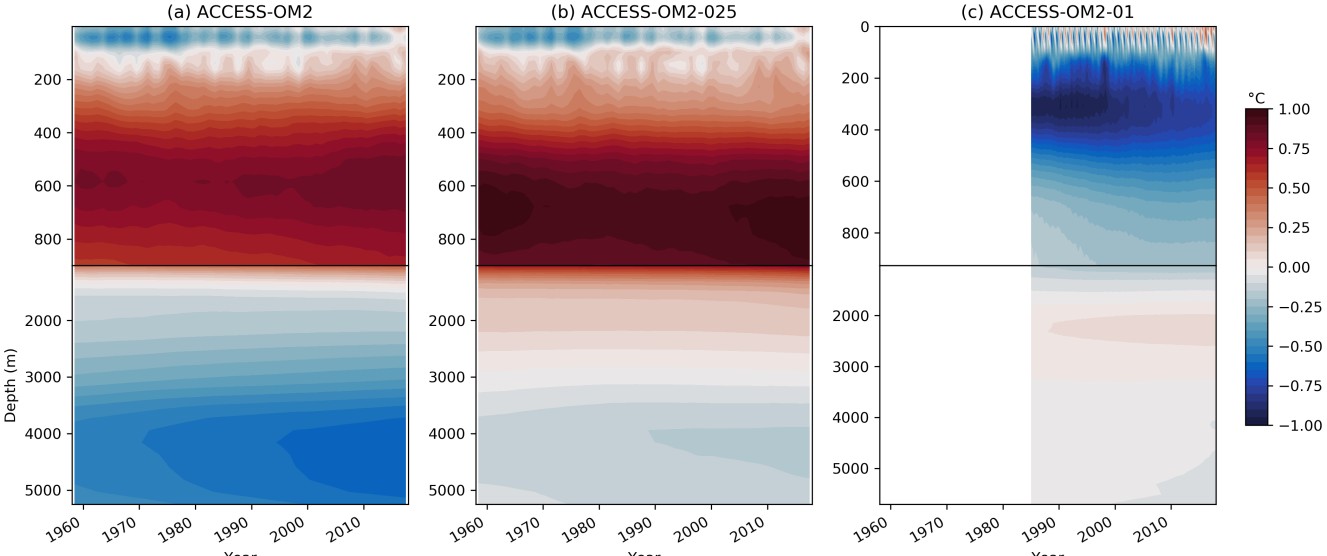

**Figure 9.** Horizontally averaged temperature anomaly (°C) relative to WOA13, as a function of depth and time over the last interannual forcing cycle for (a) ACCESS-OM2; (b) ACCESS-OM2-025 and (c) ACCESS-OM2-01. (a), (b) are annual means, and (c) are monthly means.

Model drift dominates the long-term evolution of temperature anomalies, but considerable interannual variability occurs in the upper 100 m (Fig. 9). Warming trends toward the end of the historical period are superimposed upon the cold anomalies near the surface in all ACCESS-OM2 models. Atmospheric forcing such as Coordinated Ocean-Sea ice Reference Experiment (CORE) Interannual Forcing (IAF) and JRA55-do are not designed to reproduce a long-term trend as expected from climate

change due to the adjustment performed to obtain a global surface heat budget closure over the satellite era (Tsujino et al., 2018). However, an intermodel comparison study under the CORE-IAF protocol showed that all models experienced an increase in ocean heat content in the upper 700 m and associated sea-level rise over the 1993-2007 period similar to observations (Griffies et al., 2014). A practical approach to isolate the interannual variability from the model drift in ocean-sea ice model studies is to perform a de-drift using a control run, in a similar way as performed in fully-coupled climate models (Sen Gupta

et al., 2013; Hobbs et al., 2016). Whilst the protocol of CORE-IAF/JRA55-do does not require a control run, it can be achieved using a Normal Year Forcing (CORE-NYF) or a Repeated Year Forcing (JRA55-do).

### 4.1.4 Temperature and salinity biases

Model drift can occur for a variety reasons in an ocean-sea ice model, particularly due to deficiencies in model physics and numerics, or due to unresolved processes in the model coupling (Sen Gupta et al., 2013). This drift can be further investigated

by examining model sea surface temperature (SST) biases (relative to WOA13 climatology) as presented in Fig. 10; the corresponding surface salinity (SSS) biases are shown in Fig. 11. Large warm biases associated with western boundary currents





(WBCs) are found in all ACCESS-OM2 resolutions; this is also seen in many CORE models, associated with non-eddy permitting resolution and poor representation of WBC separation and fronts (Griffies et al., 2009). These biases are reduced in ACCESS-OM2-025 and particularly in ACCESS-OM2-01, however the similarities in the spatial pattern of surface temperature suggest the possibility of systematic biases in the surface forcing, or in the surface coupling.

In the Southern Hemisphere, larger biases in highly energetic regions (e.g. the Agulhas retroflection and along the ACC path) in ACCESS-OM2 appear to be due to an unrealistic representation of fronts, showing a significant improvement in the high-resolution experiments (ACCESS-OM2-025 and ACCESS-OM2-01). The biases in the Brazil-Malvinas confluence are much smaller at high resolution, probably due to a better representation of the eddy-driven Zapiola Anticyclone (see Sect. 4.2.4 and Fig. 24).

The biases in the Subpolar North Atlantic show significant differences across model resolution, likely due to details of the representation of the AMOC transport (see Sect. 4.1.2). The configurations with a strong AMOC (ACCESS-OM2-025 and ACCESS-OM2-01; see Figs. 7 and 8a) show negative temperature biases in the North Atlantic Current (NAC), which is (at least partially) density compensated by negative salinity biases (Fig. 11). Similar biases have been previously associated with a too-zonal path of the NAC (Danabasoglu et al., 2014, and Sect. 4.2.4) and deficient overflow from the Nordic Seas (Zhang et al., 2011). ACCESS-OM2 has generally cold anomalies in the Subpolar North Atlantic but comparatively smaller biases in the NAC; the weak AMOC transport is likely related to strong fresh biases around Greenland and in the Labrador Sea (Fig. 11).

ACCESS-OM2 presents a smaller warm bias near upwelling zones on the west coast of the American and African continents in comparison with CORE models (Griffies et al., 2009). This bias has been associated with coarse resolution in both model and wind stress forcing (Bi et al., 2013b) and thus may benefit from the higher horizontal resolution of the JRA55-do forcing in
comparison with the CORE forcing (Taboada et al., 2019). However, this bias is larger in the high-resolution models (ACCESS-OM2-025 and ACCESS-OM2-01), possibly related to lower explicit vertical diffusivity in these models which leads to a stronger tropical thermocline (see Sect. 4.2.3).

The global zonal-mean anomalies of temperature and salinity relative to WOA13 are presented in Fig. 12. The distribution of heat and salt in the latitude and depth plane are controlled by the global thermohaline and wind-driven circulation. The
difference between the model 1993–2017 mean in the last cycle and the observed climatology (WOA13) reveals geographical patterns of the model drift. In the Southern Ocean, the signature of Antarctic Bottom Water (AABW) shows a cold bias in ACCESS-OM2 and ACCESS-OM2-025 that spreads into the abyssal ocean. This bias is likely associated with large areas of anomalous deep (often full-depth) convection that appear every winter and spring in the eastern Weddell Sea and western Ross Sea in the ACCESS-OM2 and ACCESS-OM2-025 simulations. The behaviour of the two coarser models is typical of CMIP5
models, which produce bottom water by spurious deep-ocean convection rather than down-slope flows (Heuzé et al., 2013). In some models this convection is associated with spurious open-ocean polynyas (Heuzé et al., 2015); however, as in the GFDL CM2.5 model (Dufour et al., 2017), persistent open-ocean polynyas do not form in the ACCESS-OM2 simulations. The deep cold bias is much reduced in ACCESS-OM2-01, which has a more realistic AABW formation in the Antarctic continental shelf, with anomalous open-ocean convection confined to a much smaller and more interannually-variable region in the northeastern
Weddell Sea (but has also had less time to drift away from climatology).

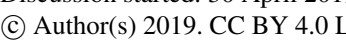
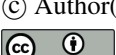


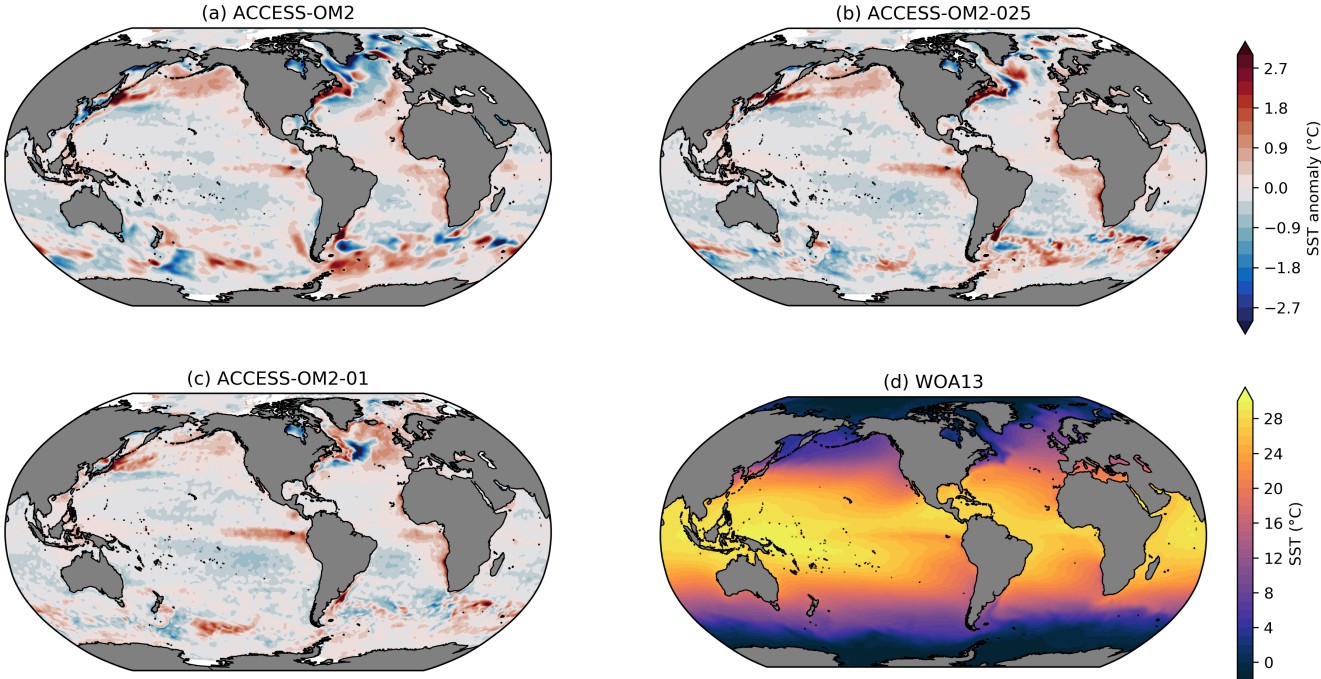

**Figure 10.** Global 1993–2017 mean sea surface temperature (SST) bias relative to WOA13 for (a) ACCESS-OM2; (b) ACCESS-OM2-025 and (c) ACCESS-OM2-01. The WOA13 temperature field is shown in (d).

The warm and salty biases north of 45°S above 1500 m are associated with weak penetration of cold and fresh Antarctic Intermediate Water, which can be caused by incorrect subduction and/or isopycnal mixing rates (Bi et al., 2013b). These biases are larger in ACCESS-OM2 and ACCESS-OM2-025, and smaller in ACCESS-OM2-01; we hypothesise that the coarser models have less isopycnal mixing (resulting from the sum of partially resolved and partially parameterised mixing) while ACCESS-OM2-01 seems to have a more realistic explicitly-resolved isopycnal mixing. Weaker along-isopycnal transport may also drive positive temperature and salinity biases in the northern hemisphere at similar latitudes, resulting in a wide band of positive biases between 45°S–45°N above 1500 m as a result of the isopycnal spreading of mode and intermediate water masses. These biases are significantly reduced in ACCESS-OM2-01, although it shows a considerable negative temperature and salinity bias at subsurface low latitudes (also seen in Figs. 19, 20 and 23), possibly due to excessive upwelling of colder and fresher water from the ocean interior and/or insufficient mixing-driven downward heat transport because of the lack of vertical background diffusivity and reduced numerical diffusion in this configuration (Sect. 2.1.4).

The biases in high northern latitudes are linked to poor Gulf Stream behaviour (Sect. 4.2.4) and the Atlantic Meridional Overturning Circulation. In ACCESS-OM2-025 and ACCESS-OM2-01, where the AMOC transport is stronger (Figs. 7 and 8a), the zonal-mean bias shows warm anomalies between 1000-3000 m at 60°N (Fig. 12c, e). On the other hand, the weak AMOC transport in ACCESS-OM2 is translated into a strong warm bias above 1000 m, just below a large fresh bias (Fig. 12a, b).



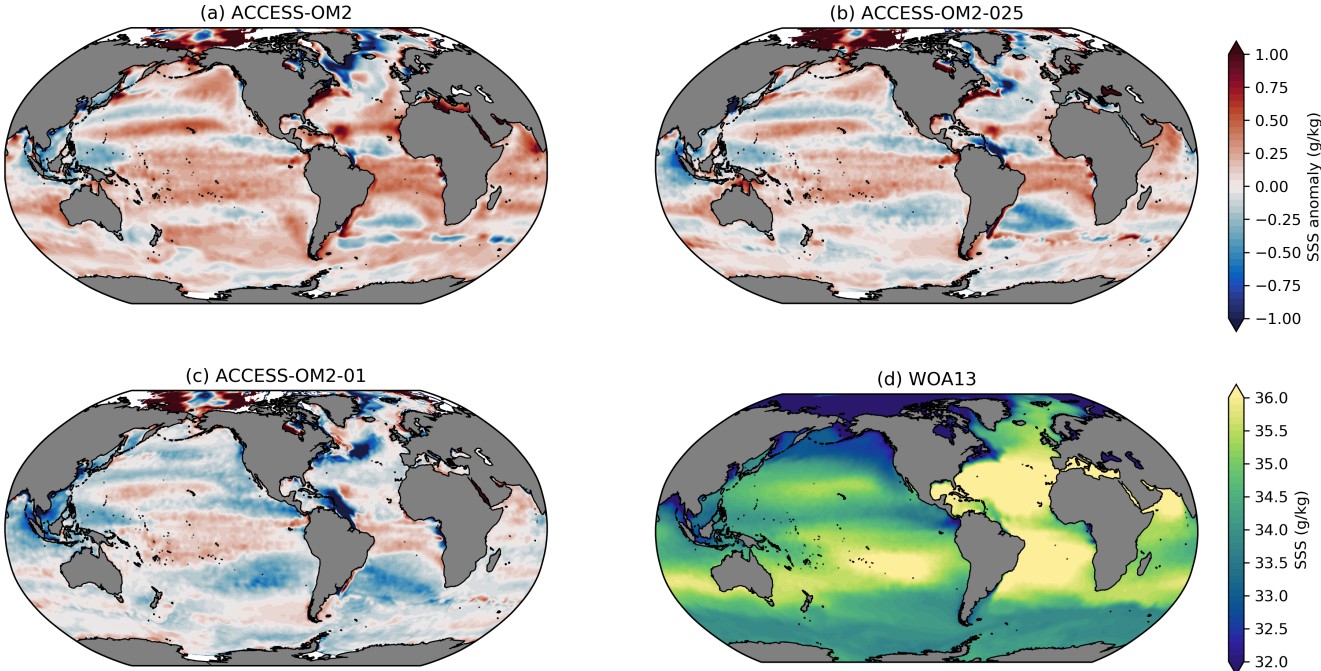

**Figure 11.** Global 1993–2017 mean sea surface salinity (SSS) bias relative to WOA13 for (a) ACCESS-OM2; (b) ACCESS-OM2-025 and (c) ACCESS-OM2-01. The WOA13 salinity field is shown in (d).

Warm biases in this region have been linked with excessive surface deep convective mixing and overturning (Griffies et al., 2009; Bi et al., 2013b).

### 4.1.5 Heat transport

All three model configurations reproduce the large-scale features of the meridional heat transport suggested by reanalysis products (Fig. 13). ACCESS-OM2 simulates a weaker northward heat transport than the other two configurations at most latitudes, associated with a weak AMOC (Figs. 7, 8). Heat transport within the Southern Ocean is consistent with observations within the spread between the observational products. Heat transport north of $40°$ N within ACCESS-OM2-025 and ACCESS-OM2-01 is stronger than suggested by the reanalysis products, but consistent within error bars with the more direct estimate from WOCE (Ganachaud and Wunsch, 2003). The local maximum in heat transport at $\sim 50°$ N is commonly seen in higher resolution models, and is thought to reflect a stronger Atlantic subpolar gyre contribution to the circulation (Griffies et al., 2015).

In the tropics, the models simulate consistently weak poleward heat transport in comparison to the reanalysis products in both hemispheres. This weak transport is a feature of many ocean-only and coupled climate models (e.g. Griffies et al., 2009, 2015). The ACCESS-OM2 configurations do not lie outside the range of model simulated transports in this regard.





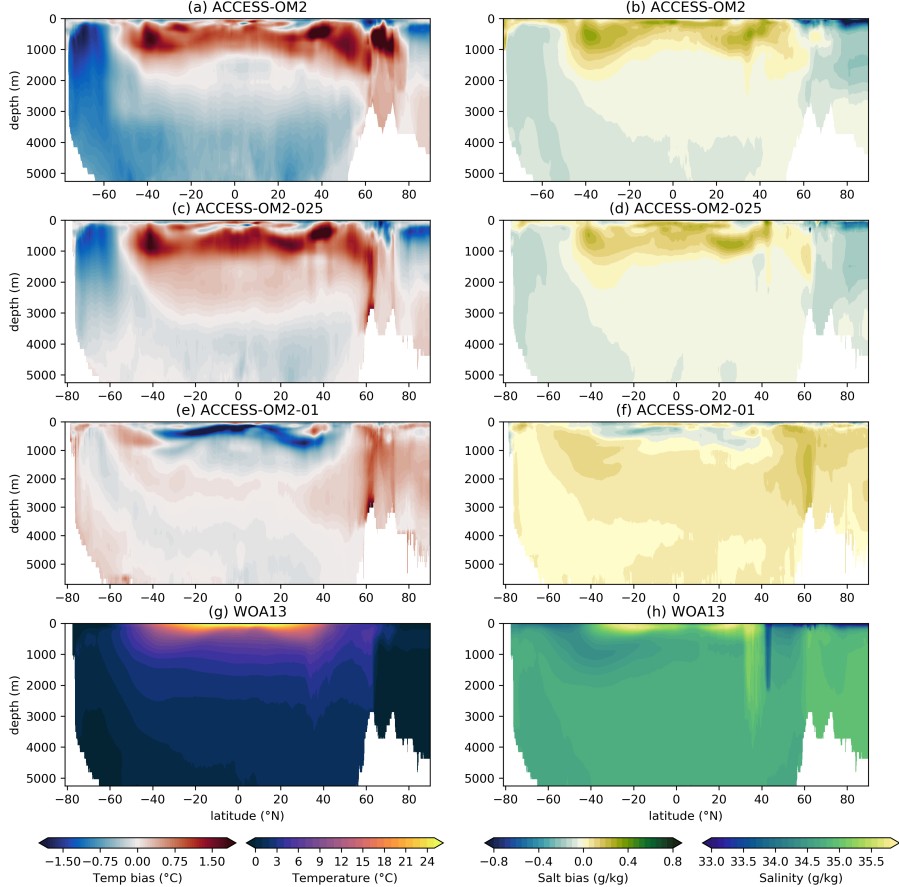

**Figure 12.** Zonally averaged temperature bias relative to WOA13 for (a) ACCESS-OM2; (c) ACCESS-OM2-025 and (e) ACCESS-OM2-01. Zonally averaged salinity bias relative to WOA13 for (b) ACCESS-OM2; (d) ACCESS-OM2-025 and (f) ACCESS-OM2-01. The WOA13 zonally averaged temperature field is shown in (g) and the WOA13 zonally averaged salinity field in (h). Model fields are 1993–2017 means.

There are well known issues with inferring poleward heat transport from reanalysis products and there are large variations between different products (e.g. Griffies et al., 2009; Valdivieso et al., 2017). Furthermore, the model simulations are more consistent with the inferred heat transport from the JRA55-do forcing itself at these low latitudes, particularly in the Southern Hemisphere (see Fig. 30 of Tsujino et al., 2018). The models still underestimate the peak in northward heat transport at 20° N

5 ($\sim 1.3$ PW in ACCESS-OM2-025 and ACCESS-OM2-01 compared to $\sim 1.5$ PW from the JRA55-do forcing). The reason for this mismatch remains unclear and worthy of further investigation. Nonetheless, the results show that ACCESS-OM2-025 has a clear advantage in representing heat transport over ACCESS-OM2, suggesting that 0.25° models may have an improved climate in the upcoming CMIP6 simulations, compared with coarser resolution climate models.





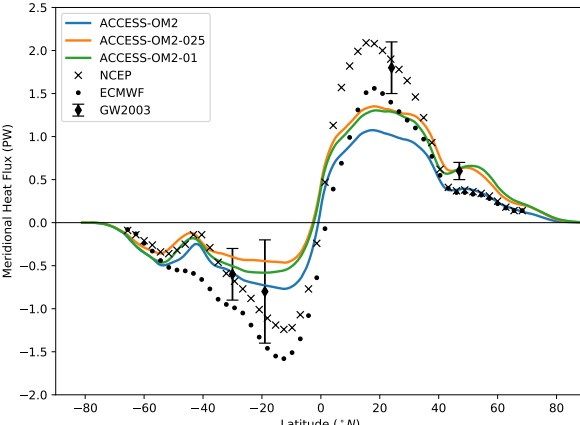

**Figure 13.** Total meridional heat transport from each of the ACCESS-OM2 configurations (solid lines). Also included are observational estimates from Trenberth and Caron (2001) inferred from both NCEP (black crosses) and ECMWF (black dots) reanalysis data over the period February 1985 – April 1989, and from the World Ocean Circulation Experiment (Ganachaud and Wunsch, 2003, black diamonds with error bars).

## 4.2 Regional ocean circulation

The second part of this model evaluation involves examining the performance of the model at a selected number of key regions. In these regional analyses we will focus on the major circulation features such as the separation of western boundary currents, the average state of equatorial currents and flow through major choke points. The regional evaluation is not intended to be exhaustive; but instead will outline regions in which the model behaves well or poorly. It is envisaged that more in-depth analyses will be published using this model in the near future.

### 4.2.1 Southern Ocean

A significant driver in moving towards high resolution ocean models is to better represent the dynamics of the Southern Ocean, where mesoscale variability is critical in capturing the evolution of the system (e.g. Hogg et al., 2015). An example of the improvement in water mass properties can be seen in Fig. 14, where transects of temperature and salinity along the SR3 hydrographic line are compared with historical observations. Here, progressively enhancing the resolution leads to better representation of the observed surface low-salinity layer, enhanced subduction into the mid-depths, and improved Antarctic shelf properties and abyssal temperature-salinity structure (bearing in mind that the ACCESS-OM2-01 simulation is less well equilibrated, and thus has had less opportunity to drift away from the initial climatology).

Figure 15 shows a meridional transect of planetary geostrophic potential vorticity ($PV = -\frac{fg}{\rho_0}\frac{\partial \sigma_0}{\partial z}$) across a Subantarctic Mode Water (SAMW) formation region at $120°$ W. The blue lines show maximum mixed layer depth (MLD) representative of the winter season when SAMW is ventilated, and black lines show minimum MLD representative of the summer season.





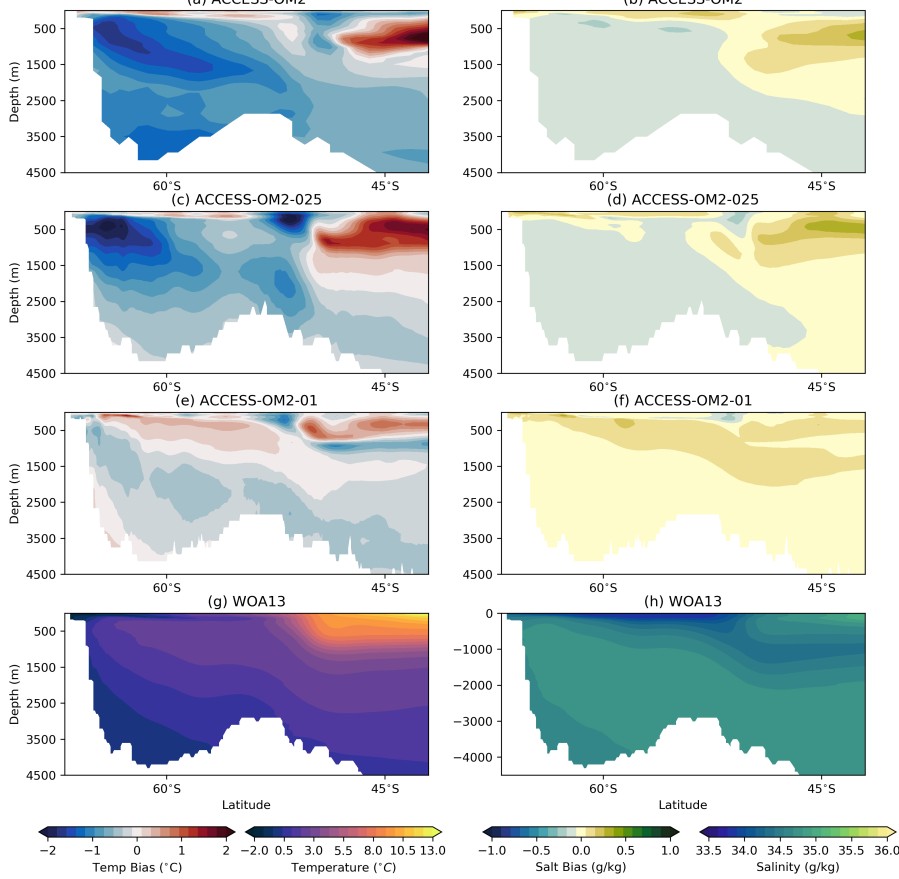

**Figure 14.** Meridional transects of 1993–2017 mean potential temperature (left panels) and salinity (right panels) south of Tasmania, along longitude 150°W, near the WOCE repeat hydrographic line SR3, for (a), (b) ACCESS-OM2; (c), (d) ACCESS-OM2-025; (e), (f) ACCESS-OM2-01; and (g), (h) gridded climatologies from WOA13 for the period 1985–2013.

For all model resolutions, mixed layer depths are very similar. Nevertheless, maximum mixed layer depths are deeper than observations suggest, especially in ACCESS-OM2 and ACCESS-OM2-025 (data not shown). Bias in the MLD may be due to a number of factors, such as bias in the surface buoyancy forcing (Sallée et al., 2013), systematic errors in the convective parameterisation, sub-grid scale turbulence and friction schemes (Dufresne et al., 2013) and the representation of submesoscales

5 (Wenegrat et al., 2018). Determining the exact cause of the bias requires a careful analysis of the mixed layer budgets and is beyond the scope of this article.

Mean mixed layer depths are insensitive to model resolution, but the distribution of planetary geostrophic PV changes substantially: for the highest resolution configuration, ACCESS-OM2-01, there are two distinct layers of high PV magnitude at about 900 m and 1400 m depth, while in both ACCESS-OM2 and ACCESS-OM2-025 these spread in the diapycnal direction

10 and merge into a single, somewhat deeper layer. At the same time, PV in the mode water layer increases in magnitude with



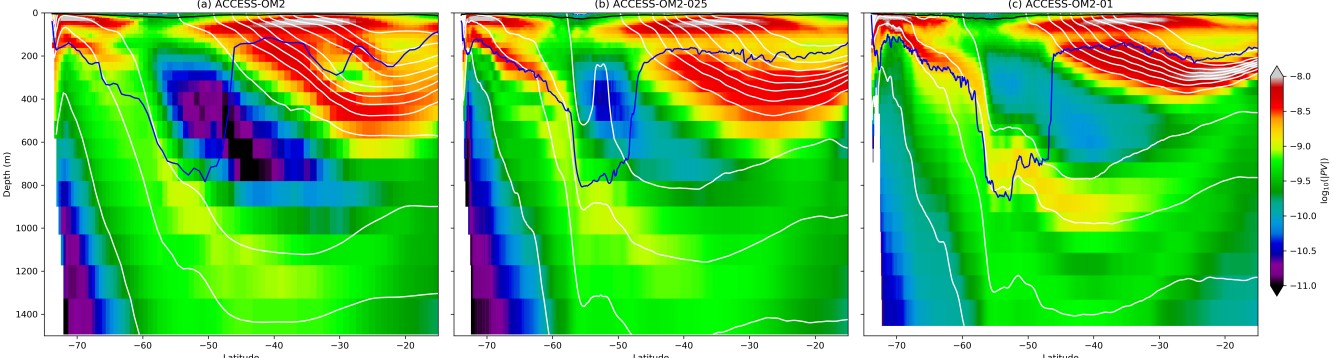

**Figure 15.** Meridional transects of time-mean planetary geostrophic potential vorticity (PV, s$^{-3}$) across a Subantarctic Mode Water (SAMW) formation region at 120° W in the three configurations. Colors represent $\log_{10}(|PV|)$, white lines represent $\sigma_0$ (contour interval 0.25 kg m$^{-3}$), and black (blue) lines represent the minimum (maximum) mixed layer depth, defined by a 0.03 kg m$^{-3}$ density criterion.

increasing resolution. From our current understanding of mode water ventilation it is not clear how these differences, i.e. winter mixed layers which are too deep in the models and differences in the distribution of PV, will affect the uptake of tracers, such as heat and carbon, into the ocean interior.

Sea level variability in the region of the Agulhas Current is shown in Fig. 16 (colours) for each model resolution, includ-
ing a comparison with observations. Variability follows contours of the barotropic streamfunction (white contours) down the Mozambique channel (de Ruijter et al., 2002) and East Madagascar coast, continuing along the southeast coast of Southern Africa. There is a peak in variability in all simulations where the Agulhas Current retroflects at the southern tip of the African continental shelf. From here, variability continues both west into the South Atlantic Basin along the path of the Agulhas Rings (Dencausse et al., 2010), and east along the Agulhas Return Current. The peak in this variability is well captured in the
ACCESS-OM2-01 simulation relative to observations, whereas variability amplitudes in the ACCESS-OM2-025 simulation are about half those observed and the variability in ACCESS-OM2 is substantially less again. The path of the circulation in the region before the retroflection, as indicated by the contours of barotropic streamfunction, is consistent between the simulations and the observations. A sea level variability hotspot is well captured to the south of the main retroflection in the ACCESS-OM2-01 simulation, upstream of the Prince Edward Islands, around 48° S, 30° E, over the south-west Indian Ridge (Ansorge
et al., 2012). ACCESS-OM2-025 also captures this feature with reduced amplitude, while it is missing in ACCESS-OM2.

#### 4.2.2 Australasia

In the south-west Pacific Ocean the westward South Equatorial Current bifurcates at the Australian coast at about 16° S, with the southward branch forming the southward-intensifying East Australian Current (EAC). Between 33°S and 35°S the EAC splits into an eddying eastward outflow (known as the Tasman Front), and the EAC extension, an alongshore southward-
weakening eddy-dominated flow (Ridgway and Dunn, 2003). Sea level standard deviation in ACCESS-OM2-01 (Fig. 17c) reproduces the observed spatial distribution of eddy activity in this region (Fig.17d) but underestimates its magnitude. The





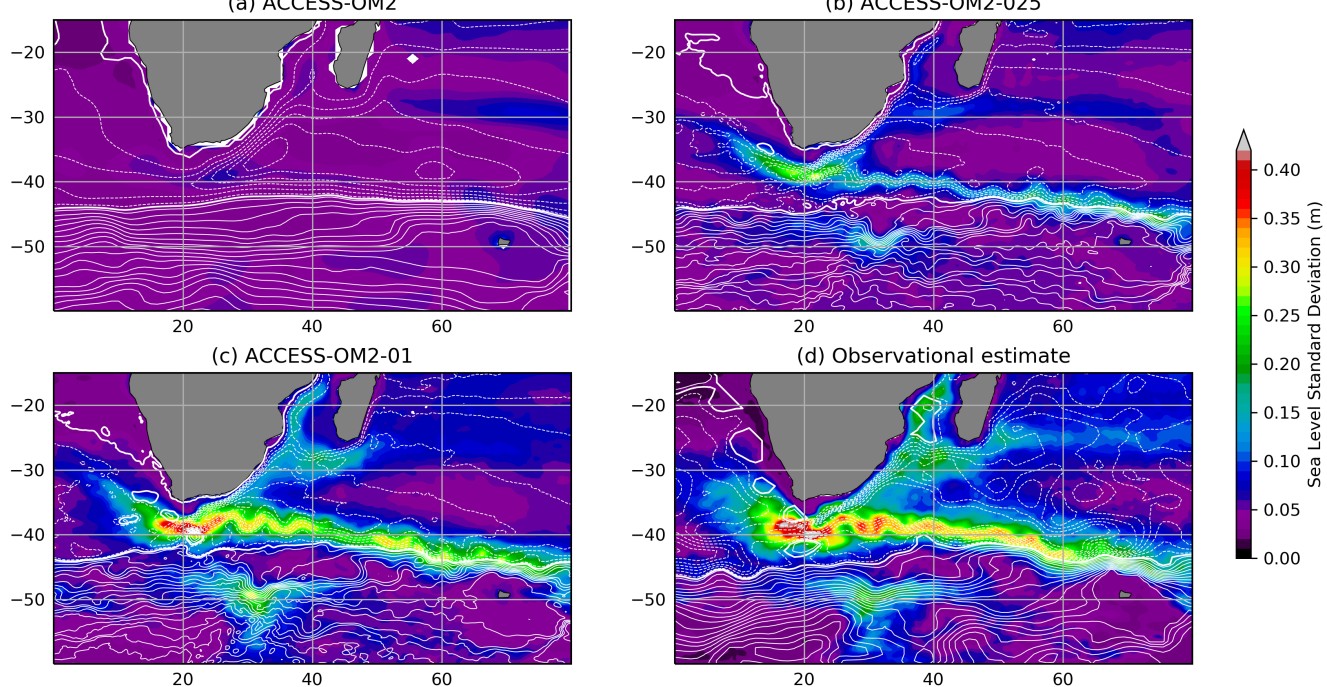

**Figure 16.** Standard deviation of Sea Level Anomaly (colours) overlain with contours of mean barotropic streamfunction (contour interval 10 Sv) in the Agulhas region for (a) ACCESS-OM2; (b) ACCESS-OM2-025; (c) ACCESS-OM2-01 and (d) AVISO gridded analysis of satellite altimetry and 1° gridded barotropic streamfunction estimated from hydrography and Argo displacements (Colin de Verdière and Ollitrault, 2016).

variability is more severely underestimated in the coarser configurations (Fig. 17a, b). ACCESS-OM2-025 retains a weak qualitative signature of both the Tasman Front and the EAC extension, whereas these are nearly absent in ACCESS-OM2.

Contours of barotropic streamfunction converge near Australia's east coast in observations (Fig. 17d), but the transport of the EAC is underestimated in all model configurations. Furthermore, the EAC broadens (as expected) with the reduction of horizontal resolution. The poleward-only 1993–2017 mean EAC transports above 2000 m at 28° S are 18.7, 17.5 and 17.2 Sv in ACCESS-OM2, ACCESS-OM2-025 and ACCESS-OM2-01 (respectively), somewhat weaker than 22.1 Sv observed by Sloyan et al. (2016) at 27° S. The South Equatorial Current is also weaker than observed (Sect. 4.2.3), suggesting a weak South Pacific wind-driven circulation in all ACCESS-OM2 models.

The Indonesian Throughflow (ITF) from the Pacific Ocean to the Indian Ocean is the only tropical inter-ocean pathway in the global ocean circulation and its magnitude through key straits is a good indicator of the robustness of the model for this region. The total ITF transport in ACCESS-OM2-01 agrees well with INSTANT observations by Sprintall et al. (2009) (green and dashed lines in Fig. 18a), but the total transport is too weak in the coarser configurations. The detailed breakdown of transport through the three main straits (Lombok Strait, Ombai Strait and Timor Passage, Fig. 18b–d) shows that there are





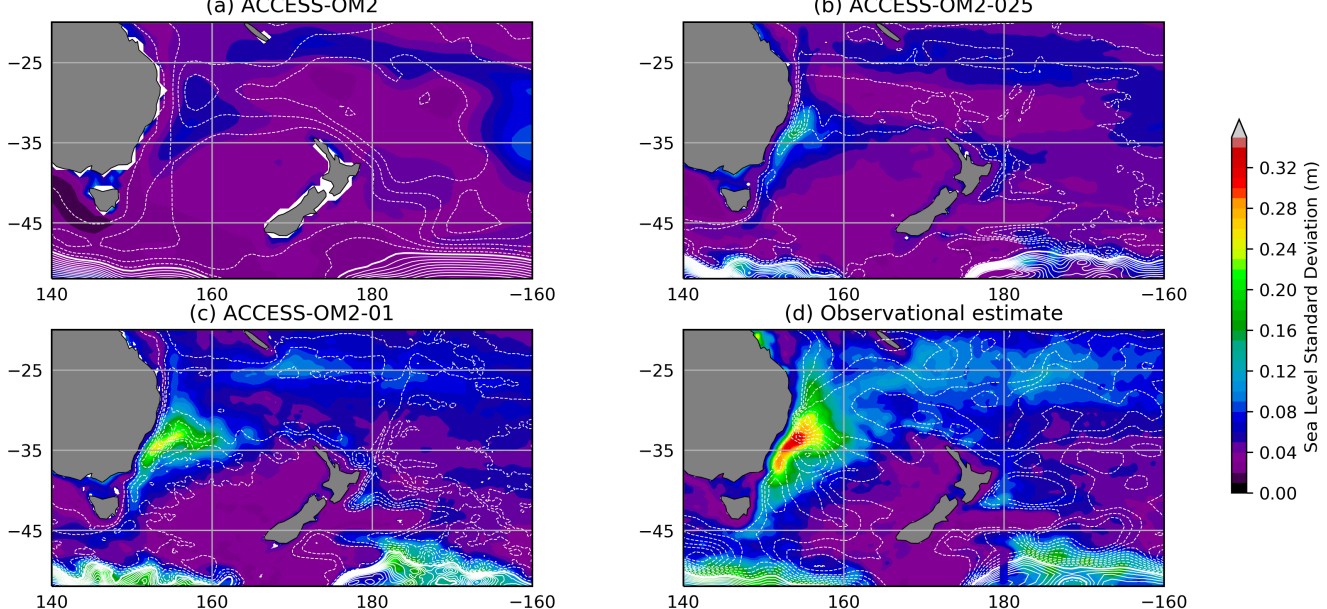

**Figure 17.** Standard deviation of Sea Level Anomaly (colours) overlain with contours of mean barotropic streamfunction (contour interval 5 Sv) in the East Australian Current region for (a) ACCESS-OM2; (b) ACCESS-OM2-025; (c) ACCESS-OM2-01 and (d) AVISO gridded analysis of satellite altimetry and 1° gridded barotropic streamfunction estimated from hydrography and Argo displacements (Colin de Verdière and Ollitrault, 2016).

compensating biases in individual straits in ACCESS-OM2-01 and the coarser configurations may over- or under-estimate transport in each strait. Models may underestimate the magnitude of the total transport, or the transport in individual straits, for three primary reasons. Firstly, the ITF transport from the Pacific to the Indian Ocean is induced by the sea level gradient between these two oceans; in ACCESS-OM2 and ACCESS-OM2-025, this sea level gradient is weaker than observed and 10%

5  smaller than in ACCESS-OM2-01 (Fig. 5); thus, it is not strong enough to reproduce the observed total transport. Secondly, Lombok and Ombai Straits are narrow (minimum width 20 and 40 km, respectively) and therefore require a high horizontal resolution to faithfully represent the strait transport. For example, the width of Lombok Strait is 1 velocity cell (∼110 km) in ACCESS-OM2, 1 cell (∼28 km) in ACCESS-OM2-025 and 2 cells (∼22 km) in ACCESS-OM2-01; Rayleigh drag (Sect. 2.1.4) is used in ACCESS-OM2 to obtain more realistic transport through Lombok and Ombai Straits. Thirdly, the ITF outflow is split

10  between the three main straits flowing first through the Lombok Strait, then the Ombai Strait and finally the Timor Strait. So if more of the water that comes through the Makassar Strait goes through Lombok strait (as in ACCESS-OM2-025), less water will go through the Timor Passage. As a consequence, the resolution of straits is critical for this region, and for this reason the ACCESS-OM2-01 configuration is more appropriate to study the Indonesian Seas.





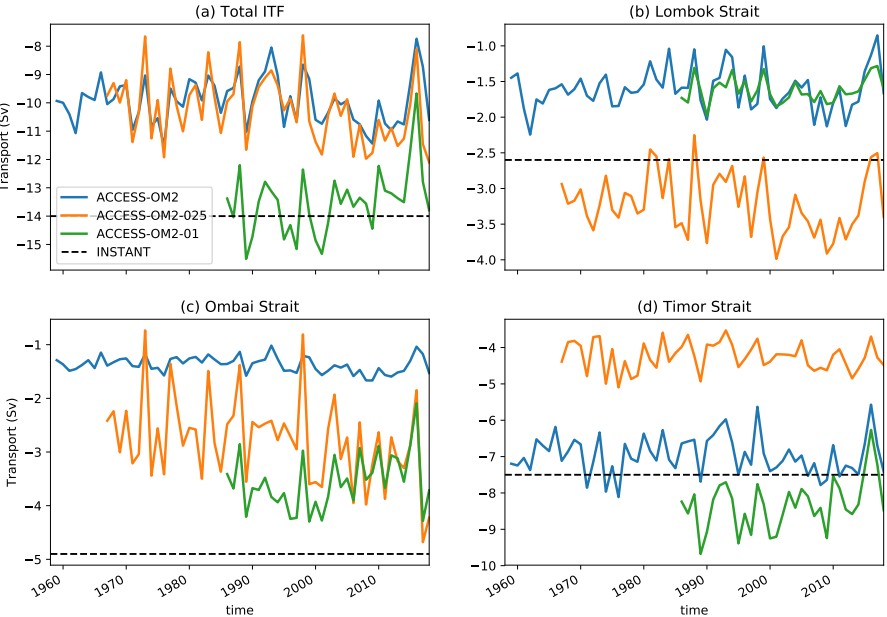

**Figure 18.** Timeseries of annual-mean transport through the Indonesian Straits. The total Indonesian Throughflow (panel a) can be broken into (b) Lombok Strait; (c) Ombai Strait and (d) Timor Strait. Black dashed lines indicate the mean throughflow during 2004–2006 from the INSTANT program (Sprintall et al., 2009). Negative values indicate southward flow.

### 4.2.3 Pacific Ocean

All three versions of ACCESS-OM2 reproduce the major features of the equatorial Pacific Ocean circulation well compared with observations from Johnson et al. (2002) (Fig. 19), which are in turn similar to measurements from the TAO array on the Equator at $140°$ W and $165°$ E (not shown). The strength of the Equatorial Undercurrent (EUC) core is within 10% of
5    observations and its latitudinal width at $140°$ W is accurate in ACCESS-OM2-025 and ACCESS-OM2-01 but somewhat too wide in ACCESS-OM2. The EUC extends too deeply in both ACCESS-OM2-025 and ACCESS-OM2-01. The strength of the thermocline is well reproduced in ACCESS-OM2 and ACCESS-OM2-025, although in ACCESS-OM2-01 it is slightly too strong. The strong Pacific thermocline in ACCESS-OM2-01 also appears in the zonal mean (Fig. 12) and Atlantic (Fig. 23) and may be a consequence of the lack of background diffusivity (e.g. Meehl et al., 2001) and reduced numerical diffusion.
10    The vertical temperature gradient above the thermocline appears to be too weak in all three configurations, a bias which may also be linked to the weak vertical shear in the upper EUC. Further, both the northern and southern branches of the South Equatorial Current (SEC, the westward surface-intensified current south of $\sim 5°$ N) are too weak in the models. These biases in the SEC and upper EUC may be associated with problems in the turbulent mixing parameterizations in this region, but a detailed sensitivity study has not yet been undertaken. The eastwards North Equatorial Counter Current (NECC) at $7°$ N is



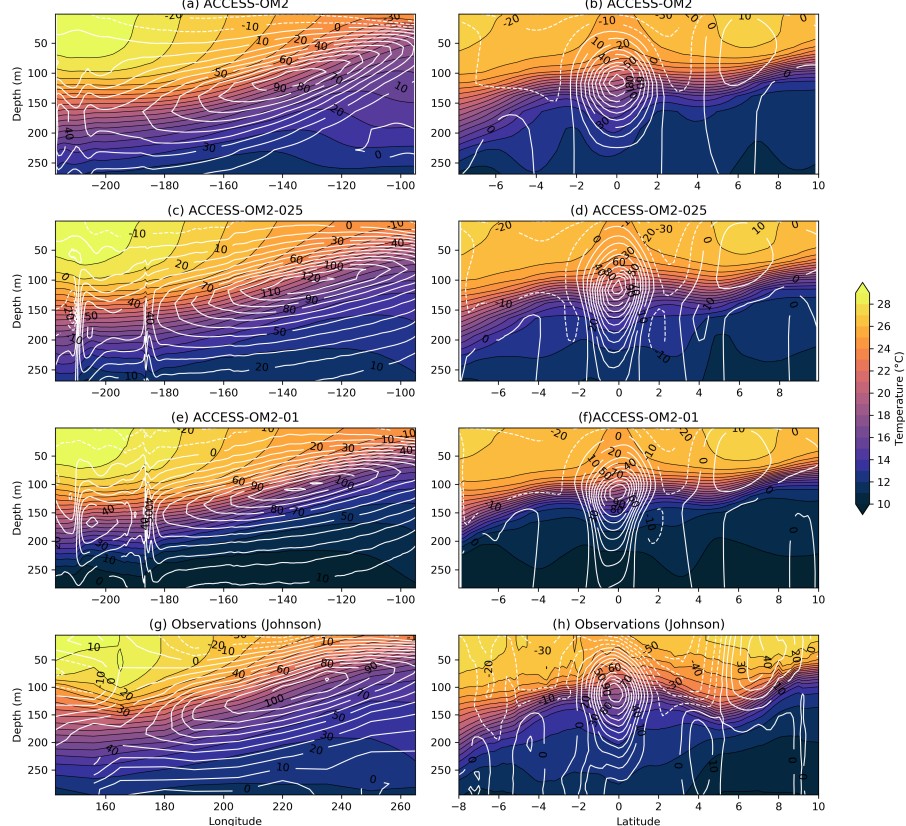

**Figure 19.** Comparison of temperature (color and contours every $1°C$) and zonal velocity (white contours every $10\,\mathrm{cm\,s^{-1}}$ with black labels in $\mathrm{cm\,s^{-1}}$) along the Equator (left) and at $220°$ E (right) in the Pacific for (a), (b) ACCESS-OM2; (c), (d) ACCESS-OM2-025; (e), (f) ACCESS-OM2-01 and (g), (h) observations (Johnson et al., 2002).

very weak in the models. A weak northern SEC branch and NECC are common biases in ocean models (e.g Large et al., 2001; Tseng et al., 2016).

Fig. 20 shows meridional transects of the climatological means of potential temperature and salinity across the approximate centre of the basin at 150°W (near the WOCE repeat hydrography line P16) for each model configuration. For comparison, we

5    include an observational estimate of the climatological mean along the same transect, taken from the gridded WOA13 product, for the period 1985-2013 (Fig. 20g, h). In general, all three model configurations produce a realistic thermal structure in this basin. In particular, the models capture the approximate depth of the thermocline and its inter-hemispheric asymmetry (with the southern hemisphere thermocline being somewhat deeper than in the northern hemisphere), the strong temperature gradients in the Southern Ocean at approximately 55°S, coincident with the location of the Antarctic Circumpolar Current, and the weak

10    vertical gradients to the north of the ACC in the regions associated with southern hemisphere mode-water production. However, at approximately 50°S, this region of weakly stratified water is substantially deeper in the high-resolution ACCESS-OM2-01





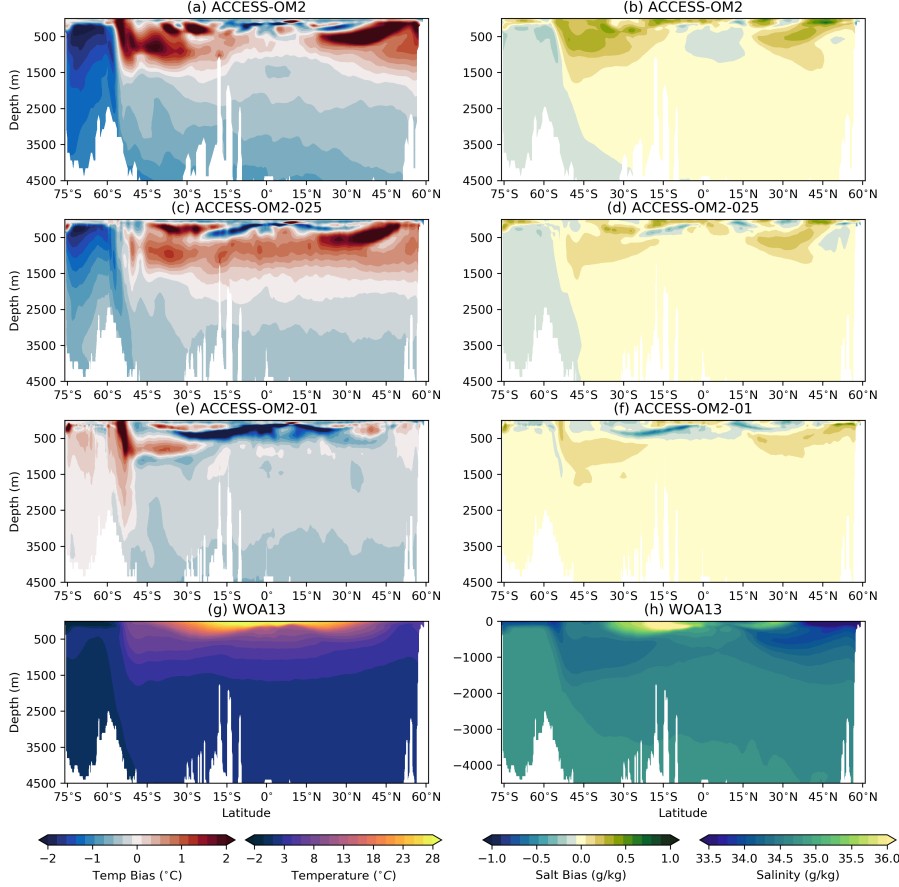

**Figure 20.** Meridional transects of 1993–2017 mean potential temperature (upper panels) and salinity (lower panels) in the central Pacific Ocean, along longitude 150°W, near the WOCE hydrographic line P16, for for (a), (b) ACCESS-OM2; (c), (d) ACCESS-OM2-025; (e), (f) ACCESS-OM2-01; and (g), (h) gridded climatologies from WOA13 for the period 1985–2013.

simulation than in either the ACCESS-OM2 or ACCESS-OM2-025 configurations, or the WOA13 observations, suggestive of overproduction of sub-Antarctic mode water.

In contrast to the temperature structure, which was simulated reasonably well by the various models in this suite, the meridional haline structure of the central Pacific is not well simulated. In particular, none of the models reproduce the observed deep
5  salinity minimum in either hemisphere, although there is some suggestion of penetration of relatively fresh waters into the interior at approximately 55°S and 45°N in the ACCESS-OM2-025 and ACCESS-OM2-01 configurations. As such, it is likely that the models' representation of Pacific mode and intermediate waters will be affected by the poor representation of the deep salinity structure, which could, in turn, have implications for the local overturning circulation (Thompson et al., 2016).

Fig. 21 shows sea level variability and barotropic streamfunction in the North Pacific, including the region of the Kuroshio
10  Current. ACCESS-OM2 simulations at each resolution show variability focused near the separation from the coast of Japan,





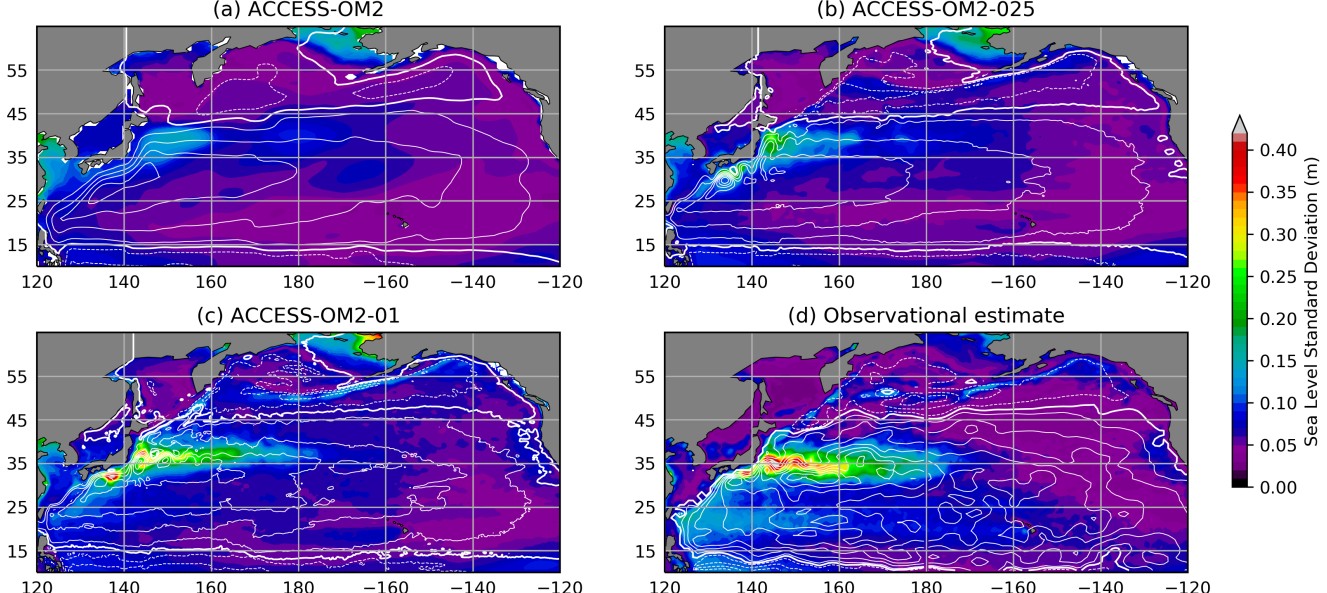

**Figure 21.** Standard deviation of Sea Level Anomaly (colours) overlain with contours of mean barotropic streamfunction (contour interval 10 Sv) in the Kuroshio region for (a) ACCESS-OM2; (b) ACCESS-OM2-025; (c) ACCESS-OM2-01 and (d) AVISO gridded analysis of satellite altimetry and $1°$ gridded barotropic streamfunction estimated from hydrography and Argo displacements (Colin de Verdière and Ollitrault, 2016).

decaying eastward along the Kuroshio extension. However, the observed variability continues with significant amplitude further along the extension. Peak variability here in ACCESS-OM2-01 matches the observed amplitude ($\sim 0.4$ m), whereas ACCESS-OM2-025 has a reasonable distribution of variability with reduced magnitude (peak $\sim 0.25$ m), and ACCESS-OM2 substantially underestimates this (0.15 m). ACCESS-OM2-01 has variability upstream of the separation point, higher than that observed, where the Kuroshio's 'large meander' appears on interannual timescales (Kawabe, 1995). In ACCESS-OM2-01 the barotropic streamfunction has a similar structure to the observational estimate, but seems somewhat weaker in amplitude (white contours in Figs. 21c, d). The directly observed mean Kuroshio transport on the WOCE PCM-1 line between Taiwan and the southern Ryukyu Islands was $21.5\pm2.5$ Sv between September 1994 and May 1996 (Johns et al., 2001). Corresponding transports over the same period are close to this value in ACCESS-OM2-025 and ACCESS-OM2-01 (17.5 Sv and 20.1 Sv, respectively), but much weaker (7.6 Sv) in ACCESS-OM2, which is lower than in other models of this resolution under CORE-II forcing (Tseng et al., 2016).

### 4.2.4 Atlantic Ocean

Accurately representing the horizontal circulation of the North Atlantic is a persistent challenge for ocean modellers. In particular, models commonly fail to simulate a Gulf Stream that separates from the North American coast at Cape Hatteras ($35°$ N,



75° W), and a North Atlantic Current that flows north along the east side of the Grand Banks from 40° to 51° N (Rossby, 1996; Chassignet and Marshall, 2008). Consistent with other modeling studies (e.g. Bryan et al., 2007), we find considerable improvement in the Gulf Stream transport and separation latitude at eddy resolving resolution (Fig. 22). In ACCESS-OM2 and ACCESS-OM2-025, the Gulf Stream flow is too weak and overshoots the separation latitude by nearly 5° latitude relative to

observations; in ACCESS-OM2-025 the separation latitude is also highly variable. The mean Gulf Stream separation latitude in ACCESS-OM2-01 is correct, but it is more variable than observed, occasionally separating 2–3° too far north; this appears as excessive SSH variability shortly after separation and broadening of the separated jet in Fig. 22c relative to observations. In addition, the transport is characterized by an overly strong anti-cyclonic recirculation near the separation (35° N, 70° W), whereas observations indicate a more uniform flow towards the Grand Banks. None of these models adequately captures the

north flow of the North Atlantic Current along the east side of the Grand Banks, resulting in significant biases in the sea surface temperature and salinity (Figs. 10 and 11). Large biases in North Atlantic tracer distributions and air-sea fluxes are commonly attributed to the misrepresentation of these flows (Bryan et al., 2007). ACCESS-OM2-025 and ACCESS-OM2-01 also simulate the Loop Current (although with lower variability than the altimetric estimate), which flows through the Florida Straits to join the Florida Current and Gulf Stream. In contrast, the Caribbean circulation is incorrect in ACCESS-OM2, with the gyre circu-

lation closed primarily via the Bahamas rather than the Florida Straits. Transport between Florida and Grand Bahama Island at 27° N is 23.4 Sv in ACCESS-OM2-025 and 20.4 Sv in ACCESS-OM2-01, significantly weaker than the observed 32.1 Sv (Meinen et al., 2010). This reduced inertia may contribute to the poor Gulf Stream separation, as seen in idealised experiments by Özgökmen et al. (1997), but more investigation is required. Identifying a numerical recipe that accurately simulates these North Atlantic features remains an ongoing challenge for modellers.

Fig. 23 shows meridional transects of the mean potential temperature and salinity across the approximate centre of the Atlantic basin at 25°W (near the WOCE repeat hydrography line A16) for each model configuration and the gridded WOA13 product. In general, while all three model configurations produce the basic hydrographic structure of the central Atlantic, several aspects are poorly represented, particularly by the two coarser resolution simulations. For example, observations show marked inter-hemispheric asymmetry in the thermocline structure (with the southern hemisphere thermocline being substan-

tially shallower than the northern hemisphere), while the ACCESS-OM2 and ACCESS-OM2-025 configurations have approximately equal thermocline depths in both hemispheres thanks to a strong warm bias at mid-depth in the southern hemisphere. This problem is ameliorated in ACCESS-OM2-01, which produces a southern hemisphere thermocline with a 10°C isotherm that is approximately 800m deeper at 40°N than 40°S, which is similar to that obtained from the WOA13 product (∼900m), although we note that the thermocline is deeper in the North Atlantic in the WOA13 observations than in the ACCESS-OM2-01

fields.

Similarly, the ACCESS-OM2 and ACCESS-OM2-025 models do not produce the southern hemisphere deep salinity minimum observed at a latitude of approximately 50°S and at a depth of approximately 1000 m. However, the salinity minimum is reproduced quite well by the ACCESS-OM2-01 model, which is able to capture both the structure and approximate magnitude. Curiously, while the high resolution ACCESS-OM2-01 is not able to capture the northern hemisphere deep salinity maximum

(present in the WOA13 observations at an approximate latitude of 40°N and an approximate depth of 1100m), both the lower





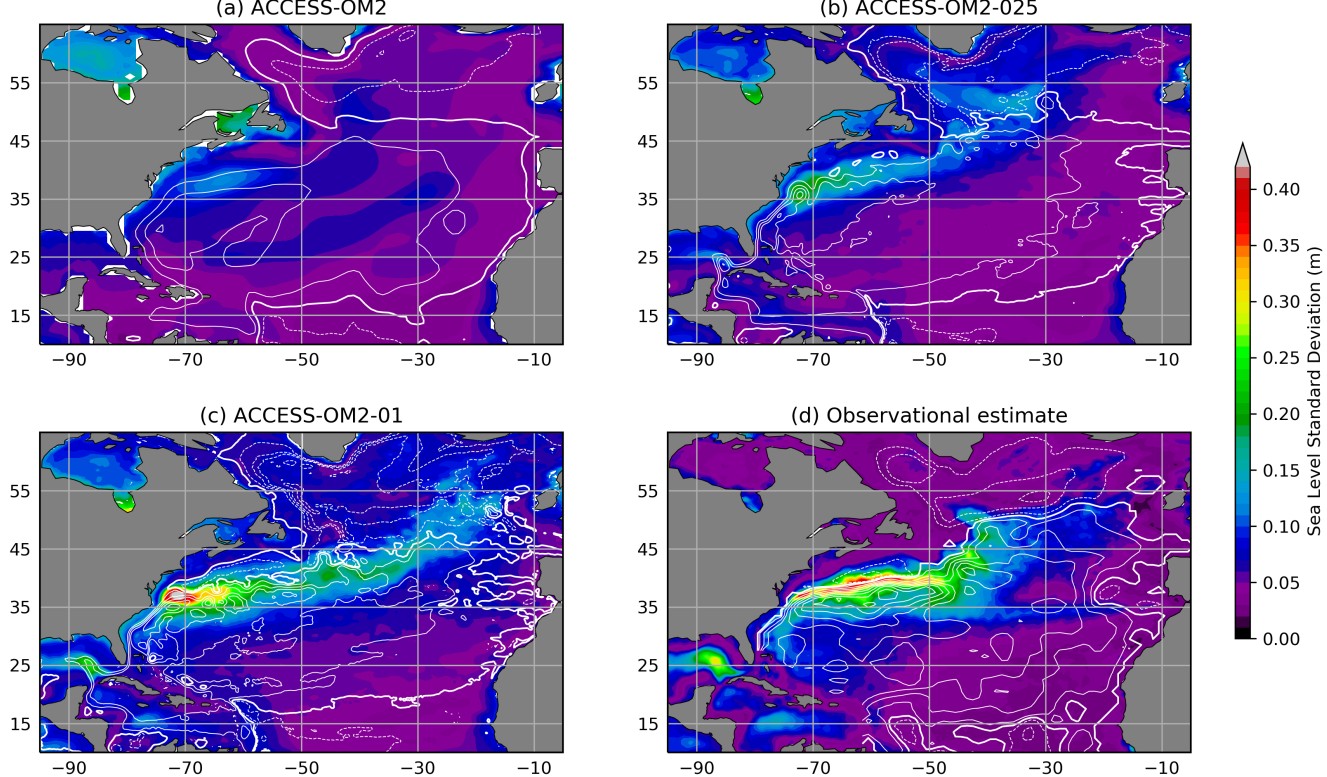

**Figure 22.** Standard deviation of Sea Level Anomaly (colours) overlain with contours of mean barotropic streamfunction (contour interval 10 Sv) in the Gulf Stream region for (a) ACCESS-OM2; (b) ACCESS-OM2-025; (c) ACCESS-OM2-01 and (d) AVISO gridded analysis of satellite altimetry and 1° gridded barotropic streamfunction estimated from hydrography and Argo displacements (Colin de Verdière and Ollitrault, 2016).

resolution ACCESS-OM2 and ACCESS-OM2-025 configurations capture this feature with varying degrees of fidelity (the high salinity zone is too broad in the ACCESS-OM2 simulation and too shallow by 100-200m in both configurations).

The Brazil Current flows southward along the western boundary of the South Atlantic Ocean, separating at $\sim 40°$ S. The mean surface speed of this current in ACCESS-OM2-025 and ACCESS-OM2-01 is of comparable magnitude to, but weaker than, observations (Fig. 24b–d), and is strongly underestimated in ACCESS-OM2 (Fig. 24a). This underestimation is most clear for the upper 400 m of this current (not shown). Weakening of the Brazil Current is expected at the lowest resolution due to the broadening of the current by the enhanced viscosity near the western boundary. The Malvinas Current, flowing northwards along the boundary south of 40° S, is highly steered by bathymetry (Fig. 24d, h) and is well represented in ACCESS-OM2-01 (Fig. 24c, g), including its northward penetration along the shelf break up to 40° S. The Brazil-Malvinas Confluence mean latitude ($\sim$38° S, Fig. 24d, h) is captured in ACCESS-OM2-01, but is too far south in ACCESS-OM2-025 and ACCESS-OM2. The Zapiola Anticyclone (ZA) appears clearly at $\sim$42–48° S, $\sim$36–48° W in the Colin de Verdière and Ollitrault (2016)



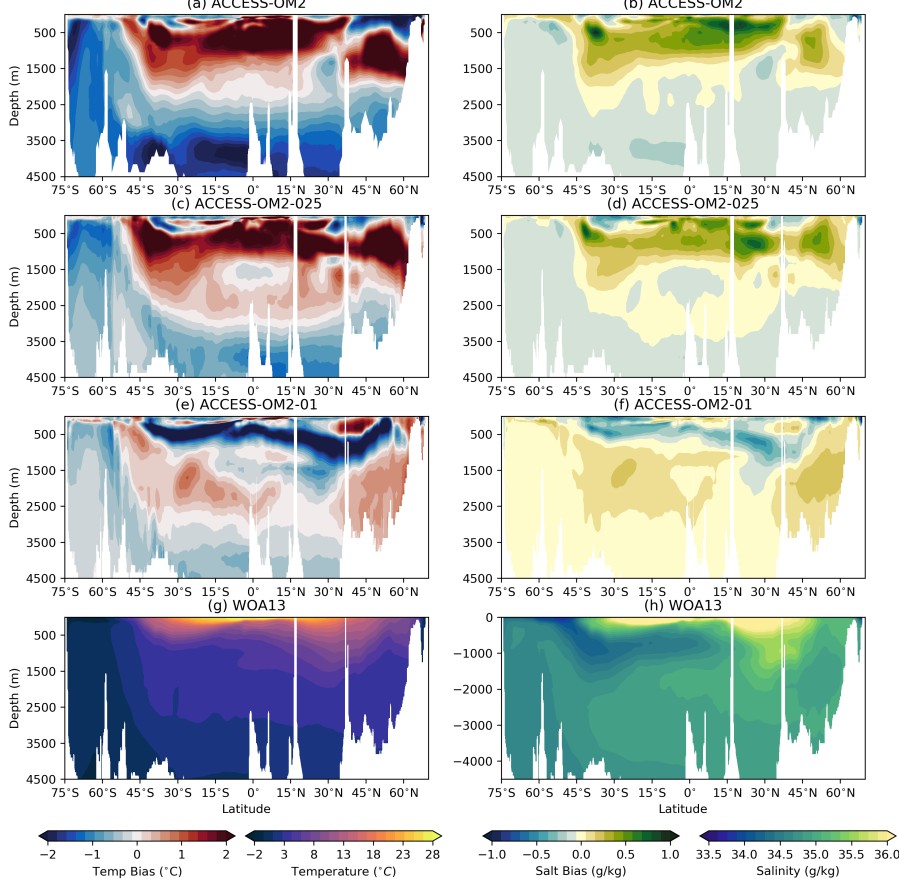

**Figure 23.** Meridional transects of 1993–2017 mean potential temperature (left panels) and salinity (right panels) in the central Atlantic Ocean, along longitude 25°W, near the WOCE hydrographic line A16, for (a), (b) ACCESS-OM2; (c), (d) ACCESS-OM2-025; (e), (f) ACCESS-OM2-01; and (g), (h) gridded climatologies from WOA13 for the period 1985–2013.

barotropic streamfunction (contours in Fig. 24h), but is weaker in ACCESS-OM2-01, indistinct in ACCESS-OM2-025 and absent in ACCESS-OM2 (Fig. 24g, f, e), consistent with the ZA being eddy-driven (Dewar, 1998; de Miranda et al., 1999); the poor representation at coarse resolution is associated with significant SST and SSS biases (Figs. 10 and 11). The sea level standard deviation forms a distinctive horseshoe pattern (Fu, 2006) around the ZA in the AVISO product (colours in Fig. 24h), which is partially captured in ACCESS-OM2-01, although at lower amplitude. The sea level variability pattern and amplitude in ACCESS-OM2-025 differ significantly from AVISO, and variability is negligible in ACCESS-OM2.

### 4.2.5 Indian Ocean

As in previous sections, we plot time-mean transects of potential temperature and salinity across the central Indian ocean (longitude 95°E, near the I08–09 WOCE repeat hydrography line) from the three different model configurations, as well as





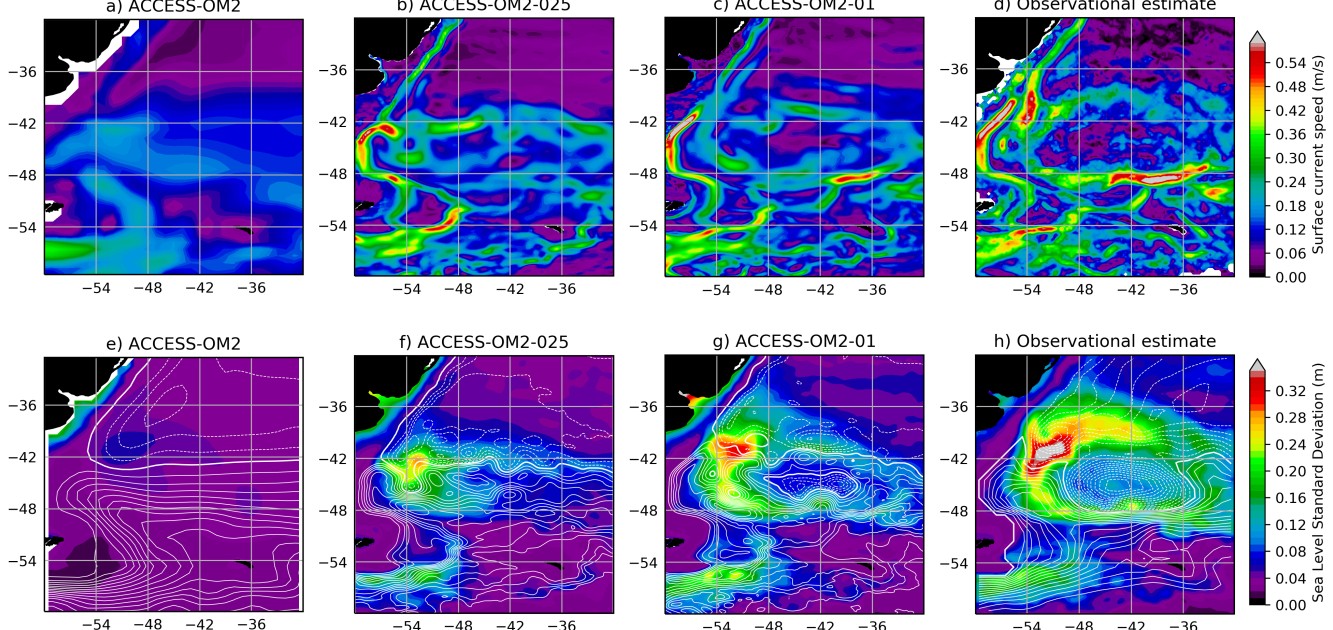

**Figure 24.** Mean surface current speed in the southwest Atlantic in (a) ACCESS-OM2, (b) ACCESS-OM2-025, (c) ACCESS-OM2-01, and (d) estimated from drifter data (Laurindo et al., 2017); standard deviation of Sea Level Anomaly (colours) overlain with contours of mean barotropic streamfunction (contour interval 10 Sv) for e) ACCESS-OM2, (f) ACCESS-OM2-025, (g) ACCESS-OM2-01 and (h) AVISO gridded analysis of satellite altimetry and 1° gridded barotropic streamfunction estimated from hydrography and Argo displacements (Colin de Verdière and Ollitrault, 2016).

from the WOA13 climatology, for the period 1985–2013. In the Indian Ocean, all three model configurations reproduce the basic structure of both temperature and salinity extremely well, including the high meridional temperature gradients near 50°S associated with the ACC, the low vertical temperature gradients near 40° S associated with the formation of mode and intermediate waters, the deep salinity minima in the southern hemisphere at around 1500m depth, and the band of very

5   fresh surface waters north of the equator. The primary model biases are the cool water generated by convection near the southern boundary and a deep midlatitude thermocline in ACCESS-OM2 and ACCESS-OM2-025, while ACCESS-OM2-01 has minimal biases in this region.

We also assess the annual and seasonal mean variability of the thermocline depth (D20) in the western tropical Indian Ocean (50°E-75°E, 5°S-10°S), known as the Seychelles Dome (Yokoi et al., 2008; Hermes and Reason, 2008) for 1985-2013 using

10   the 20°C isotherm proxy. All three model resolutions are able to simulate the basic large-scale annual mean D20 structure of the Indian Ocean (Fig. 26). In the coarse models (Fig. 26a, b), the D20 is deeper than observed (Fig. 26d) in four areas: the western Arabian Sea, the Seychelles Dome (black box), along the Mozambique Channel and across 15°S–25°S (particularly on the eastern side). The same model differences relative to WOA13 are evident over all seasons (not shown), for both the Seychelles





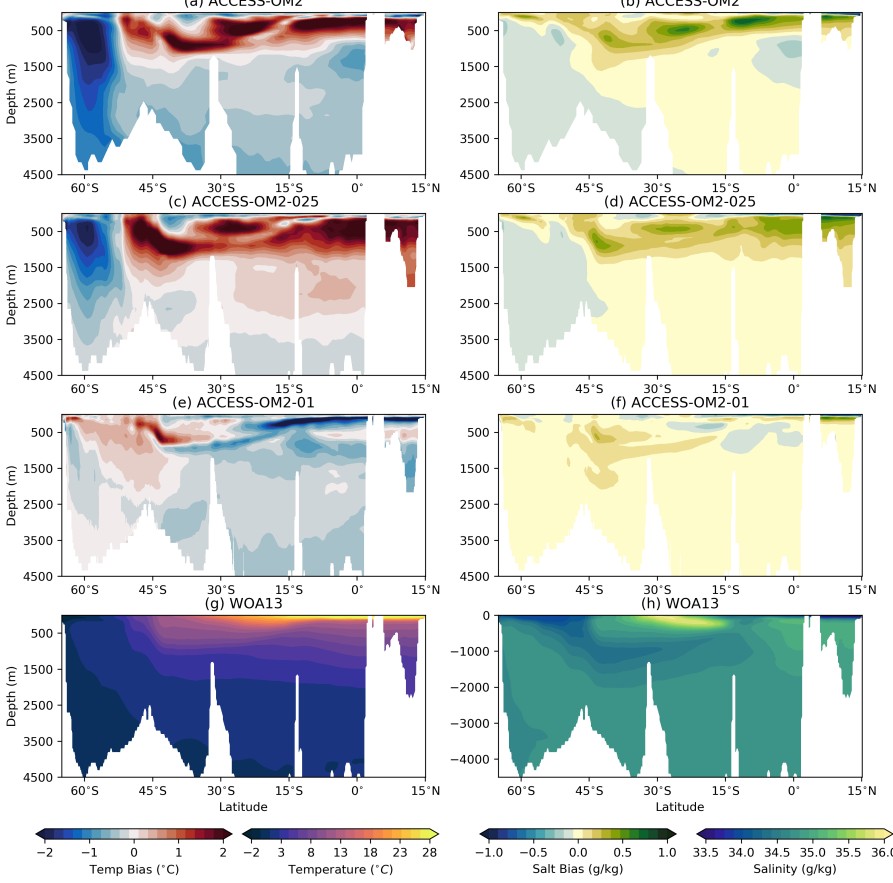

**Figure 25.** Meridional transects of 1993–2017 mean potential temperature (upper panels) and salinity (lower panels) in the central Indian Ocean, along longitude 95°E, near the WOCE hydrographic lines I08 and I09, for (a), (b) ACCESS-OM2; (c), (d) ACCESS-OM2-025; (e), (f) ACCESS-OM2-01; and (g), (h) gridded climatologies from WOA13 for the period 1985–2013.

Dome region and the large-scale Indian Ocean. Despite a general tendency to underestimate the D20 in the ACCESS-OM2-01 within the tropical Indian Ocean (north of 10° S; Fig. 26c), the higher resolution model compares best with the observed annual and seasonal spatial pattern in the Seychelles Dome region. This result suggests that higher ocean resolution simulations are important to capture the Seychelles Dome thermocline variability and its role in regional weather and climate, from tropical cyclones (Xie et al., 2002) to rainfall in Africa (Annamalai et al., 2005; Behera et al., 2005), India (Izumo et al., 2008) and Australia (Taschetto et al., 2011).

## 4.3 Sea ice

Our coupled ocean-ice model runs yield acceptable simulations of the Arctic and Antarctic sea ice extent and concentration at all three horizontal resolutions; however, there are some shortcomings when compared with observations. The final decades of



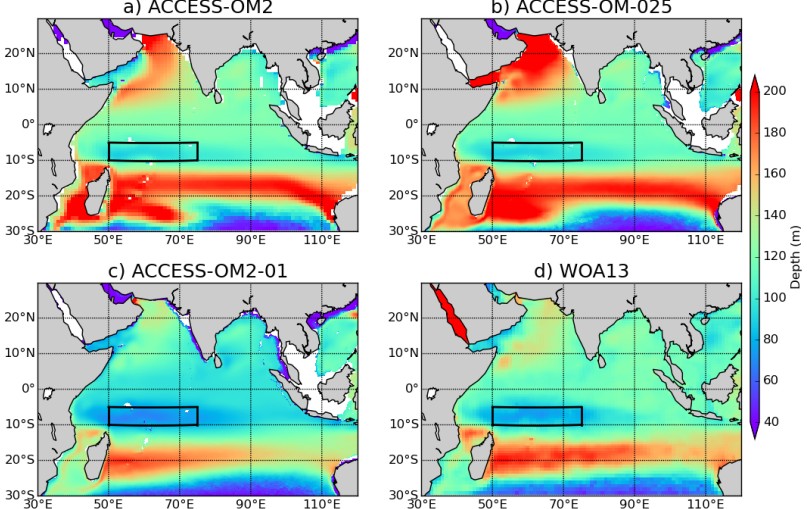

**Figure 26.** Annual mean depth of the 20°C isotherm (D20). (a) ACCESS-OM2; (b) ACCESS-OM2-025; (c) ACCESS-OM2-01 and (d) WOA13. The black box (50°E-75°E, 5°S-10°S) represents the position of the shallowest D20, which is used as a proxy for the thermocline ridge.

our simulations cover a period of dramatic changes in sea ice (Fig. 27a, b): the Arctic sea ice underwent a drastic decline of the annual minimum extent (e.g. Stroeve et al., 2014), whereas the Antarctic annual maximum sea ice extent ramped up from 2012 to the maximum extent on record in 2014 before decreasing sharply from 2015 to the current (2019) minimum (e.g. Turner and Comiso, 2017). The Arctic sea ice decline during winter has been linked to an anomalous atmospheric circulation pattern

bringing increased inflow of warm air masses from lower latitudes and a general polar warming, while during summer the positive feedback via absorption of short-wave radiation into and warming of the ocean mixed layer contributes to the loss of summer sea ice in the Arctic (e.g. Stroeve and Notz, 2018). The interannual variability in the Antarctic sea ice extent has been attributed to a combination of thermodynamics (likely driven by increased glacial melt; Bintanja et al., 2015) and dynamics (e.g., Holland and Kwok, 2012; Schlosser et al., 2018). These observed changes in sea ice extent are closely tracked by the

ACCESS-OM2 suite at all resolutions in both hemispheres (Fig. 27a, b) and are also reflected in sea ice volume (Fig. 27c, d). Like CORE-II (Large and Yeager, 2009), the JRA-55 reanalysis (Kobayashi et al., 2015) incorporates observed sea ice concentration; however JRA-55 treats regions with <55% ice concentration as ice-free, and regions exceeding this threshold as 100% sea ice (unlike CORE-II, which combines ocean and ice fluxes in proportion to their concentration). We speculate that this hard ice edge causes a stronger imprint of the observed sea ice in the JRA55-do atmospheric state (e.g., reducing the

10 m air temperature over ice), which then drives the modelled ice concentration to a state resembling the observations. We now assess the quality of the spatial distribution of sea ice in the two hemispheres.



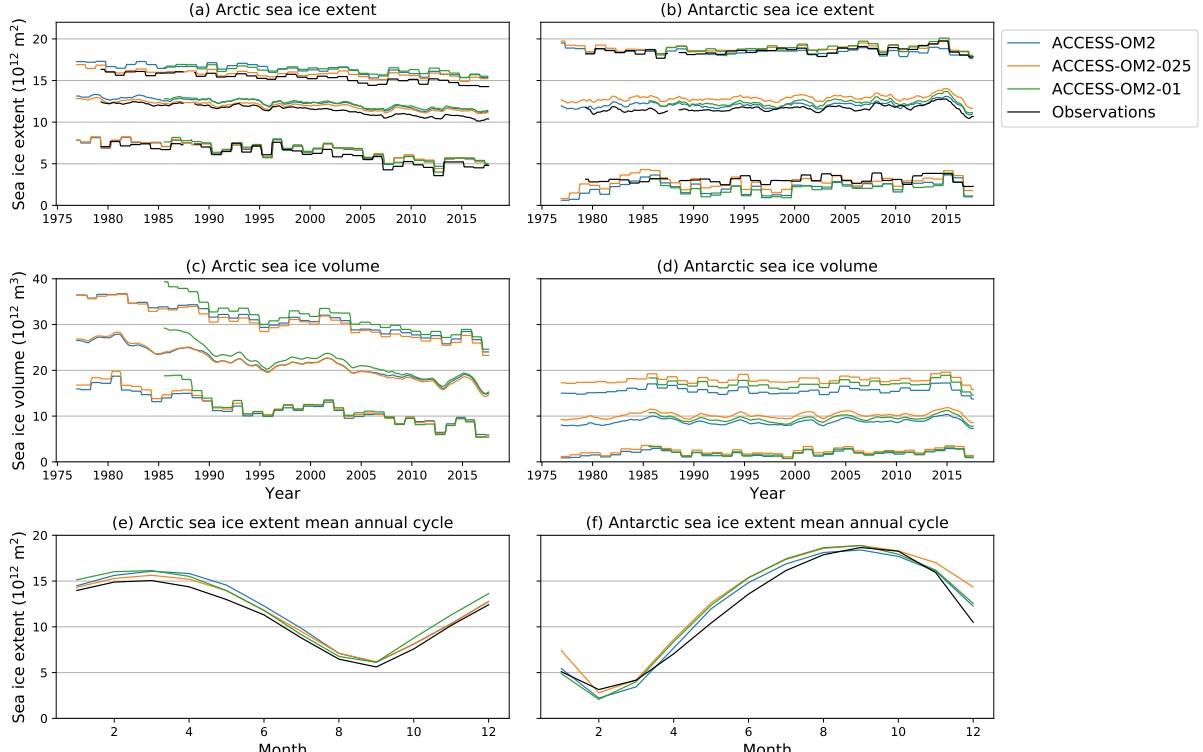

**Figure 27.** (a), (b) sea ice extent; (c), (d) volume and (e), (f) 1993–2017 mean annual cycle of extent in the (a), (c), (e) Arctic and (b), (d), (f) Antarctic in the three configurations. (a)–(d) show running 12-month minima, means and maxima. Observational estimates are based on passive microwave retrievals (NSIDC Sea Ice Index version 3, Fetterer et al., 2017, updated daily). Note that the three models and observations use different land masks, so the timeseries would not be expected to agree perfectly in magnitude.

### 4.3.1 Arctic Ocean

The Arctic sea ice biases in the ACCESS-OM suite appear considerably smaller than in most of the CORE-II-forced models investigated by Wang et al. (2016). The mean annual cycle of sea ice extent is close to the observed estimate (Fig. 27e) but seems to decline slightly too slowly in late spring. At all resolutions the simulated 1993–2017 monthly mean Arctic sea ice extent agrees well with observational estimates in most regions, as does the monthly mean concentration (contours and colour in Fig. 28b, c, e, f); however, some issues that warrant further investigation have been identified. At all resolutions the simulations exhibit a broad zone of sparse sea ice concentration in the eastern Sea of Okhotsk and southeast of Fram Strait at the March maximum, neither of which is observed (Fig. 28b, c). In ACCESS-OM2-01 the Canadian Archipelago, Central Arctic, and Siberian coast exhibit slightly excessive ice concentration during March, whereas in Baffin Bay simulated ice concentration is low compared to observations. These biases are also present at coarser resolution, although the Central Arctic bias is weaker (not shown). The ice extent in the Labrador Sea is also excessive in March at 1°. At the September minimum the ice extent



closely matches observations (apart from some excess coverage in eastern Siberia), but the overall concentration is slightly too low in ACCESS-OM2-01 (Fig. 28e, f) and in the coarser resolutions (not shown). In broad agreement with results from the Pan-Arctic Ice Ocean Model and Assimilation System (PIOMAS, Zhang and Rothrock, 2003), the thickest ice in our runs is found close to the Canadian Archipelago (Fig. 28a, d). However, rather than being transported northward through the Beaufort

Gyre and eventually out of the Central Arctic via the transpolar drift, much of the sea ice in our simulation remains within the Canadian Arctic. Investigation of the causes of these biases is beyond the scope of this paper, but possible contributing factors include SST biases (Fig. 10), issues regarding the modelled mixed-layer depth, and bias in the 0.1° initial condition.

Finally, we note that the sea ice in the 0.1° simulation displays many long, narrow, linear zones of low ice concentration and high strain rate (not shown) which open and close on timescales of days, largely in response to varying wind stress. These

lead-like linear kinematic features are too narrow to resolve with existing satellite passive microwave instruments (Lemieux et al., 2015) and are characteristic of high-resolution models with EVP rheology (e.g. Hutchings et al., 2005) as a result of strain localisation. While their spatiotemporal scaling may only be partially correct at this resolution (Hutter et al., 2018), we consider the presence of these lead-like features to be an improvement over the very smooth fields obtained in coarser EVP models. Linear kinematic features are also evident in the Antarctic ice, but are less ubiquitous than in the Arctic.

**4.3.2  Southern Hemisphere**

At all resolutions the modelled 1993–2017 mean annual cycle of Antarctic sea ice extent closely matches observational estimates, although ice growth appears to occur slightly more rapidly than observed (Fig. 27f). The spatial structure of the modelled 1993–2017 mean Antarctic spring maximum sea ice extent agrees well with that derived from passive microwave observations at all resolutions, and is particularly realistic in the 0.1° simulation (contours in Fig. 29b, c). During Antarctic winter and

early spring the simulated sea ice concentration is too high near the coast at 0.1°, and a fraction too low in the wider pack-ice zone (Fig. 29b, c); at coarser resolutions the concentration becomes smaller in both regions, reducing the positive bias near the coast but increasing the negative bias in the outer pack (not shown). We note that the high-concentration coastal ice in the model is very thin (Fig. 29a), with most of the ice cover in the thinnest thickness category (<0.64 m; not shown) and high frazil production (not shown), consistent with the presence of newly forming sea ice in coastal polynyas. Passive microwave

products are known to underestimate the concentration of thin ice, such as in polynyas or marginal ice zones during autumn and winter (e.g. Meier et al., 2014; Ivanova et al., 2015), suggesting that the discrepancy with the model may be partly due to a bias in the observational estimate.

The simulated annual minimum sea ice extent is much too low in the Weddell Sea and most of East Antarctica, and too high in the Ross Sea, at all three resolutions (contours in Fig. 29e, f). This is associated with low concentrations at all resolutions

in all regions other than the northern Ross Sea; this bias is also typical of a wide variety of models driven by CORE-II forcing (Downes et al., 2015). The thickness distribution also reveals broad regions of very low-concentration sea ice extending well beyond the model's 15% concentration contour (Fig. 29d). This ice is mostly in the thickest two categories (>2.47 m), i.e., second-year ice. Sea ice concentration builds up rapidly from its low minimum, reaching realistically high values in the outer pack by May-June, but the concentration then declines early relative to microwave observations, apparently preconditioning





**Figure 28.** 1993–2017 mean Arctic sea ice thickness (ice volume per unit area; a, d) and concentration (ice area per unit area; b, e) in ACCESS-OM2-01 and concentration estimates from NOAA G02202 V3 passive microwave Goddard merged monthly data (Peng et al., 2013; Meier et al., 2017, c, f) for (a)–(c) March and (d)–(f) September. The concentration scale is nonlinear to highlight differences at high concentrations. Contour lines show the sea ice extent (defined as the 15% concentration contour) in all three model configurations and observations.

the ice for an overly rapid melt rate towards its too-low minimum. Further investigation is needed to understand the interplay between the oceanic mixed layer, ice advection and thickness, and sea ice growth or melt processes, but is beyond the scope of this paper.



**Figure 29.** As for Fig. 28 but in the Antarctic for (a)–(c) September and (d)–(f) February.

## 5    Summary

The ACCESS-OM2 model suite is specifically designed as a model hierarchy, with supported configurations at three different horizontal resolutions. This feature of the model makes it ideal for studies investigating the sensitivity of solutions to model resolution, and thereby support improved models at all resolutions. However, it is advantageous for other reasons. For example, when building or testing new configurations, these tests can be first done at low resolution, confirming basic conservation properties before proceeding to more expensive, high resolution cases. It is also designed to lead to a convergence of operational uses of the model, with high resolution ocean prediction systems able to align with the code base used for coupled simulations.

This manuscript has outlined the development of this model, and in particular documented the new features of this model and the coupling between its components. It has also enabled a moderately thorough evaluation of ACCESS-OM2 at each



resolution, with the goal of informing which aspects of the model can be used for different research or operational objectives. This approach also has the benefit of providing a benchmark for future developments.

In general terms, the model does a good job of representing many features of the ocean, particularly at high resolution. Historical sea ice extent trends are well-represented, and the surface properties and transects in each ocean basin compare well

with the observational record. The large scale overturning circulation, flow through the Indonesian archipelago and patterns of boundary currents are generally realistic, supporting the notion that this suite of models is competitive with similar models from other institutions. Areas for improvement include the Gulf Stream behaviour and associated North Atlantic SST and SSS biases, the weaker than observed Drake Passage transport and the weak AMOC in the 1° configuration. In addition, more work is needed to understand differences between observed and modelled meridional heat transport.

One feature of the model evaluation exercise was to highlight a general improvement of many model metrics at higher resolution. In particular, Southern Ocean water masses, the Antarctic shelf region, the overturning circulation and the western boundary current regions are all much improved in ACCESS-OM2-01 compared with the coarser resolutions. However, the highest resolution model also has the weakest Antarctic Circumpolar Current transport, has an overly strong Equatorial Pacific thermocline and continues to have biases in western boundary current regions, despite the high resolution. These features will

continue to be investigated.

A feature of modern model development is the continuous and collaborative process of building software. A version of ACCESS-OM2 has been frozen to enable other users to replicate these simulations (see code availability below) but it will be continuously developed. These incremental improvements will be publicly available via the model GitHub site, allowing users to both adopt and contribute to the future evolution of the model.

*Code and data availability.* The ACCESS-OM2 source code and configurations used for the simulations in this paper are available from doi:10.5281/zenodo.2653246 (Hannah et al., 2019). The ACCESS-OM2 code is undergoing continuous development, and these developments (including the MOM5 and CICE5 distributions) are publicly available at https://github.com/COSIMA/access-om2. All model components (MOM5, CICE5, OASIS3-MCT, YAML and libaccessom2) are open-source. The model output for the simulations presented in this manuscript will be stored on the COSIMA data collection, available from doi:10.4225/41/5a2dc8543105a. The Ssalto/Duacs altimeter prod-

ucts were produced and distributed by the Copernicus Marine and Environment Monitoring Service (http://www.marine.copernicus.eu). The mean dynamic topography (MDT) altimeter products were produced by SSALTO/DUACS and distributed by AVISO+, with support from CNES (https://www.aviso.altimetry.fr). This study has been conducted using gridded sea-level anomalies obtained from E.U. Copernicus Marine and Environmental Services.

*Author contributions.* This manuscript is a broad collaboration between a number of institutions. The technical side of the project was led

by Hannah, with substantial contributions from Ward, Fiedler and Heerdegen, and many other contributions from the community. The model simulations were conducted primarily by Kiss, Hogg and Savita. Analysis of the output and the writing of text for the manuscript was a broad





effort, coordinated by Hogg and Kiss with substantial contributions from Boeira Dias, Chamberlain, Chapman, Duran, Heil, Holmes, Hogg, Kiss, Klocker, Pilo, Richet, Savita, Spence, Ward and Zhang.

*Competing interests.* The authors declare that they have no conflict of interest.

*Acknowledgements.* This manuscript is the first significant research publication from the Consortium for Ocean-Sea Ice Modelling in Australia (COSIMA; cosima.org.au) which is funded by the Australian Research Council through its Linkage Program (LP160100073). This research was supported by the ARC Centre of Excellence for Climate Extremes which is supported by the Australian Research Council via grant CE170100023. This research was undertaken with the assistance of resources from the National Computational Infrastructure (NCI), which is supported by the Australian Government. O. Richet was supported by the Centre for Southern Hemisphere Oceans Research (CSHOR). F. Boeira Dias, C. M. Domingues, P. Heil, A. Klocker S. J. Marsland and A. Savita were supported by the Australian Government's Cooperative Research Centres Programme through the Antarctic Climate and Ecosystems Cooperative Research Centre. The International Space Science Institute (Bern, Switzerland) is thanked for supporting scientific collaborations in this study through project 406. This work contributes to the Australian Antarctic Science projects 4301 and 4390. S. Marsland and P. Dobrohotoff are supported by the Earth Systems and Climate Change Hub of the Australian Government's National Environmental Science Program. C. M. Domingues was supported by the Australian Research Council (FT13101532 and DP160103130). A. K. Morrison was supported by an Australian Research Council DECRA Fellowship DE170100184. Our thanks to Alain Colin de Verdière, who provided barotropic streamfunction data for model comparisons.



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
