# Peer review of "ACCESS-OM2 v1.0: A Global Ocean-Sea Ice Model at Three Resolutions"

_Geoscientific Model Development, 2019_

## Referee Comment (RC1) · P. Hyder (Referee) · 6 Aug 2019

Review Comments on 'ACCESS-OM2: A Global Ocean-Sea Ice Model at Three Resolutions' – Geoscientific Model Development

Comments from Pat Hyder (Met Office)

I recommend publication with minor changes. This is an extremely thorough and generally well written documentation and assessment of the ACCESS ocean-only configurations at three ocean model resolutions. I have some general and minor comments, which I have included below.

General comments

1) The paper is very thorough and present a wealth of interesting results. As a result it is fairly long so I have not been able to comment on the wording of all the individual sentences. However, my feeling is that in quite a few places sentences are rather long and could usefully be split into two or more sentences to aid clarity. I also thought in several places the inferences seemed to me at least to be a bit speculative so could perhaps from a few more caveats.

2) This paper appears to suffer from the same frustrating difficulty in identifying clear benefits of ocean model resolution that all similar coupled and ocean-only studies have experienced. As previous studies, the only clear benefits are seen in metrics related to boundary and frontal currents and associated eddy variability. These difficulties are no surprise because both ocean forcing sets and coupled atmospheric models have large errors, giving rise for enormous potential for error cancellation between forcing set errors and changes due to ocean model resolution. This seems likely to be what is giving rise to a mixture of improvements and degradation that all studies of coupled and ocean-only simulations find. A few more points on model errors are included below:

- Errors are often no smaller in ocean forcing sets than unconstrained atmospheric models, particularly in radiative fluxes and near surface air temperature and relative humidity. This is because all of these parameters are virtually unconstrained by atmospheric data assimilation and therefore in reanalyses, such as JRA-55, they just reflect the underlying model atmospheric biases. The reason SST biases are smaller in ocean-only simulations is simply that the SST are far too strongly tied by surface fluxes to prescribed air temperatures from the re-analyses (e.g. Hyder et al 2018, Nature Communications), which are in turn tied to the observed SSTs used in the re-analyses.

- For ocean-only models the problem is compounded by the deficient surface boundary layer which assumes that the near surface atmosphere has large heat capacity compared to the underlying ocean which is clearly incorrect. Potentially this could give rise to errors, including substantial over-estimation of ocean convection. Also in the real world, coupled feedbacks of the ocean onto the atmosphere (and associated energetic constraints) are important so it is not clear that resolution-dependence can be adequately represented in ocean-only simulations. For example, we find the resolution dependence of SST biases, i.e. differences between configurations at different ocean resolutions are completely different in ocean-only and coupled simulations.

In my view, it might be good to set out these problems in the introduction and refer back to them in the conclusions?

3) Surprisingly, this study does not find the commonly-seen benefits in the Gulf stream northward turn representation at high ocean resolution, or the very cold bias in the this region at low resolution, e.g. see Storkey et al, 2018, fig 16 ( www.geosci-model-dev.net/11/3187/2018/gmd-11-3187-2018.pdf - it might also be worth referring to this ocean-only resolution study) or Hewitt et al 2016 or Griffies et al studies of GFDL coupled models. It would be great, if possible, to understand better why this is, perhaps the formulation of the slip coastal/topographic vorticity boundary condition and formulation of Coriolis, momentum advection or viscosity? Do the GFDL coupled (or ocean-only) simulations have similar issues in this region?

Minor Comments

All of the following suggested changes are optional from my perspective. There are also lots of comments and questions, which don't require changes or need including (unless you think anything is relevant). Feel free to e mail me if any of the comments are not clear.

**Abstract – Is it worth making it clearer in the first sentence that these are ocean-only simulations?**

**Page 2, line 7 – It might be useful to include a paragraph or two setting out the issues with ocean-only simulations including errors in the forcing set and surface boundary condition that hinder the identification of clear resolution dependent benefits in most studies (see general comments above)?**

**Page 2, line 8 – Is it worth changing it to read 'is currently 1o'**

**Page 4, line 8 – Since both vertical and horizontal resolution are changed, it is worth saying something along the lines of 'so we cannot distinguish between these changes in our results'?**

**Page 5 – Is it worth stating that bathymetric inconsistencies could contribute to resolution-related differences in performance, particularly near topography or coastlines?**

**Page 6 – Parameterisation section – As you mention later in the paper, it would be great, if possible, to better understand the rather unusual changes in representation of the Gulf Stream with resolution in your models. Your low resolution model does not have the common extremely cold bias in the northward turn region (which is up to 5 degrees in other models). However, you also do not appear to see the representation benefits with increasing resolution that are seen in other systems, in particular, much reduced cold biases in the northward turn region. Typically, Gulf Stream, and associated eddy variability, are influenced by viscosity, formulation of slip condition, GM if it's used, etc. One might expect the Gent-McWillliams parameterisation that you use to damp the Gulf Stream and eddies in the $\frac{1}{4}$ degree model, making it behave more like the low resolution model (this is why we haven't used it so far). We are also wondering if the high resolution model might be adversely affected by the no slip condition (we use free slip at all resolutions)?**

**Page 6 line 32 – Does 'as a consequence of the B grid' mean that one cannot formulate a free slip condition on a B grid?**

**Page 7 line 5 – Is there a more plain English way of say 'obviating', i.e. perhaps avoiding or removing?**

**Page 8 – Is it worth defining 'roundrobin'?**

**Page 8 line 32 – Is it really true that there is evidence that the JRA-55 forcing set is**

a major step forwards compared to CORE-II? From what I understand, all re-analyses have major issues, particularly with clouds and associated downward shortwave and longwave forcing and the near surface boundary layer including near-surface temperatures and humidities. These errors are not corrected for by the atmospheric data assimilation in the re-analyses and from what I have seen in met office assimilative models are largely unaffected by increased atmospheric resolution. The corrections one can apply to these deficiencies are rather crude, in part due to the very limited near surface observations, and often not physically-based so my suspicion is that errors in the JRA-55 forcing set are of a similar magnitude to those in CORE-II.

**Page 9 line 14 - Do you find that using relative winds rather than absolute winds over-damps eddies and EKE? We also use relative wind stress to be consistent with our coupled configurations but I think the DRAKKAR NEMO ocean-only community often to use absolute winds in ocean-only for this reason.**

**Page 9 line 19 – is there a reason to use the slightly higher surface albedo than CORE-II or is this tuning?**

**Page 9 line 28 – Is there not still substantial global mean salinity and sea-surface height drift due to the global mean imbalance between prescribed precipitation minus prognostic evaporation and the prescribed run off?**

**Page 10 line 20-22 – Is it worth clarifying this sentence?**

**Page 10 line 22-24 – We tend to start sea-ice off from the restart file of a separate spun up run to avoid contaminating the salinity with melt/formation as the ice-spins up. However, perhaps this does not matter too much when you run the models for such a long time?**

**Page 10 line 26 – Might it be useful to explain your different run lengths for the different models before you discuss the initial conditions?**

**Page 11 line 2 – Perhaps you could say 'coupled ocean-sea-ice model' to make it**

clearer that it's not a fully coupled model?

**Page 12 line 22 – Would it be useful to state the years per day for the medium resolution $\frac{1}{4}$ degree model?**

**Page 12 line 31 – Is it worth defining 'sectrobin' and 'roundrobin'?**

**Page 14 line 16 to Page 16 line 3 – Do you think it might be worth calling this section experimental strategy and moving it to just before you discuss the initial conditions? In my opinion, this would be a bit clearer.**

**Page 16 line 2-3 – Is it worth clarifying the last sentence in the paragraph?**

**Page 16 line 6-7 – I guess one might expect SST to be tightly coupled to forcing in ocean-only configurations since the hugely over-active surface flux response to SST changes ties the SST strongly to prescribed air temperature (e.g. see Hyder et al, 2018, Nat. Comms)?**

**Page 16 line 13 – Is it worth changing 'by the following analyses' to 'in the following sections'?**

**Page 16 line 23-25 - Is it worth clarifying these two sentences and adding a bit more explanation?**

**Page 16 line 26-29 – Is it worth clarifying this text, perhaps by splitting the sentence into two sentences?**

**Page 16 line 32 – Is it worth changing this text to something along the lines of 'but a shallower MDSL, i.e. larger errors compared to AVISO, in the ACCESS-OM2-01 case.'**
**Page 17 Figure 5 – This is a great figure but I am wondering if also plotting differences compared to the observational estimates might help to highlight the resolution dependence differences?**

**Page 18 line 1-4 - It might be useful to state the frequency of the data used to generate the standard deviation (i.e. are they daily data)?**

\# Page 18 line 19 – Is it worth adding 'as expected since the eddies are not resolved but parameterised in the low resolution configuration'?

\# Page 18 line 20 – Change to 'of larger sea-level variability...'

\# Page 19 line 15 and Page 20 line 3-4 – Can the dense shelf water really cascade down the slope accurately in the high resolution model since it still uses z-coordinate in which the bottom boundary layer is not at all well resolved (or is it perhaps also localised spurious but intermittent convection, perhaps in polynyas, that is making the bottom water)? In the UK ORCHESTRA project 1/12th degree sector model we find one need to use a sigma coordinate to get the dense water to cascade down slope, even at high resolution.

\# Page 21 line 12 – Is there perhaps also a contribution from numerical mixing?

\# Page 22 line 6 – Perhaps SSTs and trends might also be expected to be tied to any trends in air temperatures in the forcing set over the forcing period in an ocean-only model?

\# Page 23 line 9 comment – In the Gulf Stream region, your models seem to behave rather differently to other ocean-models. For example, your low resolution model doesn't seem to have the typical very large ($\sim$5 degree) cold bias seen in our low resolution model (Storkey et al, 2018) but also doesn't show the clear improvement in this region with resolution. This could be due to the experimental design and particularly the new JRA55 forcing or the long spin ups you use. However, as I discussed in the earlier comment on the parameterisation section, it could also be due to your configuration. In particular, we wonder if the $\frac{1}{4}$ degree model might be adversely affected by GM and all resolutions might be adversely affected by the no slip condition (see earlier comment for details).

\# Page 23 line 21 – Is it worth trying to explaining why you think it is due to the vertical diffusivity? From what I understand, the upwelling regions often get better, i.e. colder,

at higher resolution? However, there are often substantial cloud biases in upwelling regions in atmospheric models and hence ocean forcing sets, usually too much downward short wave – this would be consistent across resolutions but I guess potentially could give rise to error cancellation?

**Page 23 – line 33-35 –Does the z coordinate model really allow the dense water to cascade down the slope correctly (see earlier comment on this)?**

**Page 24 line 8-11 – Is it worth clarifying this sentence, perhaps by splitting it into two sentences?**

**Page 25 line 4 & fig 13 – This probably doesn't make much difference for your long averaging period but does the Trenberth and Carron method assume equilibrium, i.e. no heat content tendency.**

**Page 26 line 7 – To me this seems to me like quite a strong inference from the results?**

**Page 27 line 5 – Could you perhaps use 'comprehensive' instead of 'exhaustive'?**

**Page 27 line 8 – Would it be clearer to change this text to 'A significant motivation for moving..'?**

**Page 27 line 9 – Is it worth changing this to read 'mesoscale variability plays a critical dynamical role in the evolution. . .'?**

**Page 28 line 5 - Is it worth adding 'representation of surface boundary condition in ocean-only runs' as a possible factor? For example, the prescribed air temperatures effectively give air an infinite heat capacity compared to water, which is clearly wrong. In the real world or a coupled model, air temperature adjusts very rapidly towards SSTs because of the much smaller heat capacity of air compared to water.**

**Page 28 line 7-10 – Is it possible to clarify this sentence and the associated Fig 15? Perhaps you could also label the regions you describing with arrows on the plot as I could not really follow exactly what you were referring to?**

**Page 29 line 1-3 – Could this sentence be clarified?**

**Page 29 line 4 and Fig 16 (and all subsequent sea level STD plots) – Might be worth mentioning the frequency of the data used for the standard deviations?**

**Page 29 line 6 and Fig 17 – Could this suggest you need higher still resolution to represent the EAC?**

**Page 29 line 10-11 – Is it worth clarifying this text: 'indicator of the overall robustness of the model for this region'?**

**Page 32 final paragraph – Some of the inferences seems a bit speculative and could perhaps benefit from more caveats?**

**Page 34 line 3-8 – Is it worth adding a caveat to the final inference as it seems a bit speculative, to me at least?**

**Page 36 line 3 and Fig 22 – See earlier comments on representation of the Gulf Stream and its dependence on configuration (the viscosity and slip condition in particular).**

**Page 36 line 13 – Perhaps you could say 'the loop current in the Gulf of Mexico'?**

**Page 37 Figure 22 – Again the Gulf Stream eddy variability improvement with resolution does not seem as good as often seen, e.g. by GFDL in their coupled models?**

**Page 39 line 1 and Fig 25 - From the figure it appears that the biases are much reduced in the high resolution model? This did not come over clearly to me from this paragraph so could the text be clarified?**

**Page 40 line 7 – Is it worth referring to Fig 27 in first sentence and changing the text to include something along the lines of 'acceptable and very similar sea-ice extent and concentration at all three resolutions? I guess one might expect this because the sea ice is tied to the forcing fields in ocean-only runs by dramatic change in air temperature over ice and non-ice regions in the re-analysis, which are in turn tied to the observed**

[Figure]

sea-ice it uses as a lower boundary condition.

**Page 44 line 2 – Perhaps this text could be changed to 'ocean mixed layer and associated SST and SST biases'?**

**Page 45 line 1 – Summary - It is worth starting by referring back to the introduction and in both sections summarising the pros and cons of ocean-only versus coupled, problems with forcing and surface boundary condition errors and substantial potential for error cancellation which makes it hard to robustly identify resolution related benefits.**

**Page 45 line 1 – We use the term 'traceable' to refer to minimising configuration changes between resolutions.**

**Page 46 – line 3 – Again, is it worth stating that with the exception of the boundary currents and strong eddying regions, where resolution related benefits appear to be clear, one sees a mixture of improvements and degradation. This might be expected from the substantial errors in ocean forcing sets and the surface boundary condition, which provide a lot of potential for error cancellation.**

---

## Referee Comment (RC2) · Anonymous Referee #2 · 11 Oct 2019

This is a thorough description of the performance of the ocean and sea-ice component of the Australian Community Climate and Earth Simulator (ACCESS-OM2). In my view this paper will be a really useful reference for anyone working with ACCESS but also for anyone looking for a good reference to illustrate the impact of model resolution on ocean simulations. The paper provides a nice overview of key ocean circulation features at horizontal resolutions of 1, 0.25 and 0.1 degrees. Even though it is well known that the circulation changes with resolution there are (to my knowledge) not too many examples of papers showing a systematic comparison of the global ocean circulation at non-eddying, eddy-permitting and eddy-rich resolutions.

[Figure]

The paper itself is clear and well written and perfectly fits the scope of Geoscientific Model Development. I strongly recommend publication subject to some clarifications of the minor points listed below.

Comments:

1) Page 4, line 5: I was a bit surprised at the choice of 50 vertical levels for the ACCESS-OM2 and ACCESS-OM2-025 rather than 75 levels. Given that most HPC time is eaten up by ACCESS-OM2-01 the amount of time saved seems minimal. This seems to go against the philosophy outlined earlier, namely to keep the three resolutions as similar as possible. It would be worth explaining a bit more why this choice was made (e.g. 50 levels are sufficient at 1 and 0.25 degree as suggested in Stewart et al. 2017).

2) Page 6, line 10: Perhaps it would be worth noting why the choice of 0.004s-1 was made for the buoyancy frequency. Is this a typical value for this depth range?

3) Page 9, line 5: "downwelling" → "downward"

4) Page 9, line 5: I suppose the river run-off is essentially climatological -especially during the first part of the forcing?

5) Page 9, lines 29-31: I am not sure I really understand what is being done here. This could be read as if salinities were being nudged to WOA +/- 0.5 psu whenever, restoring fluxes try to push SSS outside that range. Obviously this is not what is being done since in Figure 11 there are regions where the salinity mismatch exceeds 0.5 psu (e.g. in ACCESS-OM2-01 off Grand Banks, in the Arctic and at the entry into the Caribbean Sea). This needs to be explained a bit more carefully.

6) Page 10, lines 26-27: Is there any particular reason why the period from May 1984 – to April 1985 is chosen? Was this a year that was reasonably neutral for most major indices e.g. ENSO...etc i.e. "normal" year-ish"?

7) Page 10, line 33: Do you mean that the Kuroshio separation was too far north? Also

[Figure]

I suppose this refers to the 0.25 and 0.1deg versions as WBCs are so diffuse at 1 deg?

8) Page 12, line 32: Is the timestep for ACCESS-OM2-01 400 or 450s? (The latter number is given in table 2).

9) Figure 3: To me it seems that for the globally averaged SST (panel b) the last pass looks different from passes 1-4. In passes 1-4 the globally averaged SST varies between about 17.8C to 18.4C and looks very similar for all passes. However, in pass 5 the SSTs vary between about 18 and 18.4C. This seems quite a large difference for a global average. It is noticeable that ACCESS-OM2-01 starts from much colder conditions (panel c). this linked to the stronger MOC in ACCESS-OM2-01? At the end of the spinup the globally averaged temperature is about 0.2 deg colder than for the lower resolutions. Again, for a globally averaged value this is quite a big difference.

10) Page 16, line 5: The initial cold drift cannot be seen in Figure 3a.

11) Page 16, lines 23-25: It is interesting that a large ACC variability is only really seen in the first pass for 1 and 0.25 degrees where there is a pronounced and broad peak in transport during the first pass which is not seen at 0.1 deg. Is there a spike in the AABW formation during the first pass in ACCESS-OM2/-025?

13) Page 18, lines 16-18: There are clear differences for e.g. the Gulf Stream at 0.1 deg. The SSH variability suggests that Gulf Stream path may be too variable just after separating from Cape Hatteras. The SSH variability is confined to a broad patch North of Cape Hatteras rather than extending further east along the extension as suggested by AVISO.

14) Page 18, lines 25-28: I don't think that this can really be inferred from Figure 6...

15) Page 18, line 30, Figure 7, Page 19, line 10: It would be nice to plot the overturning for the full range of densities i.e. not to cut off the lightest densities . I'd also suggest to expand the higher densities e.g. between 36.5 and 37.5 as this would show the AABW cell more clearly. This I feel could be relevant to understand differences in the ACC

strength between the different resolutions (see comment 21) below.

16) Page 22, Figure 9: Is the data temporally filtered for the 1 and 0.25 degree resolutions? A seasonal signal is clearly visible in the surface layers of the 0.1 degree model but not at 1 and 0.25 degrees.

17) Pages 23, lines 3-4, Figure 10: For some regions the anomaly patterns look quite different in ACCESS-OM2-01. For example in the Northern North Atlantic there is a large-scale warm bias and a strengthening cold bias in the southern SPG whereas the rest of the northern North Atlantic is warmer in ACCESS-OM2-01 than in the other cases.

18) Page 26, lines 7-8: To me the picture does not always seem as clear cut here. Between 0-30S ACCESS-OM2 has the best agreement with Ganachaud & Wunsch.

19) Page 27, line 15: Define all terms for the PV equation.

20) Page 29, Figure 15: I suppose the maximum/minimum mixed layer depths are from September (max) and March (min)?

21) Page 34, Figure 20: I can see that there seems to be a problem with the Antarctic mode water at the lower resolutions. However, what is equally pronounced in my view is the cold bias seen south of about 60S. In addition there is also a weaker cold bias at depth extending from the high southern latitudes to the northern end of the domain for the 1 and 0.25deg resolution versions of the model. This is also seen for the latitude-depth sections through the Atlantic and Indian Oceans (Figures 23 and 25) as well as for the zonally averaged temperatures shown in Figure 12. Could this explain why the ACC transport is weaker in ACCESS-OM2-01 than in ACCESS-OM2? The cold bias around Antarctica increases the meridional density gradient across the ACC which may explain the higher ACC transport in ACCESS-OM2 (where the cold bias is strongest). The coldest bias around Antarctica in ACCESS-OM2 may seem at odds with the overturning cell associated with AABW formation shown which is weaker than

for the higher resolutions (Figure 7). However, my impression is that although weaker the overturning associated with AABW involves higher densities at 1 deg than at 0.25 and 0.1 deg. This would come out more clearly if the overturning is expanded for higher densities in Figure 7 (see comment 15).

22) Page 36, lines 8-9: Note that there are also uncertainties in the observational estimate of the barotropic streamfunction of DeVerdiere & Ollitrault. So I suppose that rather small features of the barotropic streamfunction such as this recirculation have to be taken with care.

23) Page 42, lines 3-4, Figure 27: The sea ice decline only seems too slow for ACCESS-OM2-025. For 1 and 0.1 deg there is a small positive bias compared to the observations that remains almost unchanged during the simulations but the long-term sea-ice decline looks very similar.

24) Caption Figure 16 (and other Figure captions): I suggest to replace "overlain" with "overlaid".

---

## Author Comment (AC1) · 16 Nov 2019

We thank the reviewer for their thorough review and insightful comments. Our responses are given in the attached PDF.

Please also note the supplement to this comment:
https://www.geosci-model-dev-discuss.net/gmd-2019-106/gmd-2019-106-AC1-supplement.pdf

---

## Author Response (AR1)

**Response to review comments on gmd-2019-106 'ACCESS-OM2 v1.0: A Global Ocean-Sea Ice Model at Three Resolutions' - Geoscientific Model Development**

Referee comments are in bold, author responses are in plain text, and changes in the revised manuscript are in italic.

**Comments from Referee #1 (Pat Hyder, Met Office)**

**I recommend publication with minor changes. This is an extremely thorough and generally well written documentation and assessment of the ACCESS ocean-only configurations at three ocean model resolutions. I have some general and minor comments, which I have included below.**

Our thanks to the reviewer for their thoughtful and comprehensive review. We have worked to take on board almost all of their comments, and we believe this has resulted in a significantly improved manuscript.

**General comments**

**1) The paper is very thorough and present a wealth of interesting results. As a result it is fairly long so I have not been able to comment on the wording of all the individual sentences. However, my feeling is that in quite a few places sentences are rather long and could usefully be split into two or more sentences to aid clarity. I also thought in several places the inferences seemed to me at least to be a bit speculative so could perhaps from a few more caveats.**
We have made most of the suggested changes to sentence length and caveats (listed in detail below).

**2) This paper appears to suffer from the same frustrating difficulty in identifying clear benefits of ocean model resolution that all similar coupled and ocean-only studies have experienced. As previous studies, the only clear benefits are seen in metrics related to boundary and frontal currents and associated eddy variability. These difficulties are no surprise because both ocean forcing sets and coupled atmospheric models have large errors, giving rise for enormous potential for error cancellation between forcing set errors and changes due to ocean model resolution. This seems likely to be what is giving rise to a mixture of improvements and degradation that all studies of coupled and ocean-only simulations find. A few more points on model errors are included below:**
We accept that the JRA55-do forcing dataset contains biases, and that resolution-dependent improvements or deteriorations may be due (at least in part) to changes in how well the ocean-sea ice model biases cancel the forcing biases, i.e. we may get an improved result for the wrong reasons. However, it is not clear to us that this potential error cancellation "seems likely" to explain the mix of improvement and degradation. We consider the extent to which error cancellation and resolved model dynamics govern model biases to be an important open question that will require more investigation than can fit in the scope of the present paper. We have instead added the following sentence to the concluding summary:
*It remains an open question to what extent these resolution-dependent changes are related to changes in model dynamics or to differing error cancellation with forcing biases.*

**- Errors are often no smaller in ocean forcing sets than unconstrained atmospheric models, particularly in radiative fluxes and near surface air temperature and relative humidity. This is because all of these parameters are virtually unconstrained by atmospheric data assimilation and therefore in reanalyses, such as JRA-55, they just reflect the underlying model atmospheric biases. The reason SST biases are smaller in ocean-only simulations is simply that the SST are far too strongly tied by surface fluxes to prescribed air temperatures from the re-analyses (e.g. Hyder et al 2018, Nature Communications), which are in turn tied to the observed SSTs used in the re-analyses.**

Re. "Errors are often no smaller in ocean forcing sets than unconstrained atmospheric models, particularly in radiative fluxes and near surface air temperature and relative humidity." - it is important to note that we use JRA55-do, not the JRA55 reanalysis. JRA55-do includes many adjustments of the raw JRA55 reanalysis data specifically to minimise biases in the forcing fields; indeed Tsujino et al. (2018) show that the near surface air temperature and humidity biases in JRA55-do are smaller than for the raw JRA55 reanalysis (see their figures 42-45).

We have commented on the SST being too strongly coupled to the atmosphere in sections 2.3 and 4 (see below).

**- For ocean-only models the problem is compounded by the deficient surface boundary layer which assumes that the near surface atmosphere has large heat capacity compared to the underlying ocean which is clearly incorrect. Potentially this could give rise to errors, including substantial over-estimation of ocean convection. Also in the real world, coupled feedbacks of the ocean onto the atmosphere (and associated energetic constraints) are important so it is not clear that resolution-dependence can be adequately represented in ocean-only simulations. For example, we find the resolution dependence of SST biases, i.e. differences between configurations at different ocean resolutions are completely different in ocean-only and coupled simulations.**

We thank the reviewer for making this point. We have added this text to the introduction:

*Driving a model suite from a common prescribed atmosphere enables a relatively clean assessment of resolution dependence; however, such models will be more strongly driven by the atmosphere than coupled ocean-atmosphere models since they lack negative feedback of*
*sea surface temperature and currents onto the atmosphere (e.g. Hyder et al., 2018; Renault et al., 2016).*
and this text to section 2.3:

*We also note that the lack of coupled negative feedback with a dynamic atmosphere may produce (for example) overly large heat and momentum fluxes in response to the imposed JRA55-do forcing fields (Hyder et al., 2018; Renault et al., 2016).*
And this text to section 4:

*On the other hand, the global average Sea Surface Temperature (SST) in the models is dominated by the forcing field (as expected due to the lack of ocean-atmosphere feedback; see Hyder et al., 2018), with only weak variations between each cycle (Fig. 3b).*

**In my view, it might be good to set out these problems in the introduction and refer back to them in the conclusions?**

Done - see above.

**3) Surprisingly, this study does not find the commonly-seen benefits in the Gulf stream northward turn representation at high ocean resolution, or the very cold bias in the this region at low resolution, e.g. see Storkey et al, 2018, fig 16 ( www.geosci-model-dev.net/11/3187/2018/gmd-11-3187-2018.pdf - it might also be worth referring to this ocean-only resolution study) or Hewitt et al 2016 or Griffies et al studies of GFDL coupled models. It would be great, if possible, to understand better why this is, perhaps the formulation of the slip coastal/topographic vorticity boundary condition and formulation of Coriolis, momentum advection or viscosity? Do the GFDL coupled (or ocean-only) simulations have similar issues in this region?**

Although the Gulf Stream path improves at high resolution in some models (e.g. Storkey et al, 2018; Hewitt et al., 2016), this is not always the case (e.g. the GFDL coupled climate model experiments with a MOM ocean reported in Griffies et al., 2015).

The causes underlying Gulf Stream misbehaviour in our model are currently under investigation, and we thank the reviewer for these suggestions, but the dynamics of this region are complex (Chassignet and

Marshall, 2008) and it is beyond the scope of the present model description paper to follow up these suggestions with the detail required. We find that the Gulf Stream separation latitude is quite realistic in the 0.1 degree model for the first ~20 years, before shifting several degrees too far north. Thus there is nothing in the model formulation (no-slip boundary condition, Coriolis, momentum advection or viscosity, etc) which precludes more realistic separation (at least transiently), but the slow timescale on which the separation changes makes it difficult to investigate this issue with parameter sensitivity experiments given the expense of running at this high resolution.

The ACCESS-OM2 ocean model configurations use a no-slip velocity condition against land points because this is the only condition supported by MOM5 on a B-grid. Although the no-slip condition makes a crucial difference to the initial separation of western boundary currents in idealised models with vertical sides (e.g. Kiss, 2002, JMR, https://doi.org/10.1357/002224002321505138), its importance in more realistic models is not so clear, particularly with respect to the later trajectory around the Grand Banks.  Bryan et al. (2007) obtained realistic Gulf Stream separation and North Atlantic Current path around the Grand Banks in POP, which uses a no-slip boundary condition (although this also required suitably low viscosity). And as noted above, we obtain quite realistic Gulf Stream separation with no-slip for the initial ~20 years of ACCESS-OM2-01.

Although we cannot investigate this issue thoroughly, we have made several changes and additions to provide a clearer picture of the model behaviour in this regard.

We now provide more detail in section 2.1.4 on the level of viscosity in these configurations:
*The biharmonic viscosity varies in space and time; at the surface in western boundary currents it is of order $10^{14}$, $10^{12}$ and $10^{10}$ m4s−1 in ACCESS-OM2, ACCESS-OM2-025 and ACCESS-OM2-01 (respectively), corresponding to viscous western boundary current widths (Haidvogel et al., 1992) of about 350, 100 and 60 km (respectively), which are well-resolved by the grid in all cases.*

We have also rewritten part of section 4.2.4 to better describe the Gulf Stream's separation and subsequent path and compare this with Storkey el al. (2018), Bryan et al., 2007 and  Griffies et al. (2015):
*Accurately representing the horizontal circulation of the North Atlantic is a persistent challenge for ocean modellers. In particular, models commonly fail to simulate a Gulf Stream that separates from the North American coast at Cape Hatteras (35° N, 75° W), and a North Atlantic Current that flows north along the east side of the Grand Banks from 40° to 51° N (Rossby, 1996; Chassignet and Marshall, 2008). In ACCESS-OM2 and ACCESS-OM2-025, the Gulf Stream flow is too weak and overshoots the separation latitude by about 4° relative to observations (Fig. 22a,b); in ACCESS-OM2-025 the separation latitude is also highly variable. The Gulf Stream structure in ACCESS-OM2-01 appears much closer to observations (Fig. 22c); however, the situation is more complex than this time-mean suggests. Between 1985 and about 2008 the ACCESS-OM2-01 Gulf Stream generally separates at or slightly north of Cape Hatteras, but the path of the separated current gradually changes from eastward to northeastward (not shown). After about 2008 the Gulf Stream adopts a configuration similar to ACCESS-OM2-025 (Fig. 22b), usually separating about 4° too far north, with a compact anticyclonic recirculation immediately south of the separated current (not shown). This long-term shift appears as excessive SSH variability shortly after separation and broadening of the separated jet in the long-term mean relative to observations (Fig. 22c,d). None of these models adequately captures the north-ward flow of the North Atlantic Current along the east side of the Grand Banks, resulting in significant cold and fresh biases in the sea surface temperature and salinity (Figs. 10 and 11). These biases are similar to those of Griffies et al. (2015) but contrast with other studies (e.g. Bryan et al., 2007; Storkey et al., 2018) which found significant improvement in the Gulf Stream's separation and path around the Grand Banks at high resolution (although Bryan et al., 2007, found this also required low viscosity). The cause of Gulf Stream misbehaviour in the ACCESS-OM2 models is currently*

*under investigation. Among the possible culprits are a weak deep western boundary current (Zhang et al., 2011), or low inertia (Özgökmen et al., 1997).*

**Minor Comments**

**All of the following suggested changes are optional from my perspective. There are also lots of comments and questions, which don't require changes or need including (unless you think anything is relevant). Feel free to e mail me if any of the comments are not clear.**

**# Abstract - Is it worth making it clearer in the first sentence that these are ocean-only simulations?**
Done. The first 2 sentences now read
*We introduce ACCESS-OM2, a new version of the ocean-sea ice model of the Australian Community Climate and Earth System Simulator. ACCESS-OM2 is driven by a prescribed atmosphere (JRA55-do) but has been designed to form the ocean-sea ice component of the fully coupled (atmosphere-land-ocean-sea ice) ACCESS-CM2 model.*

**# Page 2, line 7 - It might be useful to include a paragraph or two setting out the issues with ocean-only simulations including errors in the forcing set and surface boundary condition that hinder the identification of clear resolution dependent benefits in most studies (see general comments above)?**
We have mentioned these issues in new text in the introduction (3rd paragraph), section 2.3 and the final summary - see major point 2 above.

**# Page 2, line 8 - Is it worth changing it to read 'is currently 1o'**
Done.

**# Page 4, line 8 - Since both vertical and horizontal resolution are changed, it is worth saying something along the lines of 'so we cannot distinguish between these changes in our results'?**
Done. Added text reads:
*Since we refine both vertical and horizontal resolution in ACCESS-OM2-01, we cannot distinguish their effects in our results.*

**# Page 5 - Is it worth stating that bathymetric inconsistencies could contribute to resolution-related differences in performance, particularly near topography or coastlines?**
Done. The text now includes:
*These bathymetric and land mask inconsistencies may contribute to the differences in model behaviour at different resolutions.*

**# Page 6 - Parameterisation section - As you mention later in the paper, it would be great, if possible, to better understand the rather unusual changes in representation of the Gulf Stream with resolution in your models. Your low resolution model does not have the common extremely cold bias in the northward turn region (which is up to 5 degrees in other models). However, you also do not appear to see the representation benefits with increasing resolution that are seen in other systems, in particular, much reduced cold biases in the northward turn region. Typically, Gulf Stream, and associated eddy variability, are influenced by viscosity, formulation of slip condition, GM if it's used, etc. One might expect the Gent-McWilliams parameterisation that you use to damp the Gulf Stream and eddies in the 1/4 degree model, making it behave more like the low resolution model (this is why we haven't used it so far). We are also wondering if the high resolution model might be adversely affected by the no slip condition (we use free slip at all resolutions)?**

As noted above (major point 3), not all studies of this type demonstrate an improvement in Gulf Stream path with increasing resolution, and the no-slip condition does not preclude realistic Gulf Stream separation. We have investigated the effect of different Gent-McWilliams and Redi coefficients on ¼ degree model behaviour, but the results were mixed, and the configuration used here appeared to be the best compromise in most respects.

**Page 6 line 32 - Does 'as a consequence of the B grid' mean that one cannot formulate a free slip condition on a B grid?**

MOM5 does not support anything other than no-slip on a B grid. From Griffies "Elements of the Modular Ocean Model" (2012), section 25.6.1: "There are very special cases where one may specify a free-slip boundary, but these cases make unrealistic assumptions about the land-sea masking, such as for a zonally periodic channel. MOM does not make these assumptions, which means that all B-grid simulations with MOM utilize a no-slip side boundary condition."
Changed to
*The lateral boundary condition for velocity is no-slip, which is the only boundary condition supported by MOM5 on a B-grid (Griffies 2012).*

**Page 7 line 5 - Is there a more plain English way of say 'obviating', i.e. perhaps avoiding or removing?**

Done. Text now says *removing*.

**Page 8 - Is it worth defining 'roundrobin'?**

Not done - this is a minor technical point and is difficult to explain briefly. Instead we have provided a reference which explains these details.

**Page 8 line 32 - Is it really true that there is evidence that the JRA-55 forcing set is a major step forwards compared to CORE-II? From what I understand, all re-analyses have major issues, particularly with clouds and associated downward shortwave and longwave forcing and the near surface boundary layer including near-surface temperatures and humidities. These errors are not corrected for by the atmospheric data assimilation in the re-analyses and from what I have seen in met office assimilative models are largely unaffected by increased atmospheric resolution. The corrections one can apply to these deficiencies are rather crude, in part due to the very limited near surface observations, and often not physically-based so my suspicion is that errors in the JRA-55 forcing set are of a similar magnitude to those in CORE-II.**

While all such datasets have deficiencies, the biases in JRA55-do are smaller than those in CORE for many fields. Tsujino et al (2018) provide many comparisons of JRA55-do and CORE to observations. In comparison to buoy data (primarily in the tropical Pacific), JRA55-do has a smaller root-mean-square error and a higher correlation coefficient than CORE for 10m air temperature, 10m humidity, and wind speed (their Figs 43, 45, 47). JRA55-do wind stress and speed are generally closer than CORE to SCOW (their Figs 28, 29, 12, 46, 47). JRA55-do surface air temperature bias is smaller than CORE (their Figs 42, 43). However the JRA55-do humidity bias is larger in JRA55-do than in CORE (their Figs 44, 45). JRA55-do has more realistic Greenland runoff (an order of magnitude larger than in CORE) and spatially-varying Antarctic runoff. JRA55-do also has improved spatial and temporal resolution, time coverage, and dynamical self-consistency compared to CORE.
These sentences have been changed to
*The ACCESS-OM2 configurations are forced with the JRA55-do v1.3 atmospheric product (Tsujino et al., 2018) which has improved spatial and temporal resolution (55 km, 3-hourly) and temporal extent (1958--2018) compared to the Large and Yeager (2009) CORE-II dataset (200 km, 6-hourly, 1948--2009) used in many previous modelling studies. JRA55-do is more dynamically self-consistent than CORE-II and has smaller biases in surface wind and temperature (figures 12, 28, 29, 42, 43, 46, 47 in Tsujino et al.,*

*2018) but a larger bias in specific humidity (figures 44, 45 in Tsujino et al., 2018). The improved spatial resolution of wind is important for better representation of coastal polynyas (Stössel et al., 2011; Zhang et al., 2015) and upwelling (Taboada et al., 2019). JRA55-do has more realistic Greenland runoff (an order of magnitude larger than in CORE-II) and includes recent Antarctic calving and basal melt estimates from Depoorter et al. (2013), which are spatially variable and somewhat larger than the uniform values in CORE-II (Tsujino et al., 2018).*

**Page 9 line 14 - Do you find that using relative winds rather than absolute winds over-damps eddies and EKE? We also use relative wind stress to be consistent with our coupled configurations but I think the DRAKKAR NEMO ocean-only community often to use absolute winds in ocean-only for this reason.**

The question of how best to apply surface stress to eddying ocean-only models is complex and the subject of ongoing debate. In simplified terms, if the JRA55-do dataset supplied absolute wind, then it would be more appropriate to use relative rather than absolute winds to calculate the stress, since this will represent the eddy-killing effect of atmospheric drag (although eddy-killing will be overestimated due to the omission of stress and thermal feedback onto the atmosphere, e.g. Renault et al., JPO 2016). However, JRA55-do has been adjusted so that its time-mean matches the time-mean scatterometer wind (Tsujino et al., 2018), which is relative to the ocean, and so includes the signal from mean currents. As such, calculating stress using relative wind doubly-subtracts the mean currents (to the extent that model mean currents match the mean currents present during scatterometer measurements), whereas absolute winds fail to capture the atmospheric drag on eddies, or on mean model currents that differ from those present during scatterometer measurements. It is unclear how best to approach these issues.

Due to computational constraints we have only been able to investigate this question at 0.25deg. At this resolution we found that the RMS of SSH was reduced by up to 30% when relative winds were used. Presumably the effect would be greater at higher resolution.

Given that this is a complex issue which we have not fully resolved, we have not included a comment in the revised manuscript.

**Page 9 line 19 - is there a reason to use the slightly higher surface albedo than CORE-II or is this tuning?**

This was actually an oversight, which we document in this paper for reference. Large & Yeager (2009) CORE-II meridionally-varying albedo will be used in updated configurations.

**Page 9 line 28 - Is there not still substantial global mean salinity and sea-surface height drift due to the global mean imbalance between prescribed precipitation minus prognostic evaporation and the prescribed run off?**

We impose a constraint of zero net water flux with the atmosphere by applying an offset to P-E so that P-E+R=0 at each timestep. Over a 60-year forcing cycle the global mean salinity varies by about 0.0002 psu and the global mean SSH varies by about 3.5 cm (see figures below), but this is due to mass exchange with the sea ice, not an imbalance of P-E+R.

We have added:

*We impose a constraint of zero net water flux into the ocean from the coupler, by removing the area-mean of precipitation P minus evaporation E plus runoff R from P-E so that the integrated P-E+R is zero at each timestep; this does not constrain water exchanges between the ocean and sea ice.*

[Figure]

**Page 10 line 20-22 - Is it worth clarifying this sentence?**

Done - this long sentence has been broken into two:

*The ACCESS-OM2 and ACCESS-OM2-025 experiments were run for five 60-year cycles (1 Jan 1958 -- 31 Dec 2017) of JRA55-do.*

*The ocean was initially at rest, with zero sea level and with temperature and salinity from the World Ocean Atlas 2013 v2 (Locarnini et al., 2013; Zweng et al., 2013) 0.25° ``decav'' product (the average of six decadal averages spanning 1955--2012).*

**Page 10 line 22-24 - We tend to start sea-ice off from the restart file of a separate spun up run to avoid contaminating the salinity with melt/formation as the ice-spins up. However, perhaps this does not matter too much when you run the models for such a long time?**

The sea-ice initial condition is described in the paper. Importantly, it is not an ice-free condition.

The total sea ice volume in this initial condition is close enough to the adjusted state that there is no significant drift in total ocean salt mass as the sea ice spins up (see figures for ACCESS-OM2-025 below; other resolutions are similar). We have added this clarifying text:

*The total sea ice and snow volumes in this initial condition are very close to the adjusted state;*

*there is therefore no significant drift in total ocean salt as the sea ice spins up.*

[Figure]

[Figure]

Top two plots: Arctic and Antarctic sea ice volume by category in initial spinup at 0.25deg.
Next two plots: Arctic and Antarctic snow volume in initial spinup at 0.25deg.
Bottom row: monthly mean total mass of ocean salt in the first several years at 0.25deg.
The top four plots show monthly means and 12-month rolling means.
The ocean salt plot has a large offset indicated at the top of the y-axis. The annual cycles are due to sea ice formation and melting, which vary the total ocean salt by less than one part in 10^6.

**Page 10 line 26 - Might it be useful to explain your different run lengths for the different models before you discuss the initial conditions?**

The differing run lengths were already stated on lines 22 and 26. We have now clarified the first of these, breaking it into two sentences with the first stating

*The ACCESS-OM2 and ACCESS-OM2-025 experiments were run for five 60-year cycles (1 Jan 1958 -- 31 Dec 2017) of JRA55-do.*

**Page 11 line 2 - Perhaps you could say 'coupled ocean-sea-ice model' to make it clearer that it's not a fully coupled model?**

The word "coupled" has been removed. The sentence now reads:
*Figure 1 shows the fields that are transferred between the model components.*

**Page 12 line 22 - Would it be useful to state the years per day for the medium resolution 1/4 degree model?**

Done. Sentence now reads
*While higher resolutions require considerably more computational time, MOM5 still scales outstandingly well -- to over 2800 cores for ACCESS-OM2-025 (achieving 46 minutes per model year, i.e. over 30 model years per day) and up to 16,000 cores for ACCESS-OM2-01 -- and runtimes can be sustained when provided with a sufficient number of CPUs.*

**Page 12 line 31 - Is it worth defining 'sectrobin' and 'roundrobin'?**

Not done - this is a minor technical point and is difficult to explain briefly. Instead we have provided a reference which explains these details.

**# Page 14 line 16 to Page 16 line 3 - Do you think it might be worth calling this section experimental strategy and moving it to just before you discuss the initial conditions? In my opinion, this would be a bit clearer.**
We have retained the present structure as we want to keep the model results in section 4 (e.g. Fig 3) separate from the model configuration in section 2, and we consider the current structure to be sufficiently clear. However, we have added this to the start of section 2.4:
*The experiments discussed in Sect. 4 ran for different lengths of time (Fig. 3).*

**# Page 16 line 2-3 - Is it worth clarifying the last sentence in the paragraph?**
The sentence was removed, as this point was already implicit in the previous sentence.

**# Page 16 line 6-7 - I guess one might expect SST to be tightly coupled to forcing in ocean-only configurations since the hugely over-active surface flux response to SST changes ties the SST strongly to prescribed air temperature (e.g. see Hyder et al, 2018, Nat. Comms)?**
Done. This sentence now reads
*On the other hand, the global average Sea Surface Temperature (SST) in the models is dominated by the forcing field (as expected due to the lack of ocean-atmosphere feedback; see Hyder et al., 2018), with only weak variations between each cycle (Fig. 3b).*

**# Page 16 line 13 - Is it worth changing 'by the following analyses' to 'in the following sections'?**
Done.

**# Page 16 line 23-25 - Is it worth clarifying these two sentences and adding a bit more explanation?**
We feel these sentences are already sufficiently clear and detailed. Further investigation of the lower ACC transport at 0.1deg is beyond the scope of this already long paper.

**# Page 16 line 26-29 - Is it worth clarifying this text, perhaps by splitting the sentence into two sentences?**
Done. The first two sentences have been rearranged into three:
*To evaluate the capacity of the different model configurations to represent the mean state of the broad-scale horizontal ocean circulation, the simulated Mean Dynamic Sea Level (MDSL) is compared to the CNES-CLS13 Mean Dynamic Topography (MDT) observational product distributed by AVISO.*
*Both the MDSL and MDT data are averages over the years 1993--2012.*
*The global MDT product is a time-mean that combines data from satellites, surface drifters and in-situ measurements, as described by Rio et al. (2009).*

**# Page 16 line 32 - Is it worth changing this text to something along the lines of 'but a shallower MDSL, i.e. larger errors compared to AVISO, in the ACCESS-OM2-01 case.'**
Changed to
*There is a noticeable improvement in western boundary current structure in the higher resolution models, but a shallower MDSL minimum near Antarctica in the ACCESS-OM2-01 case, consistent with the reduced Drake Passage transport relative to observations (Fig. 4).*

**# Page 17 Figure 5 - This is a great figure but I am wondering if also plotting differences compared to the observational estimates might help to highlight the resolution dependence differences?**

Differences would not be very informative, since there is an offset between the observations and model, and in any case SSH gradients are of greater interest than SSH itself. We have therefore opted to keep this figure as-is.

**# Page 18 line 1-4 - It might be useful to state the frequency of the data used to generate the standard deviation (i.e. are they daily data)?**
This clarification has been added to the figure caption:
*The model standard deviations are calculated from $\left(\overline{\eta^2} - \bar{\eta}^2\right)^{1/2}$ using the time-means of $\eta$ and $\eta^2$ diagnosed at every baroclinic timestep, and therefore contain all model timescales.*

**# Page 18 line 19 - Is it worth adding 'as expected since the eddies are not resolved but parameterised in the low resolution configuration'?**
Done. The sentence now reads
*As expected, the ACCESS-OM2 configuration is unable to represent any significant SLA variability, since eddies are parameterised rather than explicitly resolved in this coarse-resolution model.*

**# Page 18 line 20 - Change to 'of larger sea-level variability. . .'**
Changed to "*enhanced*".

**# Page 19 line 15 and Page 20 line 3-4 - Can the dense shelf water really cascade down the slope accurately in the high resolution model since it still uses z-coordinate in which the bottom boundary layer is not at all well resolved (or is it perhaps also localised spurious but intermittent convection, perhaps in polynyas, that is making the bottom water)? In the UK ORCHESTRA project 1/12th degree sector model we find one need to use a sigma coordinate to get the dense water to cascade down slope, even at high resolution.**
Our 0.1 degree configuration produces dense water on the Antarctic continental shelf in four main locations (Ross Sea, Adelie coast, Prydz Bay region, Weddell Sea). This dense water flows down canyons on the slope at a rate of more than 10Sv across the 1000 m isobath, ultimately reaching the ocean's bottom layer in the abyss. This behaviour is the subject of a paper currently in preparation. The downslope flows are very similar to what is shown in this movie (from our previous MOM 0.1 degree configuration with CORE forcing, SIS sea ice and different bathymetry): https://youtu.be/8VMSF28J9H4?t=90

**# Page 21 line 12 - Is there perhaps also a contribution from numerical mixing?**
This is certainly another possibility. Simulations performed with a similar ocean model configuration show that numerical mixing is a significant contributor to the globally-integrated oceanic diathermal heat transport, and is larger at ¼-degree resolution (50 vertical levels) than at 1/10-degree (75 vertical levels) resolution (Holmes et al. 2019 JPO). Preliminary results indicate that numerical mixing varies significantly across the three resolutions of ACCESS-OM2 (study in progress). This variability in numerical mixing may influence the rate at which heat is moved downwards into the deep ocean and the rate at which the models drift away from observations. To indicate this possibility, we have added in parentheses "*(including differences in the level of numerical mixing, Holmes et al., 2019)*" after "resolved/parameterised processes".

Reference: Holmes, R. M., Zika, J. D., and England, M. H. (2019). Diathermal heat transport in a global ocean model. Journal of Physical Oceanography, 49(1):141–161. http://dx.doi.org/10.1175/JPO-D-18-0098.1

**# Page 22 line 6 - Perhaps SSTs and trends might also be expected to be tied to any trends in air temperatures in the forcing set over the forcing period in an ocean-only model?**

We agree that SST in ocean-only models is expected to be tied to the air temperature; all three model resolutions indeed show a warming trend over the historical period in the upper 100m. However, the model trend at depth does not seem to be tied to the atmospheric forcing. Under the same forcing, the 1deg and 0.25deg cases show a different drift to the 0.1deg; this difference may be attributed to differences in physics, numerical representation or the differing initial conditions between models.

**# Page 23 line 9 comment - In the Gulf Stream region, your models seem to behave rather differently to other ocean-models. For example, your low resolution model doesn't seem to have the typical very large (~5 degree) cold bias seen in our low resolution model (Storkey et al, 2018) but also doesn't show the clear improvement in this region with resolution. This could be due to the experimental design and particularly the new JRA55 forcing or the long spin ups you use. However, as I discussed in the earlier comment on the parameterisation section, it could also be due to your configuration. In particular, we wonder if the 1/4 degree model might be adversely affected by GM and all resolutions might be adversely affected by the no slip condition (see earlier comment for details).**
Please see our response above to major point 3 and the page 6 parameterization section.

**# Page 23 line 21 - Is it worth trying to explaining why you think it is due to the vertical diffusivity? From what I understand, the upwelling regions often get better, i.e. colder, at higher resolution? However, there are often substantial cloud biases in upwelling regions in atmospheric models and hence ocean forcing sets, usually too much downward short wave - this would be consistent across resolutions but I guess potentially could give rise to error cancellation?**
We have investigated this more closely and it is now less clear that vertical diffusivity is to blame, so we have replaced this sentence with
*However, this bias is larger in the high-resolution models (ACCESS-OM2-025 and ACCESS-OM2-01); the underlying cause of this bias is under investigation.*

**# Page 23 - line 33-35 -Does the z coordinate model really allow the dense water to cascade down the slope correctly (see earlier comment on this)?**
Yes - see response above.

**# Page 24 line 8-11 - Is it worth clarifying this sentence, perhaps by splitting it into two sentences?**
We have split this into two sentences as suggested:
*These biases are significantly reduced in ACCESS-OM2-01, although it shows a considerable negative temperature and salinity bias at subsurface low latitudes (also seen in Figs. 19, 20 and 23). This bias in ACCESS-OM2-01 is possibly due to excessive upwelling of colder and fresher water from the ocean interior and/or insufficient mixing-driven downward heat transport because of the lack of vertical background diffusivity and reduced numerical diffusion in this configuration (Sect. 2.1.4), but further investigation is required.*

**# Page 25 line 4 & fig 13 - This probably doesn't make much difference for your long averaging period but does the Trenberth and Carron method assume equilibrium, i.e. no heat content tendency.**
Trenberth and Caron (2001) do not assume an exact equilibrium for the ocean heat content tendency. As part of several adjustments that are made to the raw residual-inferred ocean heat transport calculations to account for biases and tendency, they subtract a constant 0.3Wm-2 globally from the fluxes to account for background warming (estimated from Levitus et al. 2000, Warming of the world ocean, Science 287, 2225-2229). However, they do ignore other changes in ocean heat storage (e.g. regional changes, interannual variability) because they do not have sufficient data available to assess them (although they do

argue that they are small). We have now included error bars for the Trenberth and Caron (2001) curves in Fig. 13 (see below) to capture these uncertainties.

[Figure]

**Page 26 line 7 - To me this seems to me like quite a strong inference from the results?**
The 2nd half of this sentence has been removed. It now reads
*Nonetheless, the results show that ACCESS-OM2-025 has a clear advantage over ACCESS-OM2 in representing heat transport at most latitudes, with the possible exception of 0--30° S.*

**Page 27 line 5 - Could you perhaps use 'comprehensive' instead of 'exhaustive'?**
Done

**Page 27 line 8 - Would it be clearer to change this text to 'A significant motivation for moving..'?**
Done

**Page 27 line 9 - Is it worth changing this to read 'mesoscale variability plays a critical dynamical role in the evolution. . .'?**
Done

**Page 28 line 5 - Is it worth adding 'representation of surface boundary condition in ocean-only runs' as a possible factor? For example, the prescribed air temperatures effectively give air an infinite heat capacity compared to water, which is clearly wrong. In the real world or a coupled model, air temperature adjusts very rapidly towards SSTs because of the much smaller heat capacity of air compared to water.**
We thank the reviewer for this suggestion. We have added:
*and the inability of the prescribed air temperature to adjust towards the SST, as would occur in a coupled ocean-atmosphere model*

**Page 28 line 7-10 - Is it possible to clarify this sentence and the associated Fig 15? Perhaps you could also label the regions you describing with arrows on the plot as I could not really follow exactly what you were referring to?**
This sentence has been clarified, and how reads:
*...there are two  distinct layers of high PV magnitude at about 900 m and 1400 m depth at 35-50° S separated by a slight minimum at about 1200 m...*

**Page 29 line 1-3 - Could this sentence be clarified?**

The sentence has been rewritten as

*It is not clear from our current understanding of mode water ventilation how the uptake of tracers (e.g. heat and carbon) into the ocean interior will be affected by the excessive winter mixed layer depth and differences in the distribution of PV in the models.*

**Page 29 line 4 and Fig 16 (and all subsequent sea level STD plots) - Might be worth mentioning the frequency of the data used for the standard deviations?**

Done. The caption in Fig 16 now references Fig 6, whose caption explains that all timescales are included (see above). Figs 17, 21, 22 and 24 now reference Fig 16 for brevity.

**Page 29 line 6 and Fig 17 - Could this suggest you need higher still resolution to represent the EAC?**

Presumably page 30 was intended. This is probably not a resolution issue - the EAC transport is more realistic in OFAM3 (e.g. 21.6Sv at Brisbane (with standard deviation 10.6Sv), vs. observed 19.8+/-9.3; Oke et al., 2013 table 2). OFAM3 is a different MOM 0.1-degree configuration, forced by ERA-I.

**Page 29 line 10-11 - Is it worth clarifying this text: 'indicator of the overall robustness of the model for this region'?**

Presumably page 30 was intended. This sentence now reads

*The Indonesian Throughflow (ITF) from the Pacific Ocean to the Indian Ocean is the only tropical inter-ocean pathway in the global ocean circulation and its magnitude through key straits is an important indicator of the fidelity of the model in this region.*

**Page 32 final paragraph - Some of the inferences seems a bit speculative and could perhaps benefit from more caveats?**

We think it was already pretty clear that these are speculative ideas requiring further work, but we have removed the comment about diffusivity. The relevant sentence now reads

*The strong Pacific thermocline in ACCESS-OM2-01 also appears in the zonal mean (Fig. 12) and Atlantic (Fig. 23); the cause of this bias is currently under investigation.*

**Page 34 line 3-8 - Is it worth adding a caveat to the final inference as it seems a bit speculative, to me at least?**

We have changed "likely" to "possible" in the final sentence, which now reads

*As such, it is possible that the models' representation of Pacific mode and intermediate waters will be affected by the poor representation of the deep salinity structure, which could, in turn, have implications for the local overturning circulation (Thompson et al., 2016).*

**Page 36 line 3 and Fig 22 - See earlier comments on representation of the Gulf Stream and its dependence on configuration (the viscosity and slip condition in particular).**

See our response to major point 3 above.

**Page 36 line 13 - Perhaps you could say 'the loop current in the Gulf of Mexico'?**

Done

**Page 37 Figure 22 - Again the Gulf Stream eddy variability improvement with resolution does not seem as good as often seen, e.g. by GFDL in their coupled models?**

See our response to major point 3 above.

**Page 39 line 1 and Fig 25 - From the figure it appears that the biases are much reduced in the high resolution model? This did not come over clearly to me from this paragraph so could the text be clarified?**

The paragraph already ends with "ACCESS-OM2-01 has minimal biases in this region", which we think is sufficiently clear.

**Page 40 line 7 - Is it worth referring to Fig 27 in first sentence and changing the text to include something along the lines of 'acceptable and very similar sea-ice extent and concentration at all three resolutions? I guess one might expect this because the sea ice is tied to the forcing fields in ocean-only runs by dramatic change in air temperature over ice and non-ice regions in the re-analysis, which are in turn tied to the observed sea-ice it uses as a lower boundary condition.**

Done. This sentence now reads

*Our coupled ocean-ice model runs yield acceptable and very similar timeseries of Arctic and Antarctic sea ice extent and volume at all three horizontal resolutions (Fig. 27); however, there are some shortcomings when compared with observations.*

**Page 44 line 2 - Perhaps this text could be changed to 'ocean mixed layer and associated SST and SST biases'?**

Done. This sentence how reads

*Further investigation is needed to understand the interplay between the oceanic mixed layer and associated SST and SSS biases, ice advection and thickness, and sea ice growth or melt processes, but is beyond the scope of this paper.*

**Page 45 line 1 - Summary - It is worth starting by referring back to the introduction and in both sections summarising the pros and cons of ocean-only versus coupled, problems with forcing and surface boundary condition errors and substantial potential for error cancellation which makes it hard to robustly identify resolution related benefits.**

Ocean-only models play an important role in building understanding of this important component of the fully coupled climate system, as has already been argued elsewhere (e.g. Griffies et al., 2009). See our response to major point 2 above. We now refer to potential difficulties related to the prescribed surface forcing:

*It remains an open question to what extent these resolution-dependent changes are related to changes in model dynamics or to differing error cancellation with forcing biases.*

**Page 45 line 1 - We use the term 'traceable' to refer to minimising configuration changes between resolutions.**

We now use this terminology:

*The model configurations are highly consistent (largely "traceable", in the terminology of Storkey et al., 2018); this feature makes the model suite ideal for studies investigating the sensitivity of solutions to model resolution.*

**Page 46 - line 3 - Again, is it worth stating that with the exception of the boundary currents and strong eddying regions, where resolution related benefits appear to be clear, one sees a mixture of improvements and degradation. This might be expected from the substantial errors in ocean forcing sets and the surface boundary condition, which provide a lot of potential for error cancellation.**

As noted above, we have added this sentence:

*It remains an open question to what extent these resolution-dependent changes are related to changes in model dynamics or to differing error cancellation with forcing biases.*

**Comments from Referee #2 (anonymous)**

**This is a thorough description of the performance of the ocean and sea-ice component of the Australian Community Climate and Earth Simulator (ACCESS-OM2). In my view this paper will be a really useful reference for anyone working with ACCESS but also for anyone looking for a good reference to illustrate the impact of model resolution on ocean simulations. The paper provides a nice overview of key ocean circulation features at horizontal resolutions of 1, 0.25 and 0.1 degrees. Even though it is well known that the circulation changes with resolution there are (to my knowledge) not too many examples of papers showing a systematic comparison of the global ocean circulation at non-eddying, eddy-permitting and eddy-rich resolutions.**

**The paper itself is clear and well written and perfectly fits the scope of Geoscientific Model Development. I strongly recommend publication subject to some clarifications of the minor points listed below.**

Our thanks to the review for their insightful comments which have been implemented as detailed below. We found this review to be very helpful in improving the manuscript.

**Comments:**

**1) Page 4, line 5: I was a bit surprised at the choice of 50 vertical levels for the ACCESS-OM2 and ACCESS-OM2-025 rather than 75 levels. Given that most HPC time is eaten up by ACCESS-OM2-01 the amount of time saved seems minimal. This seems to go against the philosophy outlined earlier, namely to keep the three resolu- tions as similar as possible. It would be worth explaining a bit more why this choice was made (e.g. 50 levels are sufficient at 1 and 0.25 degree as suggested in Stewart et al. 2017).**

There are two main reasons why we retained a 50-level configuration at lower resolutions. The first reason is that we want our primary 1° configuration to be as close as possible to Australia's CMIP6 model, as noted in the introduction. The second reason is that enhanced vertical resolution is only required when the horizontal resolution is sufficient to resolve more baroclinic modes (Stewart et al., 2017). As suggested by the reviewer, we have added this justification to the manuscript:

*The vertical grids are optimised for resolving baroclinic modes, based on the KDS grids recommended by Stewart et al. (2017), who suggest that finer horizontal resolution necessitates finer vertical resolution.*

**2) Page 6, line 10: Perhaps it would be worth noting why the choice of 0.004s-1 was made for the buoyancy frequency. Is this a typical value for this depth range?**

This is the default value in MOM5.1. We have added a note:

*(these three values are the defaults)*

**3) Page 9, line 5: "downwelling" → "downward"**

done

**4) Page 9, line 5: I suppose the river run-off is essentially climatological -especially during the first part of the forcing?**

River runoff is from Suzuki et al. (2017); as explained by Tsujino et al. (2018) this is daily, interannually-varying runoff from the JRA55 land surface model at 0.25° resolution, adjusted to match the observational estimates of Dai et al. (2009). We have added this clarification:

*JRA55-do also provides total runoff (river, calving and basal melt), at 0.25° resolution; river runoff is daily and interannually-varying (Suzuki et al., 2017), Greenland runoff is monthly climatological (Bamber et al., 2012), and Antarctic calving and basal melt are climatological means (Depoorter et al., 2013). Liquid runoff*

*is deposited at the coast in the top 40 m of the ocean, whereas solid runoff and basal melt are deposited as liquid at the ice shelf edge at the surface. The total runoff is spread horizontally if needed to keep the flux below a threshold (see Sec. 2.3.1).*

**5) Page 9, lines 29-31: I am not sure I really understand what is being done here. This could be read as if salinities were being nudged to WOA +/- 0.5 psu whenever, restoring fluxes try to push SSS outside that range. Obviously this is not what is being done since in Figure 11 there are regions where the salinity mismatch exceeds 0.5 psu (e.g. in ACCESS-OM2-01 off Grand Banks, in the Arctic and at the entry into the Caribbean Sea). This needs to be explained a bit more carefully.**

The SSS restoring flux is calculated using the SSS bias or +/-0.5 psu, whichever is larger. We have clarified this sentence:

*The SSS restoring flux is determined from the difference between model and WOA13 SSS; the restoring flux is calculated from the maximum of this difference or +/-0.5 psu, in order to avoid excessively large fluxes.*

**6) Page 10, lines 26-27: Is there any particular reason why the period from May 1984 – to April 1985 is chosen? Was this a year that was reasonably neutral for most major indices e.g. ENSO. . .etc i.e. "normal" year-ish"?**

Yes. A clarifying statement has been added.

*This 12-month period was chosen because it is particularly neutral in terms of the major modes of climate variability (Stewart et al., 2019).*

**7) Page 10, line 33: Do you mean that the Kuroshio separation was too far north? Also I suppose this refers to the 0.25 and 0.1deg versions as WBCs are so diffuse at 1 deg?**

Yes, this is exactly what we meant, but we were actually referring only to the 0.1° case. We have clarified this sentence:

*… produced a northward bias in the separation of the Kuroshio Current in the ACCESS-OM2-01 initial condition (i.e.\ the end of the RYF spinup), which largely disappeared under the subsequent interannually-varying forcing (Fig. 21).*

**8) Page 12, line 32: Is the timestep for ACCESS-OM2-01 400 or 450s? (The latter number is given in table 2).**

The tests used 400s but the production runs used 450s as shown in the table. This has now been clarified. In the MOM section:

*The tests used baroclinic timesteps of 5400 s, 1800 s and 400 s at 1°, 0.25° and 0.1° (respectively).*

In the CICE section:

*The tests used thermodynamic timesteps of 5400 s, 1800 s and 400 s at 1°, 0.25° and 0.1° (respectively).*

*Table 2 caption:*

In table 2 caption:

*Outline of model grid, size, cores and typical performance for production runs.*

*The timestep given is the ocean baroclinic timestep (at 0.1° this differs from the 400 s timestep used in the scaling tests), which equals the ice thermodynamic timestep (but is three times longer than the ice dynamic timestep at 0.1°).*

**9) Figure 3: To me it seems that for the globally averaged SST (panel b) the last pass looks different from passes 1-4. In passes 1-4 the globally averaged SST varies between about 17.8C to 18.4C and looks very similar for all passes. However, in pass 5 the SSTs vary between about 18 and 18.4C. This seems quite a large difference for a global average. It is noticeable that ACCESS-OM2-01 starts from much colder conditions (panel c). this linked to the stronger MOC in ACCESS-OM2-01? At the**

**end of the spinup the globally averaged temperature is about 0.2 deg colder than for the lower resolutions. Again, for a globally averaged value this is quite a big difference.**

We thank the reviewer for pointing out the scaling error in the last cycle in Figure 3b, which has now been corrected; see revised figure below.

Regarding global average temperature (presumably the reviewer meant panel a, not c), the two coarse models begin from the WOA climatology, whereas the 0.1 degree configuration begins from a 40-year spinup under 1984-5 repeat-year forcing, resulting in a colder initial state (as stated on p 16, line 5).

[Figure]

**10) Page 16, line 5: The initial cold drift cannot be seen in Figure 3a.**
Thank you for pointing this out. The drift primarily occurs during the RYF spinup, so is effectively part of the initial condition for this run. We have rewritten this sentence as follows:
*The ACCESS-OM2 and ACCESS-OM2-025 experiments and the RYF spinup prior to ACCESS-OM2-01 all start with nearly identical global average temperature from the WOA13 initial condition, from which ACCESS-OM2-025 drifts warm due to heat uptake (Fig. 3a), whereas ACCESS-OM2 remains relatively stable. ACCESS-OM2-01 is cold relative to the WOA13 initial state due to a cold drift during the repeat year spinup prior to the interannually-forced run shown in Fig. 3a.*
, and also appended
 *(not shown in the figure)*
to the sentence discussing SSS (Fig. 3c).

**11) Page 16, lines 23-25: It is interesting that a large ACC variability is only really seen in the first pass for 1 and 0.25 degrees where there is a pronounced and broad peak in transport during the first pass which is not seen at 0.1 deg. Is there a spike in the AABW formation during the first pass in ACCESS-OM2/-025?**
The variability in the first cycle of the lower resolution cases is indeed due to the formation of dense water in open ocean convection (as noted in section 4.1.4 of the manuscript). This open ocean convection does not occur in ACCESS-OM2-01. Accordingly, ACC transport during the preliminary spinup (reproduced in the figure below, but not shown in the manuscript) does not spike in the same way, but smoothly reduces towards the quasi-equilibrium value.

[Figure]

[referee did not provide a comment 12]

**13) Page 18, lines 16-18: There are clear differences for e.g. the Gulf Stream at 0.1 deg. The SSH variability suggests that Gulf Stream path may be too variable just after separating from Cape Hatteras. The SSH variability is confined to a broad patch North of Cape Hatteras rather than extending further east along the extension as suggested by AVISO.**
This sentence now points out these issues in the Gulf Stream region, which are discussed further in Sect. 4.2.4:
*The SLA variability magnitude in ACCESS- OM2-01 is closer to the observational estimate but still somewhat low, and with a differing pattern in the Gulf Stream region (discussed further in Sect. 4.2.4); the highest values are found south of the African continent in the Agulhas retroflection region, and the Gulf Stream and Kuroshio extension.*

**14) Page 18, lines 25-28: I don't think that this can really be inferred from Figure 6. . .**
The paragraph has been made clearer and now reads:
*Fig. 6d also shows broad regions of enhanced sea level variability at lower latitudes, with less amplitude than the western boundary currents. These patterns are typically associated with slower modes of climate variability. ENSO cycles drive variability in the Eastern Equatorial Pacific, and the Western Pacific, east of the Philippines and Papua New Guinea (Han et al., 2017; Mu et al., 2018). All resolutions simulate these patterns of variability associated with ENSO, though they all underestimate the observed variability by 10–20%. In the Indian ocean, variability is associated with both the Indian Ocean Dipole and ENSO (Li and Han, 2015). Anomalies in the tropical South Indian Ocean (5–15° S, 60–80° E) are driven by ENSO-related wind anomalies (Xie et al., 2002) and the associated pattern of variability is simulated in each model resolution. Enhanced variability from the coasts of Indonesia (Potemra and Lukas, 1999) and the West Australian coast in Fig. 6d extends westward into the Indian Ocean due to Rossby wave propagation. The pattern of variability in ACCESS-OM2-01 is consistent with this,*
*although somewhat weaker, whereas the pattern is much more muted in the coarser resolution simulations.*

**15) Page 18, line 30, Figure 7, Page 19, line 10: It would be nice to plot the overturning for the full range of densities i.e. not to cut off the lightest densities . I'd also suggest to expand the higher densities e.g. between 36.5 and 37.5 as this would show the AABW cell more clearly. This I feel could be relevant to understand differences in the ACC strength between the different resolutions (see comment 21) below.**
Done - Fig 7 now shows the full density range and uses a non-linear scale to show the AABW cell more clearly:

[Figure]

**16) Page 22, Figure 9: Is the data temporally filtered for the 1 and 0.25 degree resolu- tions? A seasonal signal is clearly visible in the surface layers of the 0.1 degree model but not at 1 and 0.25 degrees.**

Yes, as stated in the caption, they are annual averages at 1 and 0.25 degrees, and monthly averages at 0.1 degrees.

**17) Pages 23, lines 3-4, Figure 10: For some regions the anomaly patterns look quite different in ACCESS-OM2-01. For example in the Northern North Atlantic there is a large-scale warm bias and a strengthening cold bias in the southern SPG whereas the rest of the northern North Atlantic is warmer in ACCESS-OM2-01 than in the other cases.**

We agree, and have added the following comment:

*ACCESS-OM2-01 does, however, differ from lower resolutions in the northern North Atlantic ocean, with a stronger cold bias in the southern part of the subpolar gyre and a large-scale warm bias elsewhere; this is discussed further in Sect. 4.2.4.*

**18) Page 26, lines 7-8: To me the picture does not always seem as clear cut here. Between 0-30S ACCESS-OM2 has the best agreement with Ganachaud & Wunsch.**

This sentence has been clarified and now reads

*Nonetheless, the results show that ACCESS-OM2-025 has a clear advantage over ACCESS-OM2 in representing heat transport at most latitudes, with the possible exception of 0--30° S.*

**19) Page 27, line 15: Define all terms for the PV equation.**

Done. We have added

*where f is the Coriolis parameter, g is the acceleration due to gravity, ρ0 is the reference density, σ0 is the potential density anomaly referenced to 0 dbar, and z is the vertical coordinate, positive upwards*

**20) Page 29, Figure 15: I suppose the maximum/minimum mixed layer depths are from September (max) and March (min)?**

They are the maximum and minimum monthly mean mixed layer depths over the 300 months in the 25 years 1993-2017 (rather than climatological monthly max/mins). The caption has been amended for clarity:

*black (blue) lines represent the minimum (maximum) of the monthly mean mixed layer depth (defined by a 0.03 kg m$^{-3}$ density criterion) over 1993--2017.*

**21) Page 34, Figure 20: I can see that there seems to be a problem with the Antarctic mode water at the lower resolutions. However, what is equally pronounced in my view is the cold bias seen south of about 60S. In addition there is also a weaker cold bias at depth extending from the high southern latitudes to the northern end of the domain for the 1 and 0.25deg resolution versions of the model. This is also seen for the latitude-depth sections through the Atlantic and Indian Oceans (Figures 23 and 25) as well as for the zonally averaged temperatures shown in Figure 12. Could this explain why the ACC transport is weaker in ACCESS-OM2-01 than in ACCESS-OM2? The cold bias around Antarctica increases the meridional density gradient across the ACC which may explain the higher ACC transport in ACCESS-OM2 (where the cold bias is strongest). The coldest bias around Antarctica in ACCESS-OM2 may seem at odds with the overturning cell associated with AABW formation shown which is weaker than for the higher resolutions (Figure 7). However, my impression is that although weaker the overturning associated with AABW involves higher densities at 1 deg than at 0.25 and 0.1 deg. This would come out more clearly if the overturning is expanded for higher densities in Figure 7 (see comment 15).**

We agree with the reviewer that the deep cold bias is significant. This issue was primarily dealt with in section 4.1.4, in which we propose that unrealistic open ocean convection is key here. The reviewer is correct to infer that this bias may be relevant to the ACC transport differences between the resolutions, and we have added this suggestion as a new sentence at the end of this section, which now reads:
*In the Southern Ocean, the signature of Antarctic Bottom Water (AABW) shows a cold bias in ACCESS-OM2 and ACCESS-OM2-025 that spreads into the abyssal ocean. This bias is likely associated with large areas of anomalous deep (often full-depth) convection that appear every winter and spring in the eastern Weddell Sea and western Ross Sea in the ACCESS-OM2 and ACCESS-OM2-025 simulations. The behaviour of the two coarser models is typical of CMIP5 models, which produce bottom water by spurious deep-ocean convection rather than down-slope flows (Heuzé et al., 2013). In some models this convection is associated with spurious open-ocean polynyas (Heuzé et al., 2015); however, as in the GFDL CM2.5 model (Dufour et al., 2017), persistent open-ocean polynyas do not form in the ACCESS-OM2 simulations. The deep cold bias is much reduced in ACCESS-OM2-01, which has a more realistic AABW formation in the Antarctic continental shelf, with anomalous open-ocean convection confined to a much smaller and more interannually-variable region in the northeastern Weddell Sea (but has also had less time to drift away from climatology). The differences in Southern Ocean convection may partially explain the stronger ACC transport in the lower resolution configurations (Fig. 4).*

**22) Page 36, lines 8-9: Note that there are also uncertainties in the observational estimate of the barotropic streamfunction of DeVerdiere & Ollitrault. So I suppose that rather small features of the barotropic streamfunction such as this recirculation have to be taken with care.**

This paragraph has been rewritten and no longer contains this statement.

**23) Page 42, lines 3-4, Figure 27: The sea ice decline only seems too slow for ACCESS-OM2-025. For 1 and 0.1 deg there is a small positive bias compared to the observations that remains almost unchanged during the simulations but the long-term sea-ice decline looks very similar.**

We agree that the Arctic sea ice extent decline in Fig 27a seems slower at 0.25deg than at the other resolutions, and have added this sentence:
*The long-term trends in sea ice extent are also tracked by the 1° and 0.1° configurations, but the Arctic decline is slower than observed at 0.25° (Fig. 27a).*

**24) Caption Figure 16 (and other Figure captions): I suggest to replace "overlain" with "overlaid".**
Done. Changed "overlain" to "overlaid" throughout.

[revised manuscript text omitted]